# Universal Approximation with Softmax Attention

## Abstract

We prove that with linear transformations, both (i) two-layer self-attention and (ii) one-layer self-attention followed by a softmax function are universal approximators for continuous sequence-to-sequence functions on compact domains. Our main technique is a new interpolation-based method for analyzing attention's internal mechanism. This leads to our key insight: self-attention is able to approximate a generalized version of ReLU to arbitrary precision, and hence subsumes many known universal approximators. Building on these, we show that two-layer multi-head attention or *even* one-layer multi-head attention followed by a softmax function suffices as a sequence-to-sequence universal approximator. In contrast, prior works rely on feed-forward networks to establish universal approximation in Transformers. Furthermore, we extend our techniques to show that, (softmax-)attention-only layers are capable of approximating gradient descent in-context. We believe these techniques hold independent interest.

## 1 Introduction

We study the universal approximation ability of the attention mechanism (Vaswani, 2017). We prove that either *two-layer self-attention* or *one-layer self-attention followed by a softmax* (each equipped only with linear transformations) is capable of approximating any sequence-to-sequence continuous function on a compact domain. Different from previous studies (Yun et al., 2019; Jiang & Li, 2023; Takakura & Suzuki, 2023; Kajitsuka & Sato, 2023; Hu et al., 2024), our results highlight the expressive power of Transformers derived *only* from the attention module. By focusing exclusively on attention, our analysis demonstrates that the softmax operation itself suffices as a piecewise linear approximator. Furthermore, we extend this framework to broader applications, such as in-context learning (Brown et al., 2020; Bai et al., 2024), using the same attention-only architecture.

Prior studies of Transformer-based universality lean on deep attention stacks (Yun et al., 2019) or feed-forward (FFN) sub-layers (Kajitsuka & Sato, 2023; Hu et al., 2024) or strong assumptions on data or architecture (Takakura & Suzuki, 2023; Petrov et al., 2024). These results make it unclear whether attention alone is essential or auxiliary.

To combat this, we develop a new *interpolation-based* technique for analyzing attention[1]. We discretize the target function's output range into a uniform set of "anchors," embed them into the key–query–value transformations of softmax attention, and leverage softmax for a near-$\mathrm{argmax}$ selection. Effectively and surprisingly, this procedure turns attention into a one- or two-layer piecewise linear approximator (i.e., a generalized notation of ReLU). Consequently, attention alone suffices for universal approximation — no large FFN blocks or complex positional encodings are needed. This leads to our main results — even a single- or two-layer attention configuration suffices to approximate continuous functions for sequence-to-sequence tasks.

Beyond pure universal approximation, we also extend the same technique to *in-context learning* scenarios (Brown et al., 2020; Bai et al., 2024), showing that attention alone is capable of mimicking gradient-descent-like updates and approximate statistical models, akin to (Bai et al., 2024).

Altogether, our results reveal a minimalistic yet powerful principle: *attention itself* captures the core expressive power needed for sequence-to-sequence universality. By isolating attention from

---

[1]Please see Appendix E for discussion and comparison with prior interpolation-based methods for universal approximation.

other Transformer components, we affirm that the softmax-based mechanism has a direct route to approximate continuous mappings across a compact domain.

**Contributions.** Our contributions are four-fold:

- **Attention Approximation via Interpolation Selection.** We present a new *interpolation-based* method to analyze attention's internal mechanism. First, we partition the target function's range into uniformly spaced "anchors" and embed these anchors in the key-query-value transformations. Then, by approximating an argmax-style choice over these anchors, the softmax operation replicates piecewise linear behavior. Consequently, attention simulates an interpolation scheme for approximating known universal approximators. This insight eliminates reliance on auxiliary feed-forward layers to facilitate universal approximation of transformer architectures and highlights attention's inherent ability to approximate target functions with minimal overhead. See Figure 1 for a visualization.

- **One-Layer Single-Head (Softmax-)Attention Approximates Generalized ReLUs.** With our interpolation technique, we show that, for length-$n$ input, single-head and $H$-head (softmax-)attention approximate $n$ generalized ReLUs with $O(1/n)$, and $O(1/(nH))$ precision a token-wise manner (Theorems 3.4 and 3.9), respectively.

- **Two-Layer Multi-Head Attention Suffices to Be Sequence-to-Sequence Universal Approximator.** We show that (i) stacking two *attention-only* layers or (ii) one *attention layer followed by a softmax function* suffice for universal approximation of continuous sequence-to-sequence functions (Theorem 3.14 and Corollary 3.15 or a more Transformer-native extension in Appendix I). Compared to existing Transformer-based universal approximation results (Yun et al., 2019; Kajitsuka & Sato, 2023; Hu et al., 2024), our result demonstrates that attention alone provides the core expressiveness. These findings highlight the core expressive power of attention and depart from prior works that rely on deep attention or feed-forward sub-layers for universality guarantees.

- **In-Context Approximation and Gradient Descent.** We extend our techniques and results to in-context learning settings. We prove that attention approximates generalized ReLUs in-context (Theorem D.1). Furthermore, we show that multi-head softmax attention is capable of In-Context Gradient Descent (ICGD) (Theorem 4.2), and hence simulates various statistical models, such as ridge regression and generalized linear models. These results improve upon (Bai et al., 2024), which is limited to ReLU attention and sometimes requires FFNs to facilitate ICGD.

We highlight that our results are general and require minimal assumptions. Our theory assumes only the target function is continuous on the compact domain. No assumptions are made about the data or model, making our results and techniques widely applicable.

This generality departs from prior studies (Yun et al., 2019; Jiang & Li, 2023; Takakura & Suzuki, 2023; Kajitsuka & Sato, 2023; Hu et al., 2024). In particular, Yun et al. (2019); Kajitsuka & Sato (2023); Hu et al. (2024) rely on the concept of *contextual mapping* and assume a minimal separation condition on the data. Jiang & Li (2023) achieve Jackson-type universal approximation and require target space to have finite complexity measure, which acts like smoothness conditions in classical approximation theory. Takakura & Suzuki (2023) assume infinite-dimensional data. Moreover, while most existing works require many attention or FFN layers to achieve universal approximation of transformer blocks, our theory requires only one or two attention-*only* layers. This is, to the best of our knowledge, the first work on universal approximation of the attention mechanism.

**Roadmap of Theoretical Results.** Our main theorems progress in three steps. First, Theorem 3.4 provides a single-head warm-up result to demonstrate our interpolation selection technique, showing that attention with linear transform approximate truncated linear functions. Next, Theorem 3.9 extends this construction to multi-head attention setting aligning with practical transformer. Finally, Theorem 3.14 upgrades the multi-head formulation to sequence-to-sequence universal approximation, and Theorem 4.2 further applies the same ideas to in-context approximation of gradient descent.

**Related Work.** Appendix E offers additional details on the necessary related work discussed above.

## 2 PRELIMINARIES

**Notation.** We use lower-case letters (e.g., $v$) for vectors and upper-case letters (e.g., $M$) for matrices. The vector $e_k$ denotes the one-hot vector with a 1 in position $k$ and 0 elsewhere. Let $X \in \mathbb{R}^{d \times n}$ denote the input sequence, where $d$ is the token dimension and $n$ is the sequence length; intermediate inputs/outputs are denoted by $Z \in \mathbb{R}^{d \times n}$. For a matrix $A$, $A_{:,j}$ is its $j$-th column, $A_{i,:}$ is its $i$-th row,

and $A_{ij}$ is the entry in row $i$ and column $j$. We write $\|\cdot\|_\infty$ (or $\|\cdot\|_2$) for the vector $\infty$-norm (resp., 2-norm). For a matrix $Z \in \mathbb{R}^{d \times n}$, we define the $(p, q)$-norm as

$$\|Z\|_{p,q} := \big(\sum_{j=1}^{n}\big(\sum_{i=1}^{d} |Z_{ij}|^p\big)^{\frac{q}{p}}\big)^{\frac{1}{q}}, \quad \text{and} \quad \|Z\|_{\infty,\infty} := \max_{i,j} |Z_{ij}|.$$

For a function $f$, $\|f\|_{L_\infty} := \sup_{x \in \Omega} |f(x)|$ denotes its supremum norm on the given domain $\Omega$. More generally, we define the $L_p$-norm of function $f$ as

$$\|f\|_{L_p} := \big(\int_\Omega |f(x)|^p \, dx\big)^{\frac{1}{p}}. \tag{2.1}$$

The full summary of table of notion is in Appendix A.

**Attention Layer.** Let $X = [x_1, \ldots, x_n] \in \mathbb{R}^{d \times n}$ be input sequence of length $n$.

**Definition 2.1** (Attention Layer). *Let $H$ denote the number of heads of self-attention block. For any input sequence $X \in \mathbb{R}^{d \times n}$, we define the multi-head self-attention layer as*

$$\text{Attn}_m(X) = \sum_{h=1}^{H} W_V^{(h)} X \, \mathsf{Softmax}((W_K^{(h)} X)^\top W_Q^{(h)} X) W_O^{(h)},$$

*where $W_K^{(h)}, W_Q^{(h)} \in \mathbb{R}^{d_h \times d}$, $W_V^{(h)} \in R^{d_o \times d}$, $W_O^{(h)} \in \mathbb{R}^{n \times n_o}$ for $h \in [H]$. We use $\text{Attn}_s$ to denote single-head self-attention.*

Here we pick non-identical dimensions for $W_K, W_Q, W_V, W_O$ for generality of our analysis.

## 3 MAIN THEORY

In this section, we introduce an interpolation-based method to characterize the internal mechanism of a single-head attention block. Building on this technique, we establish the universal approximation capability of attention from single-head to multi-head, then to in-context learning, and to the general sequence-to-sequence setting. Specifically, in Section 3.1 we use a single-head self-attention with a sequence-wise linear transformation[2] to illustrate our inpertolation techniques. It approximates $n$ generalized ReLUs with $O(1/n)$ precision (Theorem 3.4). Build on top of this, in Section 3.2 we construct the multi-head version with token-wise linear map aligns with standard linear map in transformer. We demonstrate that increasing the number of heads reduces the required computational complexity per head for the same approximation error $\epsilon$. Explicitly, $H$-head attention yields $O(1/(nH))$ precision for approximating generalized ReLUs. In Section 4, we extend the method to in-context learning, showing that a single-head self-attention with a linear layer approximates $n$ generalized ReLUs in-context. Lastly, in Section 3.3, we prove that such a minimalist attention layer suffices as a sequence-to-sequence universal approximator.

### 3.1 ATTENTION APPROXIMATION AS INTERPOLATION SELECTION: APPROXIMATING GENERALIZED RELUS WITH $O(1/n)$ PRECISION

A key insight of our work is that single-head self-attention approximates a generalized ReLU function. Since ReLU neural network is a well-known universal approximator, this result implies that even a *minimalist* attention configuration subsumes many established universal approximators.

**Truncated Linear Functions as Generalized ReLUs.** We first formalize the generalized ReLU function using the concept of a truncated linear function $\text{Range}_{[a,b]}(\cdot)$:

**Definition 3.1** (Truncated Linear Function). *We define the truncated linear function as follows:*

$$\text{Range}_{[a,b]}(x) = \begin{cases} a & x \le a, \\ x & a \le x \le b, \\ b & b \le x. \end{cases}$$

Intuitively, the truncated linear function is a segment of a linear function, with output value ranging from $a$ to $b$.

**Definition 3.2** (Truncated Linear Model). *Define a truncated linear model as $\text{Range}_{[a,b]}(w^\top x + t)$, where $w \in \mathbb{R}^d$ is a learnable weight and $t \in \mathbb{R}$ is a bias.*

---

[2]We remark that, this sequence-wise linear transformation is not essential to our analysis and can be removed without loss of generality. We adopt it in Section 3.1 only for proof simplicity. Please also see Remark G.3.

We remark that the truncated linear model is a generalized ReLU and subsumes many universal approximators, including ReLU (Example H.1), Hard Tanh (Example H.2) and Clipped ReLU (Example H.3). Please see Appendix H for explicit expressions. These bounded activations appear in many practical scenarios where output constraints or gradient stability are desired.

Our goal here is to show attention is able to approximate $\text{Range}_{[a,b]}(w^\top x + t)$ with arbitrary precision. ReLU networks are classic universal function approximators (Lu et al., 2017; Sonoda & Murata, 2017; Hanin, 2019; Park et al., 2020). By demonstrating that single-head attention approximates $\text{Range}_{[a,b]}(x)$ to arbitrary precision, we show that *attention alone* replicates the essential behavior of ReLUs (and even more general piecewise linear transformations). This provides a foundation for proving broader universal approximation results using *only* attention mechanisms.

**Interpolation Scheme.** To approximate $\text{Range}_{[a,b]}(\cdot)$ with attention, we partition $[a, b]$ into $p$ uniform segments:

**Definition 3.3** (Interpolation). *Let $[a, b] \subset \mathbb{R}$ be an interval with $a \leq b$ and let $p \in \mathbb{N}^*$ be a positive integer. We define*

$$\widetilde{L}_0^{[a,b]} := a, \quad \widetilde{L}_p^{[a,b]} := b, \quad \widetilde{L}_k^{[a,b]} := a + \frac{k}{p}(b-a), \quad k = \{0, ..., p-1\}.$$

*Hence, $\widetilde{L}_0 < \widetilde{L}_1 < \cdots < \widetilde{L}_p$ forms a uniform partition of $[a, b]$. We also write*

$$\Delta L := \widetilde{L}_k^{[a,b]} - \widetilde{L}_{k-1}^{[a,b]}, \quad k \in [p].$$

*We often omit the superscript $[a, b]$ when the context is clear.*

Importantly, these segments $\{\widetilde{L}_k\}_{k=0}^{p-1}$ serve as "targets" for the attention mechanism in later parts.

**Interpolation Method for Attention Approximation.** Now we present our fundamental result — a single-head self-attention with a linear transformation is capable of approximating truncated linear models in a token-wise manner. Let $X = [x_1, ..., x_n] \in \mathbb{R}^{d \times n}$ be the input sequence.

**Theorem 3.4** (Single-Head Attention Approximates Truncated Linear Models). *Fix real $a < b$, and let $\text{Range}_{[a,b]}(\cdot)$ be the truncation operator from Definition 3.1. Let $\epsilon_0 \geq 0$. For a precision parameter $p > n$ and $\beta \geq (\ln(p-2) - \ln \epsilon_0)/((\Delta L)^2/2)$, there exists a single-layer, single-head self-attention $\text{Attn}$ with a linear transformation $A : \mathbb{R}^{d \times n} \to \mathbb{R}^{(2d+d_o+2) \times p}$, such that $\text{Attn} \circ A : \mathbb{R}^{d \times n} \to \mathbb{R}^{d_o \times n}$ satisfies, for any $i \in [n]$,*

$$\|\text{Attn} \circ A(X)_{:,i} - \text{Range}_{[a,b]}(w_i^\top x_i + t_i)e_{\widetilde{k}_i}\|_\infty \leq \underbrace{\max\{|a|, |b|\} \cdot \epsilon_0}_{\text{finite-}\beta \text{ softmax error by Lemma F.1}} + \underbrace{\frac{b-a}{p}}_{\text{interpolation error}}.$$

*Here $e_{\widetilde{k}_i}$ is a one-hot vector with a value of 1 at the $\widetilde{k}_i$-th index and 0 elsewhere, and*

$$k_i := \underset{k \in \{0, 1, 2, \cdots, p-1\}}{\text{argmin}} |x_i^\top w + t - \widetilde{L}_k| \quad \text{where} \quad \widetilde{k}_i := G(k_i) \in [d_o]. \tag{3.1}$$

*Here $k_i \in \{0, ..., p-1\}$ is the index of the interpolation point closest to the $i$-th token ($i$-th truncated linear model). For all $i \in [n]$, $G : \{0, ..., p-1\} \to [d_o]$ denotes any set-to-set function sending the interpolation index $k \in \{0, ..., p-1\}$ into a position index $\widetilde{k} \in [d_o]$ specifying in the desired row index of the output.*

Intuitively, Theorem 3.4 ensures that a single-head self-attention layer with a suitable linear layer is capable of approximating $n$ "truncated" linear models with token-level granularity. We accomplish this via an interpolation method. To elaborate, a few remarks are in order.

**Remark 3.5** (Interpolation Selection with Softmax Attention). *Here, we provide a high-level overview of our proof techniques: we approximate the target function (truncated linear models of interest) using interpolation points and leverage softmax attention for interpolation point selection. We also provide conceptual visualization in Figure 1.*

*For the $i$-th column (token) of $\text{Attn} \circ A(X) \in \mathbb{R}^{d_o \times n}$, our goal is to approximate the one-hot vector $\text{Range}_{[a,b]}(w_i^\top x_i + t_i)e_{\widetilde{k}_i}$, where $\text{Range}_{[a,b]}(w_i^\top x_i + t_i)$ is a scalar (the truncated linear output), and $e_{\widetilde{k}_i}$ is a one-hot vector of dimension $d_o$. To achieve this, we require at least $n$ column vectors in $\text{Softmax}(K^\top Q)$ to represent potential outputs of the $n$ truncated linear models.*

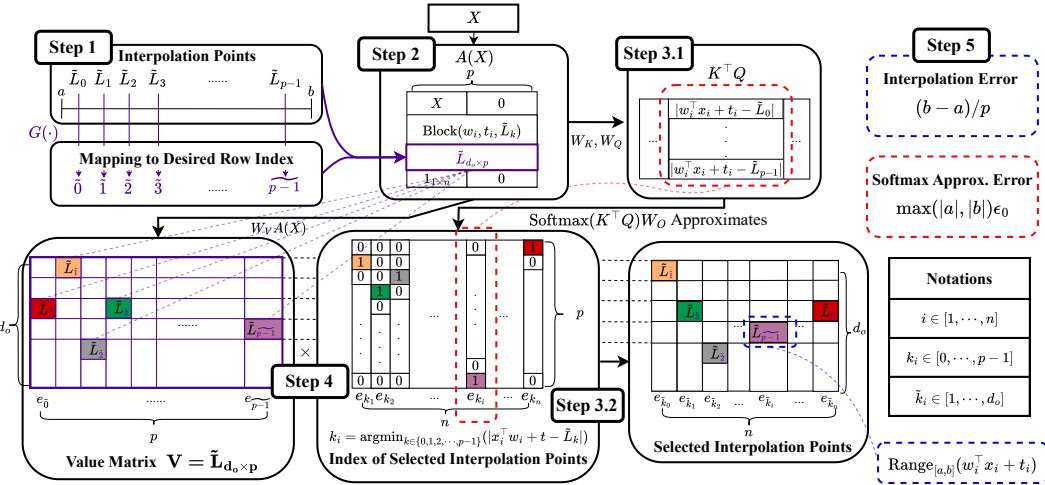

Figure 1: **Visualization of Proposed Interpolation Technique (Theorem 3.4).** Every step in the figure corresponds to a step in the proof sketch in Section 3.1. Our goal is to use softmax attention mechanism to approximate $n$ truncated linear models $\mathrm{Range}_{[a,b]}(w_i^\top x_i + t_i)$ for $i \in [n]$, and hence establish universality. To achieve this, we first divide the output range $[a, b]$ into $p$ interpolation points, and encode them into the value matrix $V$. Then, we treat the attention score $\mathsf{Softmax}(K^\top Q)$ as a *selector* to select an interpolation point closest to the desired output from $V$. Specifically, each column of $\mathsf{Softmax}(K^\top Q)_{:,i}$ (for $i \in [n]$) approximates an one-hot vector $e_{k_i}$, where $k_i$ is the index of closest interpolation point to $\mathrm{Range}_{[a,b]}(w_i^\top x_i + t_i)$. Hence, when multiplying with $V$, $V \cdot \mathsf{Softmax}(K^\top Q)$ selects out the closest interpolation points for every truncated linear model from $V$. The **same color** across matrices indicates the same interpolation point chosen by the softmax function. The color purple indicates how $G$ maps each interpolation point index $k$ into the desired row index $\widetilde{k}$. The **grey dashed lines** highlight that the position of $\widetilde{L}_k$ in the value matrix and the output matrix are the same, indicating each closest interpolation point of $i$-th token is placed correctly at the output. The **blue dashed line** illustrates the interpolation error, while the **red dashed line** shows the softmax approximation error. For simplicity, we highlight the error only for a token $x_i$.

*Since the output of* $\mathrm{Range}_{[a,b]}(w_i^\top x_i + t_i)$ *lies within* $[a, b]$ *(Definition 3.1), we apply the interpolation scheme (Definition 3.3) to partition* $[a, b]$ *into* $p$ *points. For each* $i \in [n]$, *there exists an interpolation point* $\widetilde{L}_{k_i}$ *closest to truncated linear model* $\mathrm{Range}_{[a,b]}(w_i^\top x_i + t_i)$, *where*
$$k_i := \mathrm{argmin}_{k \in \{0,1,\cdots,p-1\}}(-2x_i^\top w_i - 2t_i + \widetilde{L}_0 + \widetilde{L}_k) \cdot k \text{ is the selected interpolation index.}$$

*Our key idea is to*

1. ***Select Interpolation Index.*** *Express* $\{k_i\}_{i\in[n]}$ *as one-hot column vectors* $\{e_{k_i}\}_{i\in[n]} \in \mathbb{R}^p$.
2. ***Approximate Anchors Design.*** *Design* $K^\top Q$ *such that* $\mathrm{Softmax}(K^\top Q)_{:,i}$ *approximates* $e_{k_i}$ *for all* $i \in [n]$.
3. ***Recover the Selected Interpolation Point in Value Space.*** *Encode interpolation point* $\{\widetilde{L}_k\}_{k=0}^{p-1}$ *into* $V$ *such that, for each* $i \in [n]$, *the largest entry of* $V\mathrm{Softmax}(K^\top Q)_{:,i}$ *to be the interpolation point* $\widetilde{L}_{k_i}$, $(k_i \in [p])$. *Recall that,* $\widetilde{L}_{k_i}$ *is selected as the closet interpolation point to* $\mathrm{Range}_{[a,b]}(w_i^\top x_i + t_i)$

*We visualize in Figure 1 and summarize as follows:*

$$\max_{j\in[d_o]}[\overbrace{V}^{\substack{p \text{ column vectors containing} \\ p \text{ interp. points } \{\widetilde{L}_k\}_{k=0}^{p-1}}} \overbrace{\mathrm{Softmax}(K^\top Q)_{:,i}}^{\substack{\text{approximate one-hot} \\ \text{representation of selected} \\ \text{interpolation index } k_i}}]_{j,i}$$

$$= \underbrace{\widetilde{L}_{k_i}}_{\substack{\mathrm{argmin} \\ \{\widetilde{L}_0,\cdots,\widetilde{L}_{p-1}\}}} |\mathrm{Range}_{[a,b]}(w_i^\top x_i + t_i) - \widetilde{L}_k| + \underbrace{\text{error}}_{\text{(By finite-}\beta \text{ softmax approximation Lemma F.1)}}. \quad (3.2)$$

*This way we use attention mechanism to perform interpolation approximation to each truncated linear model output.*

**Remark 3.6** (Why $A(\cdot)$ and Its Connection to Practice). *To accomplish* (3.2), *we embed the $p$ interpolation points into $A(X)$ such that $K^\top Q = A(X)^\top W_K^\top W_Q A(X)$ contains these points among its entries. The linear map $A(\cdot)$ here includes a sequence-wise operation to only simplify the proof. It is not a standard component of Transformer architecture. Importantly, Theorem 3.4 serve as the simplest illustrative example for our interpolation selection techniques. In Theorem 3.9, we extend this technique to the multi-head setting and replace the sequence-wise $A(\cdot)$ with a token-wise linear transformation that aligns with practical Transformer architectures. See also Appendix B.*

**Remark 3.7** (Meaning of $k_i$, $\widetilde{k}_i$, and $G(\cdot)$). *Here, we clarify the distinction between $k_i$ and $\widetilde{k}_i$. The difference lies in their roles within the interpolation and output spaces. Given the $i$-th token $x_i$, $k_i \in \{0, \ldots, p-1\}$ identifies the closest interpolation point $\widetilde{L}_{k_i}$ to the target value $\mathrm{Range}_{[a,b]}(w_i^\top x_i + t_i)$. In contrast, $\widetilde{k}_i \in [d_o]$ is an output coordinate index: it specifies in which coordinate of the $d_o$-dimensional output vector we place the selected point $\widetilde{L}_{k_i}$ (grey dashed lines in Figure 1). The mapping $G : \{0, \ldots, p-1\} \to [d_o]$ connects these two roles by assigning each interpolation index $k$ a coordinate $\widetilde{k} := G(k)$, and for each token $i$ we then have $\widetilde{k}_i = G(k_i)$ (purple font in Figure 1). In the simplest case one take $G(k) \equiv 1$ for all $k \in \{1, ..., p-1\}$, so that every $\widetilde{L}_{k_i}$ is placed in the first row of the output matrix. This flexibility allows $G$ to be tailored to the scenarios considered. See Appendix G.2 for detailed discussion.*

**Remark 3.8** (Universal Approximation Implications). *Since $\mathrm{Range}_{[a,b]}(\cdot)$ acts as a bounded ReLU, demonstrating that a single-head attention layer approximates it arbitrarily well implies that attention alone is capable of replicating and generalizing known piecewise linear networks. We leverage this result to establish the universal approximation properties of attention-based architectures in broader settings (e.g., multi-head, seq-to-seq) in subsequent sections.*

*Proof Sketch.* We design the key-query matrices such that, for each token $x_i$, the column $\mathrm{Softmax}(K^\top Q)_{:,i}$ selects the closest interpolation point $\widetilde{L}_{k_i}$ to $w_i^\top x_i + t_i$. This yields a single-head attention output approximating the truncated linear model at each token.

Our proof consists of five conceptual steps:

**Step 1: Partitioning.** Partition the range $[a, b]$ into $p$ segments, defining interpolation points $\{\widetilde{L}_k\}_{k=0}^{p-1}$, so that for any $\mathrm{Range}_{[a,b]}(x_i^\top w_i + t_i) \in [a, b]$, there exists a nearest interpolation point $\widetilde{L}_{k_i}$ satisfying

$$\left| x_i^\top w_i + t_i - \widetilde{L}_{k_i} \right| \leq \frac{b-a}{p}, \quad \text{for all } i \in [n].$$

**Step 2: Linear Encoding.** Apply a linear transformation
$$A : \mathbb{R}^{d \times n} \to \mathbb{R}^{(2d+d_o+2) \times p},$$
augmenting the input $X$ with additional rows and columns to include: (i) the input tokens $x_i$, (ii) the weights $\{w_i\}_{i=1}^n$ and biases $\{t_i\}_{i=1}^n$ (to construct truncated linear models), (iii) the interpolation points $\{\widetilde{L}_k\}_{k=0}^{p-1}$, and (iv) auxiliary entries for constructing the desired key–query scores.

**Step 3: Key-Query Construction.** Design $W_K, W_Q$ such that each column of $K^\top Q \in \mathbb{R}^{p \times p}$ has entries of the form
$$[K^\top Q]_{k,i} = (-2x_i^\top w_i - 2t_i + \widetilde{L}_0 + \widetilde{L}_k) \cdot k.$$
The rationale behind this design is the equivalence between the following two objectives (see (G.8) for a proof):
$$\underset{k \in \{0,1,\ldots,p-1\}}{\mathrm{argmin}} (-2x_i^\top w - 2t + \widetilde{L}_0 + \widetilde{L}_k) \cdot k = \underset{k \in \{0,1,\ldots,p-1\}}{\mathrm{argmin}} |x_i^\top w + t - \widetilde{L}_k|,$$
where the second objective selects the interpolation point $\widetilde{L}_{k_i}$ (see (3.1)) closest to $x_i^\top w + t$ among $p$ interpolation points. Thus, $[K^\top Q]_{k,i}$ indicates the interpolation point $\widetilde{L}_k$ closest to $w_i^\top x_i + t_i$. Using Lemma F.1, the softmax function approximates the argmax, ensuring that the column vector $\mathrm{Softmax}(K^\top Q)_{:,i}$ approximates a one-hot selection of $\widetilde{L}_{k_i}$, the closest interpolation point. Specifically, $\mathrm{Softmax}(K^\top Q)_{:,i}$ approximates $e_{k_i} \in \mathbb{R}^p$.

**Step 4: Value Mapping.** Design $W_V$ such that $V = W_V A(X)$ encodes the interpolation points $\{\widetilde{L}_k\}$ from $A(X)$ into the column vectors of $V \in \mathbb{R}^{d_o \times p}$. Specifically, for $k \in \{0, \ldots, p-1\}$,

the $k$-th column of $V$ is $\widetilde{L}_k e_{\widetilde{k}}$. Then, multiplying $V$ with $\mathrm{Softmax}(K^\top Q) \in \mathbb{R}^{p \times p}$, where the $i$-th column approximates $e_{k_i} \in \mathbb{R}^p$ (from **Step 3**), gives

$$\underbrace{V}_{d_0 \times p} \underbrace{\mathrm{Softmax}(K^\top Q)_{:,i}}_{p \times 1} \in \mathbb{R}^{d_0}.$$

The largest entry of this product approximates the closest interpolation point $\widetilde{L}_{k_i}$. Post-multiplication by the projection matrix $W_O$ discards the extra $(p - n)$ columns beyond the original sequence length $n$.

**Step 5: Error Control.** We must bound two types of errors. (i) Interpolation Error: Partitioning $[a, b]$ into $p$ segments ensures each $w_i^\top x_i + t_i \in [a, b]$ lies within $(b - a)/p$ of some interpolation point $\widetilde{L}_{k_i}$. (ii) Softmax Approximation Error: Using $\mathrm{Softmax}_\beta$ instead of a hard $\arg\max$ introduces $\epsilon_0$ (Lemma F.1). Moreover, because $\max_k |\widetilde{L}_k| \leq \max\{|a|, |b|\}$, the softmax spread contributes at most $\max\{|a|, |b|\} \cdot \epsilon_0$. Consequently, for each token $i$,

$$\underbrace{\left| \mathrm{Range}_{[a,b]}(w_i^\top x_i + t_i) - \widetilde{L}_{k_i} \right|}_{\text{Interpolation error} \leq \frac{b-a}{p}} + \underbrace{\|\mathrm{Softmax}_\beta(\cdot) - e_{k_i}\| \cdot \max\{|a|, |b|\}}_{\text{Softmax approx. error} \leq \max\{|a|, |b|\}\epsilon_0} \leq \frac{b - a}{p} + \max\{|a|, |b|\}\epsilon_0.$$

By tuning $p$ and the softmax $\beta$, we make these errors arbitrarily small, proving that single-head attention approximates $\mathrm{Range}_{[a,b]}(w_i^\top x_i + t_i)$ for each token with arbitrary precision. Please see Appendix G.2 for a detailed proof. $\square$

In summary, increasing the partition size $p$ (reducing $\epsilon$) improves the approximation to arbitrary precision $O(1/n)$. As $p > n$, a longer input sequence (with larger $n$ and hence larger $p$) yields a larger attention score matrix (i.e., $\mathrm{Softmax}(K^\top Q)$), enabling higher-resolution interpolation. This highlights the expressive power of the minimalist attention layer. In contrast, typical Transformers rely on multi-head structures and feed-forward layers.

### 3.2 $H$-HEAD ATTENTION APPROXIMATES GENERALIZED RELU WITH $O(1/(nH)))$ PRECISION

In Section 3.1, we show how a single-head self-attention layer approximates $n$ truncated linear models by embedding $p$ interpolation points into its key–query–value matrices. Here, we extend this construction to *multi-head* attention. We show that $H$-head attention improves the approximation precision from $O(1/n)$ (Theorem 3.4) to $O(1/(nH))$ for approximating generalized ReLU. This establishes a tradeoff between the number of heads and the per-head complexity, determined by the size of the linear layer $A$. Intuitively, more heads allow each head to focus on a smaller subset of interpolation points, reducing the partition size $p$ needed per head to achieve the same overall error.

**Theorem 3.9** (Multi-Head Attention Approximate Truncated Linear Models). *Fix real numbers $a < b$, and let the truncation operator $\mathrm{Range}_{[a,b]}(\cdot)$ follow Definition 3.1. For a precision parameter $p > n$ with $\epsilon = O(1/p)$, number of head $H = p/(n - 2)$ there exists a single-layer, $H$-head self-attention $\mathrm{Attn}^H$ with a linear transformation $A : \mathbb{R}^{d \times n} \to \mathbb{R}^{(d+n) \times n}$, such that $\mathrm{Attn}^H \circ A : \mathbb{R}^{d \times n} \to \mathbb{R}^{d_o \times n}$ satisfies, for any $i \in [n]$,*

$$\|\mathrm{Attn}^H \circ A(X)_{:,i} - \mathrm{Range}_{[a,b]}(w_i^\top x_i + t_i)e_{\widetilde{k}_i}\|_\infty \leq \underbrace{\max\{|a|, |b|\} \cdot \epsilon_0}_{\text{finite-}\beta \text{ softmax error}} + \underbrace{\frac{b - a}{(n - 2)H}}_{\text{interpolation error}}.$$

*Here $e_{\widetilde{k}_i}$ is a one-hot vector with a value of $1$ at the $\widetilde{k}_i$-th index and $0$ elsewhere, and*

$$k_i := \operatorname*{argmin}_{k \in \{0,1,2,\cdots,p-1\}} |x_i^\top w + t - \widetilde{L}_k| \quad where \quad \widetilde{k}_i = G(k_i) \in [d_o].$$

*Here $k_i \in \{0, ..., p - 1\}$ is the index of the interpolation point closest to the $i$-th token ($i$-th truncated linear model). For all $i \in [n]$, $G : \{0, ..., p - 1\} \to [d_o]$ denotes any set-to-constant function sending the interpolation index $k \in \{0, ..., p - 1\}$ into a position index $\widetilde{k} \in [d_o]$ specifying in the desired row index of the output.*

**Corollary 3.10** (Approximation Error). *The approximation error scales as $O(1/(nH))$.*

**Tradeoff: Multiple Heads $H$ vs. Partition Size $p$.** Whereas the single-head construction in Theorem 3.4 places all $p$ interpolation points into one attention head (possibly requiring $\ell = p - n$ extra columns in $A(X)$), multi-head attention splits these $p$ points across different heads.

Consequently, each head only needs to handle a fraction of the total interpolation range, allowing for fewer effective points per head. In practice, this reduces per-head computation (both in forming $K, Q, V$ and in performing the softmax) while preserving the same global partition resolution (i.e., the same overall approximation error $\epsilon$).

*Proof Sketch.* Our proof strategy follows Theorem 3.4, but distributes the interpolation workload:

1. **Partition the Points Across Heads.** Suppose we have $H$ attention heads and want to approximate $\mathrm{Range}_{[a,b]}(\cdot)$ with total precision $O(1/p)$. We split the $p$ interpolation points into $H$ groups, each group containing $p/H = n - 2$ points.

2. **Local Encoding.** In each head, we store (in $V$) only the portion of the $(n-2)$ interpolation points assigned to that head. We also add two sentinel columns representing "no contribution" outside the local interpolation range. This ensures that if a token's value is not covered by head $h$, the head $h$ remains inactive (outputs zero).

3. **Head Selection.** We design the key–query matrices such that each token $x_i$ "selects" the head whose local interpolation range covers $w_i^\top x_i + t_i$. Softmax in that head's output then acts as an approximate $\arg\max$ among the assigned interpolation points. We also discuss the case where the value $w_i^\top x_i + t_i$ happen at the shared endpoint of two adjacent heads.

4. **Combine Heads.** Lemma G.7 tells us every token is either (1) strictly inside one head's interval or (2) exactly on a shared endpoint of two consecutive intervals. In the interior case only that head contributes; at a boundary the two neighbouring heads output a convex sum of the same two grid points. Either way the total error is the interpolation error $(b-a)/p$ plus the softmax error $\epsilon_0$ added at most $(O(H) + |b|)\epsilon_0$, $\epsilon_0$ can be arbitrarily small by setting a large enough $\beta$.

By splitting $p$ points across $H$ heads, each head handles only $p/H$ points. Thus, the *per-head* complexity decreases while achieving the same global approximation $\epsilon = O(1/p)$. Moreover, $\epsilon = (1/(nH))$ by $H = p/(n-2)$. Please see Appendix G.3 for a detailed proof. $\qquad\square$

### 3.3 Sequence-to-Sequence Universal Approximation by Self-Attention

Building on the results so far, we now show that a two-layer multi-head attention — augmented with simple linear transformations — achieves *sequence-to-sequence* universal approximation.

**Overview of Our Proof Strategy.** Theorem 3.4 establishes that a single-head or multi-head attention layer is capable of approximating generalized ReLUs (truncated linear models) on a token-by-token basis. To extend this capability to more general sequence-to-sequence settings, we:

- **Step 1: Construct a Two-Layer ReLU Network as a Vector-to-Scalar Universal Approximator.** We construct a two-layer ReLU neural network in Lemma 3.11 that serves as a universal approximator for any continuous function $f : \mathbb{R}^N \to \mathbb{R}$ on a compact domain, with a $p$-norm error.

- **Step 2: Approximate the Constructed ReLU Neural Network with Attentions.** In Lemma 3.12, we prove that one layer multi-head attention plus one layer single head attention approximate the constructed ReLU neural network from Lemma 3.11. This proves that two-layer attention approximates any continuous function $f : \mathbb{R}^{d \times n} \to \mathbb{R}$ on compact domain with a $p$-norm error.

- **Step 3: Extend to Sequence-to-Sequence Approximation.** We generalize Lemma 3.12 to sequence-to-sequence approximation in Theorem 3.14. This involves decomposing an arbitrary continuous map $f : \mathbb{R}^{d \times n} \to \mathbb{R}^{d \times n}$ into $d \cdot n$ scalar-valued functions $f_{ij} : \mathbb{R}^{d \times n} \to \mathbb{R}$. We approximate each $f_{ij}$ with different attention layers construct in Lemma 3.12, and then aggregate these scalar outputs into a matrix form with an additional multi-head attention layer. This shows that a two-layer attention mechanism suffices as a sequence-to-sequence universal approximator. We also extend to $\infty$-norm error in Theorem F.2.

Below, we elaborate on the conceptual steps in detail and defer the proofs to appendices.

**Step 1: Universal Approximation via Two-Layer ReLU Networks.** We start with the universal approximation theorem of a two-layer feed-forward network with ReLU activation. Let $\mathcal{X} \subset \mathbb{R}^N$ be a compact domain, and $\|f\|_{L_p}$ be the $L_p$-norm following (2.1), for function $f$ on its given domain.

**Lemma 3.11** (Explicit Construction of ReLU Neural Network as Universal Approximator)**.** *Let* $f : \mathcal{X} \to \mathbb{R}$ *be a continuous function defined on* $\mathcal{X}$*. For any* $\epsilon > 0$*, there exists a two-layer feed-forward neural network* $\mathrm{FFN} : \mathbb{R}^N \to \mathbb{R}$ *with ReLU activation functions such that for all* $x \in \mathcal{X}$

$$\|\mathrm{FFN}(x) - f(x)\|_{L_p} \leq \epsilon. \tag{3.3}$$

*Proof.* Please see Appendix G.4 for a detailed proof. □

With the constructed ReLU NN, we proceed to step 2, approximating it using a two-layer attention mechanism. We achieve this by utilizing Theorem 3.4 that attention approximate $\mathrm{Range}_{[a,b]}(w_i^\top x_i + t_i)$ in a tokenwise manner.

**Step 2: Approximate the Constructed ReLU Neural Network with Attentions.** Now we prove the universal sequence-to-scalar approximation of multi-head attention.

**Lemma 3.12** (Sequence-to-Scalar Universal Approximation of Two Layer Attention)**.** *For any continuous function* $f : \mathbb{R}^{d \times n} \to \mathbb{R}$ *of compact support* $\mathcal{X}$*, and any* $\epsilon > 0$*, we prove that when composed with linear transformations, there exists a one layer multi-head attention* $\mathrm{Attn}_m$ *stacked with one layer single-head attention* $\mathrm{Attn}_s$ *composed with linear connections* $A_1$ *and* $A_2$*, such that*

$$\|f - \mathrm{Attn}_s \circ A_2 \circ \mathrm{Attn}_m \circ A_1\|_{L_p} \leq \epsilon.$$

*Proof Sketch.* We begin by discretizing the domain $\mathcal{X} = [-B, B]^{d \times n}$ into a finite grid $G_D$. For each grid point $v^{(j)} \in G_D$, we define a "bump" function $R_{v^{(j)}}(X)$ that is approximately 1 when $X$ is near $v^{(j)}$ and approximately 0 otherwise. Next, using Lemma G.14 and Lemma 3.11, we construct a multi-head attention layer (plus a linear mapping) that collectively approximates these bump functions via $|G_D| \cdot d$ heads, achieving an $\infty$-norm error of at most $|G_D| \cdot d \cdot \epsilon_0$. We then form a second linear map encoding the function values $\left[ f\left(v^{(1)}\right), \ldots, f\left(v^{(|G_D|)}\right) \right]$ alongside the approximated bump functions, organizing them into a 2-row matrix. Finally, a single-head attention layer — using softmax as a near-$\arg\max$ — selects the grid value $f\left(v^{(j)}\right)$ associated with whichever $v^{(j)}$ is nearest to $X$. This yields a piecewise approximation to $f$ within any desired error tolerance. Please see Appendix G.5 for a detailed proof. □

Note that in Lemma 3.12 the function of the second single-head attention is to utilize the softmax function to to pick out the closest grid point $v$ to the input $X$. Hence we derive a one layer multi-head attention version of Lemma 3.12 in below.

**Lemma 3.13** (Single-Layer Multi-Head Attention Version of Lemma 3.12)**.** *For any continuous function* $f : \mathbb{R}^{d \times n} \to \mathbb{R}$ *of compact support* $\mathcal{X}$*, and any* $\epsilon > 0$*, we prove that when composed with linear transformations, there exists a one layer multi-head attention* $\mathrm{Attn}_m$ *followed by a* Softmax *function and attached with linear connections* $A_1$ *and* $A_2$*, such that*

$$\|f - A_2 \circ \mathsf{Softmax} \circ \mathrm{Attn}_m \circ A_1\|_{L_p} \leq \epsilon.$$

*Proof.* Please see Appendix G.6 for a detailed proof. □

We now state our final result of sequence-to-sequence universal approximation of two-layer attention.

**Step 3.** By combining $dn$ two-layer attention blocks $\mathrm{Attn}_s \circ A_2 \circ \mathrm{Attn}_m \circ A_1$ from Lemma 3.12, we approximate each output entry of $f(X)$ individually.

**Theorem 3.14** (Two-Layer-Sequence-to-Sequence Approximation)**.** *For any continuous function* $f : \mathbb{R}^{d \times n} \to \mathbb{R}^{d \times n}$ *of compact support* $\mathcal{X}$*, and any* $\epsilon > 0$*, we prove that when composed with linear transformations, there exists a two layer multi-head attention* $\mathrm{Attn}_m$ *stacked with one layer multi-head attention* $\mathrm{Attn}_m$*, attached with linear connection* $A_1$ *and* $A_2$*, such that*

$$\|f - \mathrm{Attn}_m^{(2)} \circ A_2 \circ \mathrm{Attn}_m^{(1)} \circ A_1\|_{L_p} \leq \epsilon.$$

**Corollary 3.15** (Single-Layer Attention Sequence-to-Sequence Approximation)**.** *There exists a single-layer multi-head attention* $\mathrm{Attn}_m$ *followed by a* Softmax *function and attached with linear connections* $A_1$ *and* $A_2^{ij}$ *for* $i \in [d], j \in [n]$*, such that*

$$\left\|f - \sum_{i \in [d], j \in [n]} A_2^{ij} \circ \mathsf{Softmax} \circ \mathrm{Attn}_m^{(1)} \circ A_1\right\|_{L_p} \leq \epsilon.$$

*Proof Sketch.* We first decompose the target function $f : \mathbb{R}^{d \times n} \to \mathbb{R}^{d \times n}$ into $dn$ scalar subfunctions $\{f_{ij}\}$, where $f_{ij} : \mathbb{R}^{d \times n} \to \mathbb{R}$ for $i \in [d], j \in [n]$. By Lemma 3.12, each $f_{ij}$ is approximated by one-layer multi-head attention $(\mathrm{Attn}_m)$ combined with one-layer single-head attention $(\mathrm{Attn}_s)$ and linear transformations $(A_1, A_2)$, yielding a per-subfunction error

$$\|f_{ij}(X) - \mathrm{Attn}_s^{ij} \circ A_2 \circ \mathrm{Attn}_m \circ A_1(X)\|_p \leq \epsilon_{\text{scaler}}.$$

The first attention layer forms bump functions $R_{v^{(k)}}(x)$ to locate the relevant region of $X$, which does not depend on any particular $f_{ij}$. We then aggregate the $dn$ approximations into a single matrix output by defining a second multi-head attention layer as

$$\mathrm{Attn}_m^{(2)} = \sum_{i \in [d], j \in [n]} E^{ij} \mathrm{Attn}_s^{ij},$$

where $E^{ij} \in \mathbb{R}^{d \times n}$ is all zeros except for a single 1 in the $(i, j)$ position. Thus, each subfunction's approximation is placed in the correct row-column entry, yielding the full sequence-to-sequence approximation of $f$. The same logic applies to the proof of Corollary 3.15. Please see Appendix G.7 for a detailed proof. $\qquad\square$

# 4 IN-CONTEXT LEARNING

We extend the interpolation selection technique and Theorem 3.4 to the in-context learning setting (Brown et al., 2020; Bai et al., 2024). In Theorem 4.2, we show that standard softmax attention perform in-context gradient descent, broadening the results established for ReLU attention in (Bai et al., 2024). Specifically, we demonstrate that softmax attention is capable of doing in-context gradient descent on convex loss functions. We first define the problem setting similar to theirs.

**Definition 4.1** (In-Context Learning Problem Formulation). *The sequential input $X$ in the in-context learning scenario is defined as*

$$X := \begin{bmatrix} x_1 & x_2 & \cdots & x_n \\ y_1 & y_2 & \cdots & y_n \\ w & w & \cdots & w \\ 1 & 1 & \cdots & 1 \end{bmatrix},$$

*where $(x_i, y_i), \quad i \in [n]$ denote the input-output pairs. $w$ parametrize the model connecting $x_i$ and $y_i$, and is altered(trained) between layers. The task of in-context learning is to using the given input-output pairs $(x_i, y_i)$ to predict the output of a newcome input $x_u$.*

In this setting, we prove a multi-head Softmax attention is capable of doing in-context gradient descent on loss functions parametrized by $w^\top x_i \quad (i \in [n])$ and $t$(as linear coefficient and bias), as well as giving an according prediction to the output on $x_u$.

**Theorem 4.2** (In-Context Gradient Descent). *Let $l : \mathbb{R} \times \mathbb{R} \to \mathbb{R}$ be any $C^1$ loss function defined on $(w^\top x_i, y_i)$. With input $X$ in the form of Definition 4.1, when $X$ is bounded, there exists a multi-head self-attention $\mathrm{Attn}_m$ with skip connections and each attached with a linear layer, such that for any $\epsilon > 0$, irrelevant of $X$, we have*

$$\left\| \mathrm{Attn}_m \circ A(X) - \begin{bmatrix} x_1 & \cdots & x_n \\ y_1 & \cdots & y_n \\ w - \eta \nabla L(w) & \cdots & w - \eta \nabla L(w) \\ 1 & \cdots & 1 \end{bmatrix} \right\|_\infty \leq \epsilon,$$

*where $\eta$ denotes the learning rate and $L(w) := \sum_{i=1}^n l(w^\top x_i, y_i)$ is an empirical loss upon the given input-output pairs.*

Please see Appendix D for the proof sketch and Appendix G.10 for a detailed proof.

We note that in the original proof of (Bai et al., 2024), they rely on the approximation ability of ReLU neural networks to approximate the derivative of the loss function. Therefore, they use ReLU-based attention to approximate a sum of ReLU functions. In contrast, by leveraging Theorem 3.4, we show softmax attention approximates generalized ReLU function by approximating truncated linear models, and hence approximates in-context gradient descent. Since softmax attention is the dominant mechanism used in practice, our results provide a more realistic foundation for understanding in-context learning tasks.In Appendix D, we provide two research directions inspired by Theorem 4.2.

We defer the concluding discussion and numerical studies to Appendices B and C due to page limits.

ETHIC STATEMENT

This paper does not involve human subjects, personally identifiable data, or sensitive applications. We do not foresee direct ethical risks. We follow the ICLR Code of Ethics and affirm that all aspects of this research comply with the principles of fairness, transparency, and integrity.

REPRODUCIBILITY STATEMENT

We ensure reproducibility on both theoretical and empirical fronts. For theory, we include all formal assumptions, definitions, and complete proofs in the appendix. For experiments, we describe model architectures, datasets, preprocessing steps, hyperparameters, and training details in the main text and appendix. Code and scripts are provided in the supplementary materials to replicate the empirical results.

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

# Supplementary Material

## IMPACT STATEMENT

Given the formal nature of this work, we expect no negative social impact.

## LLM USAGE DISCLOSURE

We used large language models (LLMs) to aid and polish writing, such as improving clarity, grammar, and conciseness. We also used LLMs for retrieval and discovery, for example exhausting literature to identify potential missing related work. All technical content, proofs, experiments, and results are original contributions by the authors.

## A    TABLE OF NOTATION

Table 1: Notations and Symbols

| Symbol | Description |
|---|---|
| $d$ | Input dimension of each sequence element |
| $n$ | Input sequence length |
| $H$ | Number of attention heads |
| $p$ | Number of interpolation anchors ($p > n$) |
| $d_o$ | Output (value) dimension per head |
| $X = [x_1, \ldots, x_n]$ | Input sequence matrix in $\mathbb{R}^{d \times n}$ |
| $x_i$ | $i$-th token (column) of $X$ |
| $w_i, t_i$ | Weight and bias for the $i$-th truncated linear model |
| $f$ | Target continuous function being approximated |
| $\mathcal{X}$ | Compact domain of inputs considered |
| $\mathrm{Range}[a,b](\cdot)$ | Truncated linear (generalized ReLU) with range $[a,b]$ |
| $a, b$ | Lower and upper bounds of truncation (with $a < b$) |
| $\widetilde{L}_k$ | $k$-th uniformly spaced interpolation anchor in $[a,b]$ |
| $\Delta L$ | Anchor spacing: $\widetilde{L}_k - \widetilde{L}_{k-1}$ |
| $k$ | Interpolation index, $k \in \{0, ..., p-1\}$ |
| $\widetilde{k}$ | Row index of interpolation point $\widetilde{L}_k$ in the output space, $\widetilde{k} \in [d_o]$ |
| $G(\cdot)$ | Mapping $G : [0, ..., p-1] \to [d_o]$ with $\widetilde{k} = G(k)$ |
| $k_i$ | Index of anchor closest to $\mathrm{Range}_{[a,b]}(w_i^\top x_i + t_i)$ |
| $\widetilde{k}_i$ | $i$-th token's row index of the chosen anchor in the output space |
| $e_k$ | One-hot vector with $1$ in position $k$ |
| $\beta$ | Inverse temperature in softmax |
| $\delta$ | Gap between the two largest input entries of softmax |
| $\gamma$ | Gap between the first largest and third largest input entries of softmax |
| $\epsilon$ | Desired overall approximation error |
| $\epsilon_0$ | Softmax approximation error |

# B    DISCUSSION AND CONCLUSION

We establish the universal approximation theory of simple softmax attention layer for any continuous sequence-to-sequence function on a compact domain. Our key technique is to cast attention as softmax-based selection mechanism of the interpolation points in the output domain (Remark 3.5). This enables softmax attentions with simple linear transform to approximate the generalized ReLUs (and hence many known universal approximators) in a token-wise manner (Theorems 3.4 and 3.9). Based on this, we derive the universal approximation theory for sequence-to-sequence functions using (i) two softmax-attention layers (Theorem 3.14) or (ii) one softmax-attention layer followed by a softmax function (Corollary 3.15). We also extend our results to in-context learning (Section 4).

**Connecting to Practical Attention/Transformer.** We remark that our sequence-to-sequence universal approximation in Section 3.3 uses the single-head result of Theorem 3.4 for simplicity of presentation. The same proofs hold if we replace it with the multi-head result of Theorem 3.9. That is, the two theorems are interchangeable for establishing universal sequence-to-sequence approximation. The main differences lie in that the $A$ mapping is a sequence-wise linear operation in Theorem 3.4, while $A$ is an ordinary token-wise linear transform in Theorem 3.9. The construction of $A$ in Theorem 3.9 aligns with a practical transformer/attention. It is a token-wise linear layer augmented with positional encoding like an ordinary embedding layer. We further derive the sequence-to-sequence universal approximation result based on Theorem 3.9 explicitly in Appendix I.

**Extensions.** We further show that standard softmax attention simulates In-Context Gradient Descent (ICGD, Theorem D.4), in contrast to prior works only establish ICGD for modified attention variants (e.g., ReLU (Bai et al., 2024) or linear attention (Dai et al., 2022; Von Oswald et al., 2023)).

**Limitation and Future Work.** Our in-context learning result simulates gradient descent but does not establish universal approximation. Our theoretical results suggest that attention has the potential to be a universal in-context approximator, which we leave for future work.

## C  EXPERIMENTAL STUDIES

In this section, we provide proof-of-concept numerical experiments to back up our theoretical results. We divide our experiments into the following two objectives.

- **Objective 1: Validating the Proposed Interpolation Selection Scheme (Theorem 3.4 and Theorem 3.9).** We aim to verify the theoretical approximation rates (Figure 2): $O(1/p)$ with respect to the number of interpolation points $p$, linear scaling in the interval length $|b-a|$, and $O(1/H)$ in terms of the number of heads $H$ for multi-head attention. Furthermore, we print out the attention weights to determine that each column of $\mathsf{Softmax}(K^\top Q)$ becomes close to one-hot indicators selecting interpolation points (Figure 3).

- **Objective 2: Sequence-to-Sequence Approximation (Theorem 3.14).** We create synthetic data for a sequence-to-sequence task to verify that the approximation rate is again $O(1/p)$ and $O(1/H)$. We use two-layer ReLU network with flatten input $X_{\text{flatten}} \in \mathbb{R}^{dn}$ to mix information from all $dn$ input dimensions. In this formulation, the output token at each position depends on the entire input sequence. The result in Figure 4 shows that it aligns with the theoretical result.

### C.1  VALIDATING THE $O(1/p)$ AND $O(1/H)$ APPROXIMATION RATES

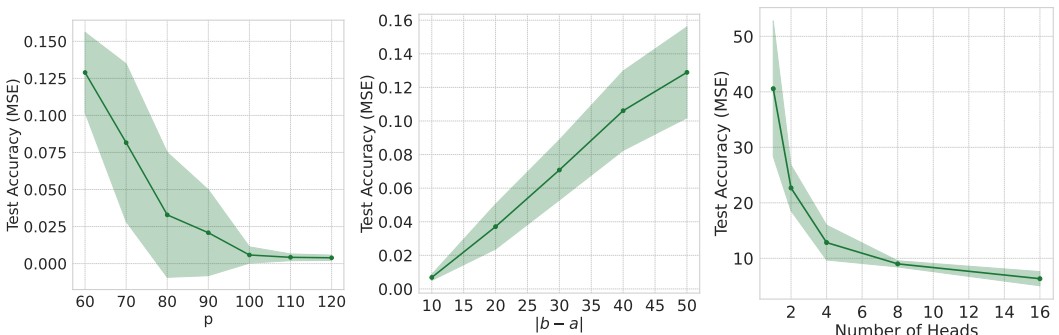

Figure 2: **Ablation Study for Three Key Parameters in Our One-Layer Attention (Theorem 3.4 and Theorem 3.9).** All the results align with the theoretical analysis that the approximation error scales as $O(1/p)$, $O(1/H)$, and grows linearly in $|b-a|$. We report test accuracy (MSE) as the mean and one standard deviation (shaded region) over 10 random seed runs. The synthetic dataset consists of 1000 samples with a 80/20 train-test split. All other hyperparameters remain fixed for three experiments ($d = 10$, $n = 50$, hidden dimension $= 32$, learning rate $= 0.001$, epoch $= 50$ and batch size $= 32$). The experiments are run on an NVIDIA A100 GPU.

**Model Architecture.**  We train single/multi-head single-layer softmax attention to model truncated linear model, with an extra linear layer $A$ applied on the input and create $p - n$ extra empty tokens. For the attention weight experiments, we guide the model by encoding the interpolation points in the last row of the key matrix $K$ as in the proof of Theorem 3.4. We also encode the interpolation point onto random column indices in the value matrix $V$ as $\widetilde{k}_i$ in the theorem.

**Data Generation.**  For the experiments of the truncated linear model, we represent each sample as $X = [x_1, \cdots, x_n] \in \mathbb{R}^{d \times n}$ and with ground truth have each label $y_i$ encoded on the first rows of the matrix $Y \in \mathbb{R}^{d \times n} := [y_1 e_1, y_2 e_1, \cdots, y_n e_1]$. For $i \in [n]$, we first fix a truncated linear model by sampling a weight vector $w_i \sim N(0, I_d)$ and bias $t_i \sim N(0, 1)$. These token-specific parameters are fixed for all samples (in one run). Then for each sample $X$, we draw every token $x_i \sim \text{Uniform}(-5, 5)$, and compute the label $y_i$ as

$$y_i = \text{Range}_{[a,b]}(w_i^\top x_i + t_i).$$

We generate $N$ samples using this process, we set $N = 1000$ and train for 50 epochs. For attention weight experiments, the difference is that we encode the ground truth $y_i$ onto the same random column indices as described in the model architecture paragraph.

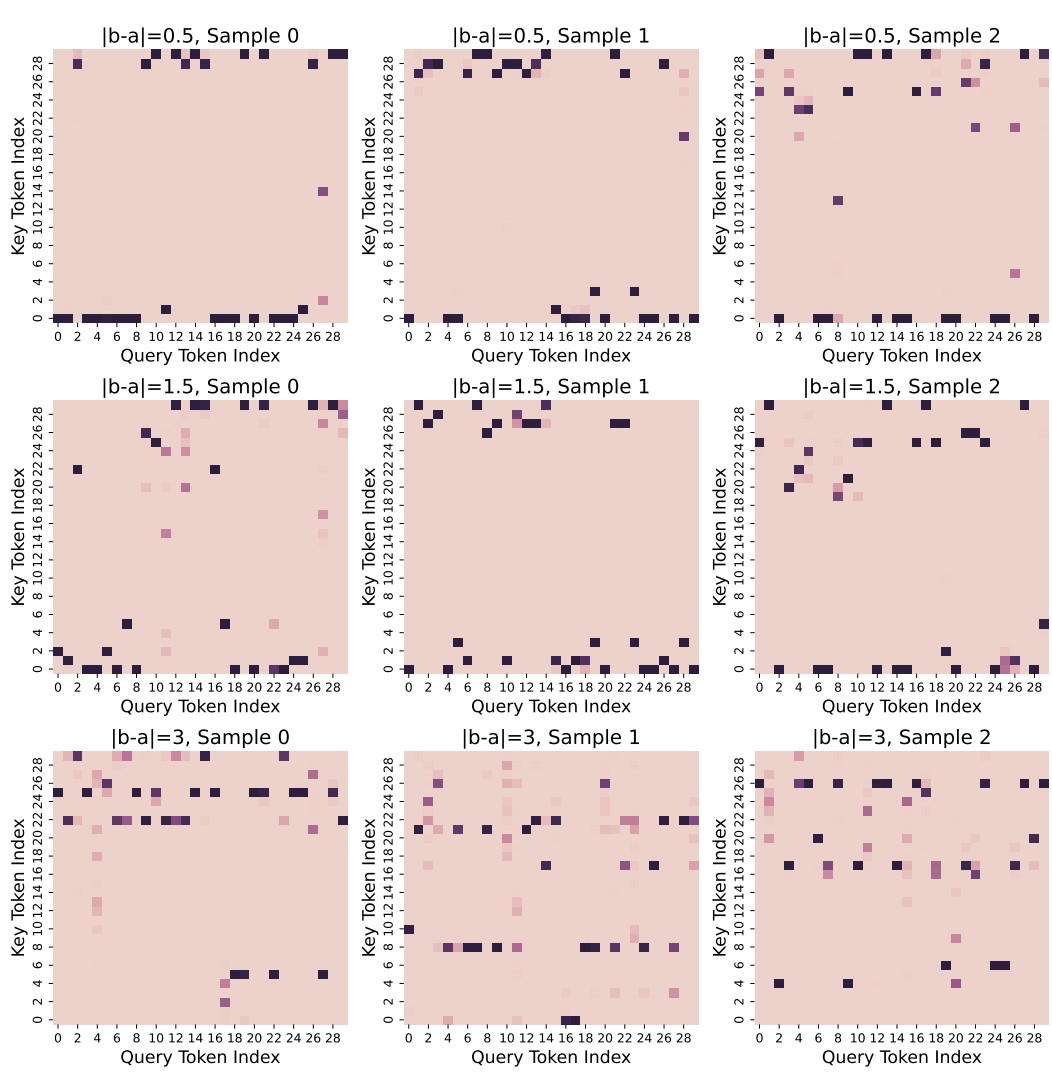

Figure 3: **Attention Heatmap for** $|b - a| = 0.5, 1.5, 3$. The figure shows the attention heatmap $\mathrm{Softmax}(K^\top Q)$ ((G.5)) for 3 random test samples with parameters $p, n = 30$. In particular, for smaller truncation intervals ($|b - a| = 0.5$), the attention distribution concentrates on boundary interpolation points, as our theoretical analysis anticipates. When expanding the truncation interval width, the attention weights transition to selecting intermediate interpolation points. We set the hyperparameters to 100 epochs, learning rate $= 0.001$, batch size 32, hidden dimension $= 10$, $\beta = 30$, and random seed $= 1234$.

**Metrics.** We train the model with the following Mean Squared Error (MSE) loss

$$\mathcal{L}_{\text{MSE}} = \sum_{i=1}^{n} (y_i - \widehat{y_i})^2,$$

where $\widehat{y_i} \in \mathbb{R}^d$ is the prediction of the attention layer.

**Results.** We present our findings in Figure 2 and Figure 3.

- **Approximation Performance.** Figure 2 shows that the MSE follows the theoretical approximation rates. It decreases as $O(1/p)$ when we increase the number of interpolation points $p$. It also scales linearly with the truncation interval $|b - a|$ and behaves as $O(1/H)$ with $H$ heads in multi-head attention. Increasing $p$ not only reduces the approximation error but also stabilizes training, as indicated by the smaller standard deviations across 10 runs.

- **Attention Heatmaps.** Figure 3 confirms the "one-hot" interpolation selection phenomenon. For a small truncated range $a = -0.5, b = 0.5$, most ground truths of $y$ lie at $\widetilde{L}_0$ and $\widetilde{L}p - 1$. As the figure shows, each token $x_i$ (query index) puts most of its attention on the 0-th and 29-th keys (the interpolation points). When we increase $|b - a|$, the attention weight spreads across more key indices.

## C.2 SEQUENCE-TO-SEQUENCE APPROXIMATION RATES

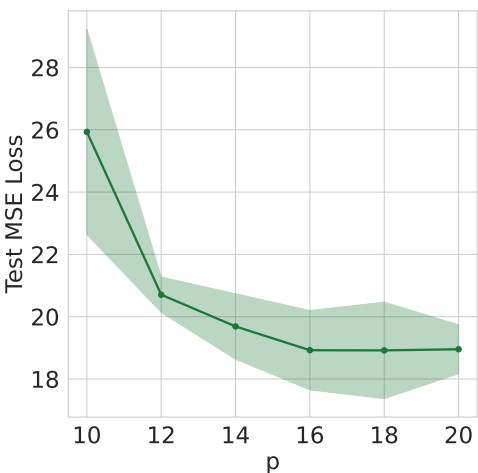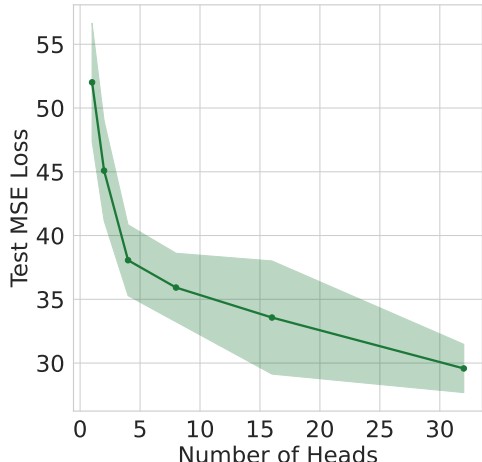

Figure 4: **Ablation Study for Two Parameters $p$ and $H$ in Theorem 3.14.** The results align with our theoretical approximation rate as $O(1/p)$ and $O(1/H)$. We report test accuracy (MSE) as the mean and one standard deviation (shaded region) over 10 random seed runs. The synthetic dataset consists of 50000 data points with $d = 5$ and a $80/20$ train-test split. For both experiments, we set the learning rate $= 0.001$, epoch $= 3$, and batch size $= 32$. For the number of interpolation points $p$ experiment (left figure), to speed up the training process, we set $n = 10$ and the hidden dimension of the model to be 16. For number of heads experiment (right figure), we increase the sequence length to $n = 20$ to make the task harder so we can see the trend when increasing the number of heads, and also increase the hidden dimension to 32 so it can be divided by $H = 32$. The experiments are run on an NVIDIA A100 GPU.

**Model Architecture.** We train a small model with 2-layer multi-head attention with linear mapping $A_1$ and $A_2$ as in Theorem 3.14. For the experiment of number of interpolation points, we set $H = 2$ to speed up the experiment. All the parameters are randomly initialized instead of hard-set to the form of weight in the proof.

**Data Generation.** Same as Appendix C.1, each sample is in the form of $X = [x_1, \cdots, x_n] \in \mathbb{R}^{d \times n}$, and we generate each $x_i$ from a uniform distribution $\text{Uniform}(0, 1)$. We generate targets via

a global sequence-to-sequence mapping. Concretely, for each sample $X$, we first flatten it into a vector $X_{\text{flatten}} \in \mathbb{R}^{dn}$ and then pass it through a two-layer ReLU network:

$$Y_{\text{flatten}} = W_2 \text{ReLU}(W_1 X_{\text{flatten}} + b_1) + b_2,$$

where $W_1 \in \mathbb{R}^{m \times dn}$, $b_1 \in \mathbb{R}^m$ are the weights and bias of the hidden layer, and $W_2 \in \mathbb{R}^{dn \times m}$, $b_2 \in \mathbb{R}^{dn}$ are those of the output layer. Finally, $Y_{\text{flatten}}$ is reshaped into a sequence in $Y = [y_1, \cdots, y_n] \in \mathbb{R}^{d \times n}$. The above data generation ensures that each output token $y_i \in \mathbb{R}^d$ is a function of the entire input sequence. In our experiments, we generate $N = 50000$ samples and again use a $80/20$ train-test split. The input dimension and hidden dimension of the ReLU network to generate synthetic data are $d = 5$ and $m = 10$.

**Metrics.**  We use Mean Squared Error (MSE) loss

$$\mathcal{L}_{\text{MSE}} = \sum_{i=1}^{n} (y_i - \widehat{y_i})^2,$$

where $\widehat{y_i} \in \mathbb{R}^d$ is the prediction of the attention layer.

**Results.**  As shown in Figure 4, the approximation rate is again in the trend of $O(1/p)$ and $O(1/H)$. This result validates the theoretical analysis for sequence-to-sequence approximation of the attention-only layer considered in Theorem 3.14. The decrease in MSE error indicates this small model (2-layer and hidden dimension = 16) captures the global dependencies in the sequence-to-sequence task.

In summary, our empirical observations confirm that (i) attention emulates truncated linear model via "selection" of key–value pairs, and (ii) a straightforward attention-only network learns nontrivial sequence-to-sequence dependence. Both align with the approximation rate provided by our theoretical results.

# D DETAILED RESULTS AND DISCUSSION OF IN-CONTEXT LEARNING

Here we provide an application to showcase the generality of our theory and techniques.

## D.1 ATTENTION APPROXIMATES TRUNCATED LINEAR MODELS IN-CONTEXT

We extend the interpolation selection technique and Theorem 3.4 to the in-context learning setting (Brown et al., 2020; Bai et al., 2024). The next theorem shows that when the length-$n$ input includes weights $\{w_i\}_{i\in[n]}$ and biases $\{t_i\}_{i\in[n]}$, attention is able to approximate $n$ truncated linear models $\{\mathrm{Range}_{[a,b]}(w_i^\top x_i + t_i)\}_{i\in[n]}$ in-context.

**Theorem D.1** (Attention Approximates Truncated Linear Models In-Context). *Fix real numbers $a < b$, and let the truncation operator $\mathrm{Range}_{[a,b]}(\cdot)$ follow Definition 3.1. Let the input be*

$$X = \begin{bmatrix} x_1 & x_2 & \cdots & x_n \\ w & w & \cdots & w \\ t_1 & t_2 & \cdots & t_n \end{bmatrix} \in \mathbb{R}^{(2d+1)\times n},$$

*where $\{w, x_i\}_{i\in[n]}$ are bounded. Let $\epsilon_0 \geq 0$. For a precision parameter $p > n$, there exists a single-layer, single-head self-attention $\mathrm{Attn}$ with a linear transformation $A : \mathbb{R}^{(2d+1)\times n} \to \mathbb{R}^{(2d+d_o+2)\times p}$, such that $\mathrm{Attn} \circ A : \mathbb{R}^{d\times n} \to \mathbb{R}^{d_o\times n}$ satisfies, for any $i \in [n]$,*

$$\|\mathrm{Attn} \circ A(X)_{:,i} - \mathrm{Range}_{[a,b]}(w_i^\top x_i + t_i)e_{\widetilde{k}_i}\|_\infty \leq \underbrace{\max\{|a|, |b|\} \cdot \epsilon_0}_{\textit{finite-}\beta\textit{ softmax error}} + \underbrace{\frac{b-a}{p}}_{\textit{interpolation error}},$$

*each $w_i$ is a elementwise multiplication of $w$ by a vector $v_i$. Here $e_{\widetilde{k}_i}$ is a one-hot vector with a value of $1$ at the $\widetilde{k}_i$-th index and $0$ elsewhere, and $\widetilde{k}_i \in [d_o]$ is*

$$\widetilde{k}_i = G(k_i) \in [d_o], \quad \textit{with} \quad k_i = \underset{k\in\{0,1,\cdots,p-1\}}{\mathrm{argmin}} (-2x_i^\top w_i - 2t_i + \widetilde{L}_0 + \widetilde{L}_k) \cdot k.$$

*Here $G : [p] \to [d_o]$ denotes any set-to-set function sending each selected interpolation index $k_i$ into an integer $\widetilde{k}_i \in [d_o]$ for $i \in [n]$. By setting $\beta \geq (\ln(n-1) - \ln \epsilon_0)/\delta$, we can make $\epsilon_0$ arbitrarily small, though the theorem fails on a arbitrarily small volumn in $\mathbb{R}^{d\times n}$. When $\widetilde{k}_i$ a constant for all $i \in [n]$, by setting $\beta \geq (\ln(n-2) - \ln \epsilon_0)/((\Delta L)^2/2)$ we achieve arbitrary small $\epsilon_0$ without any failure region.*

*Proof Sketch.* Applying our interpolation method in the in-context learning setting, we compute the interpolation point closest to the linear transformation result within attention. This involves comparing $|w^\top x - \widetilde{L}_k|_2^2$ for $k \in [p]$. Consequently, attention must compute a *quartic* polynomial of the input, while standard transformers only produce *quadratic* expressions. To combat this, we propose a technique enabling attention to perform equivalent computations for higher-order polynomials in our setting. Please see Remark G.24 and Appendix G.9 for a detailed proof. □

## D.2 IN-CONTEXT GRADIENT DESCENT

We extend Theorem D.1 to show that standard softmax attention perform in-context gradient descent, broadening the results established for ReLU attention in (Bai et al., 2024). Specifically, we demonstrate that softmax attention is capable of doing in-context gradient descent on convex loss functions.

We first define the problem setting similar to theirs.

**Definition D.2** (In-Context Learning Problem Formulation)**.** *The sequential input $X$ in the in-context learning scenario is defined as*

$$X := \begin{bmatrix} x_1 & x_2 & \cdots & x_n \\ y_1 & y_2 & \cdots & y_n \\ w & w & \cdots & w \\ 1 & 1 & \cdots & 1 \end{bmatrix},$$

*where $(x_i, y_i)$, $\quad i \in [n]$ denote the input-output pairs. $w$ parametrize the model connecting $x_i$ and $y_i$, and is altered(trained) between layers.*

**Remark D.3.** *The task of in-context learning is simplified to using the given input-output pairs $(x_i, y_i)$ to predict the output of a newcome input $x_u$.*

In this setting, we prove a multi-head Softmax attention is capable of doing in-context gradient descent on loss functions parametrized by $w^\top x_i$ $\quad(i \in [n])$ and $t$(as linear coefficient and bias), as well as giving an according prediction to the output on $x_u$.

**Theorem D.4** (In-Context Gradient Descent)**.** *Let $l : \mathbb{R} \times \mathbb{R} \to \mathbb{R}$ be any $C^1$ loss function defined on $(w^\top x_i, y_i)$. With input $X$ in the form of Definition 4.1, when $X$ is bounded, there exists a multi-head self-attention $\mathrm{Attn_m}$ with skip connections and each attached with a linear layer, such that for any $\epsilon > 0$, irrelevant of $X$, we have*

$$\left\| \mathrm{Attn}_m \circ A(X) - \begin{bmatrix} x_1 & \cdots & x_n \\ y_1 & \cdots & y_n \\ w - \eta \nabla L(w) & \cdots & w - \eta \nabla L(w) \\ 1 & \cdots & 1 \end{bmatrix} \right\|_\infty \leq \epsilon,$$

*where $\eta$ denotes the learning rate and $L(w) := \sum_{i=1}^n l(w^\top x_i, y_i)$ is an empirical loss upon the given input-output pairs.*

*Proof Sketch.* We know the universal approximation theorem for ReLU neural networks (Pinkus, 1999) ensures that there exists ReLU network $\sum_{h=1}^H \mathrm{ReLU}(a_h^{(r)} w^\top x_i + b_h^{(r)} y_i + c_h^{(r)})$ approximate $r$-th coordinate of the derivative of loss function on token $x_i$ as $\frac{\partial}{\partial_w} l(w^\top x_i, y_i) \in \mathbb{R}^d$ for $r \in [d]$. By Theorem D.1, we construct multi-head attention with linear mapping $\sum_{h=1}^H \sum_{r=1}^d \mathrm{Attn}_{h,r} \circ A_{h,r}(\cdot)$ to approximate the above ReLU neural network on every coordinate $r$, hence also approximate $\frac{\partial}{\partial_w} l(w^\top x_i, y_i) \in \mathbb{R}^d$ for $r \in [d]$. By designing $W_O^*$ we sum up the derivative of loss function $\frac{\partial}{\partial_w} l(w^\top x_i, y_i)$ on different in-context example $(x_i, y_i)$, hence approximate $\nabla L(w)$. Please see Appendix G.10 for a detailed proof. $\qquad\square$

We note that in the original proof of (Bai et al., 2024), they also rely on the approximation ability of ReLU neural networks to approximate the derivative of the loss function. Therefore, they use ReLU-based attention in their proof to approximate a sum of ReLU functions. In contrast, by leveraging Theorem 3.4, we show that softmax attention approximates generalized ReLU function by approximating truncated linear models, and hence approximates in-context gradient descent. Since softmax attention is the dominant mechanism used in practice, our results provide a more realistic foundation for understanding in-context learning tasks.

Beyond this, Theorem D.4 also suggest two advanced future works.

- **Task Composition.** Our construction naturally extends to task composition from subtasks. Suppose we have $N$ subtasks (e.g., gradient descent, lasso, linear regression, etc.). Theorem D.4 imply there exists an attention layer to approximate this task in-context (such as $\mathrm{Attn_{GD}}$ for the gradient-descent subtask). Our universal approximation results allow a *frozen* attention module to approximate these task-specific attention maps in-context, so that a single attention module realize all $N$ subtasks from input-output examples. We also remark that this step is not trivial. On top of this frozen layer, we introduce additional "routing" attention layer to select and compose these subtasks into the task of our interest in-context. This way extends our techniques to meta-learning or task composition naturally.

- **Simulation of Learning Algorithms In-Context.** By stacking Theorem D.4, softmax attention simulate multi-step in-context gradient descent and thereby recover a wide range of learning algorithms. Specifically, our Theorem D.4 shows that a single softmax-attention layer implement one step of gradient descent for any $C^1$ loss of the form $l(w^\top x_i, y_i)$. This contains a wide range of loss functions, including ridge, GLM and lasso loss functions. By stacking copies of this layer, standard convergence results for gradient descent (as in Lemma 14, Lemma C.1, Proposition A.2, and Proposition A.3 of (Bai et al., 2024)) imply that a depth-$T$ transformer approximate the $T$-step in-context learning dynamics for these algorithms. We view these two directions as promising next steps for developing a more systematic theory of task composition and algorithm learning in transformers.

# E  RELATED WORK

**Universal Approximation of Transformer.**  We first introduce the most relevant previous works about the universal approximation ability of transformer, and move on to other works investigating the expressive power of transformer with different target function classes.

Yun et al. (2019); Kajitsuka & Sato (2023) treat attention layer as the contextual mappings and derive that attention layer attached with FFNs is a universal approximator on continuous sequence-to-sequence permutation equivariant function. Specifically, Yun et al. (2019) prove that multi-head attention with two-layer FFN approximate continuous permutation equivariance sequence-to-sequence functions on a compact domain. Their construction of transformer block maintains constant width but requires $O(n(1/\delta)^{dn}/n!)$ layers, where $\delta$ is the fixed grid width of the input domain and $n$ is the sequence length. For any continuous sequence-to-sequence function, removing the factorial term $n!$ in the denominator leaves the remaining term growing exponentially with $n$. Kajitsuka & Sato (2023) further show that one-layer and single-head attention, with low-rank weight matrices, is able to carry out contextual mapping, simplifying the construction in terms of the number of layers. Takakura & Suzuki (2023) prove that one-layer transformer (attention + token-wise FFN) with one embedding layer (with positional encoding) approximates shift-equivariant $\alpha$-smoothness function. They show that the approximation error of the above function class is independent of input and output dimension hence achieving an infinite-dimension approximation result. Their result require $O\big(\log(1/\epsilon)^{1/\alpha}\big)$ number of heads. Jiang & Li (2023) use Kolmogorov representation theorem to get Jackson-type approximation rate, which hinges on explicit smoothness assumptions that yield quantitative convergence rates for single-layer single-head transformer. Despite these advances, prior works proving the universality of Transformers often depend on the FFNs attached after attention to perform token-wise transformations. Our work is different from these papers by removing the need for FFN from transformer to demonstrate the first universal approximation result of attention mechanism. Our construction requires 2-layer multi-head attention with linear transformation, with head complexity $H = O(d(1/\delta)^{dn})$. See Remark E.1 for detailed discussion.

Several other works investigate the universal approximation theorem of transformer with different variants. Yun et al. (2020) prove that universal approximation of sparse transformer. Kratsios et al. (2022) prove the constrained universal approximation theorem of probabilistic transformer. Likhosherstov et al. (2023) and Edelman et al. (2022) demonstrate that a single-layer self-attention mechanism is able to learn sparse functions of the input sequence, with sample complexity and hidden size are logarithmic relative to the sequence length. For the representation power on matrix, Bhojanapalli et al. (2020) show that when the hidden dimension of attention is smaller than sequence, multi-head attention cannot output certain positive column-stochastic matrices, and Likhosherstov et al. (2023) show that self-attention approximates any sparse matrices. Other works investigate the universal approximation of in-context learning setting. Furuya et al. (2024) shows a deep transformer block approximating any continuous mapping from an arbitrary-length prompt to its next token. Li et al. (2025) characterize a broad family of functions $f : \mathbb{R}^d \to \mathbb{R}$ that admit a sparse expansion in a fixed finite feature basis, and show that a transformer with sigmoid activation attention simulates a Lasso objective to recover those coefficients from the in-context examples. In contrast, our universal approximation result targets standard sequence-to-sequence functions outside the ICL framework and removes FFN, establishing universality for the softmax attention mechanism alone.

**Interpolation Methods for Universal Approximations.**  We also summarize and discuss prior works that utilize interpolation-based approaches to establish universal approximation theory:

- Kratsios (2023) build a "Probabilistic Transformer" that approximates regular conditional distributions by combining a feed-forward network with an attention-based final layer. They use softmax weights to form convex combinations of "anchor" distributions. Our work also relies on anchor selection via softmax, but we focus on deterministic sequence-to-sequence tasks rather than mapping to probability measures.

- Shen et al. (2022) prove that ReLU MLPs can achieve optimal approximation rates by partitioning the domain into dyadic grids. This is a classic piecewise-linear interpolation strategy. Our approach shares the same interpolation philosophy but implements it through attention (without requiring deep ReLU layers).

- Galimberti (2024) address inputs from non-metric or infinite-dimensional spaces by projecting them onto finite "anchors" and then applying MLP-like operators. This resembles our anchor-based selection, though we rely on standard self-attention rather than specialized infinite-dimensional layers.

- Fang et al. (2022) replace softmax with a hard argmax (infinite-temperature) attention to enable exact polynomial interpolation, achieving zero approximation error. In contrast, we keep continuous softmax but still realize the same anchor-selection principle for universal approximation.

- Kratsios & Furuya (2025) show that MLPs with trainable activations are universal in-context learners. They construct Voronoi partitions of the context space. Our work also employs anchor-driven interpolation for in-context tasks, but via attention-based selection rather than partitioning MLPs.

- Furuya et al. (2024) demonstrate that standard Transformers, with multi-head softmax attention and feed-forward layers, are universal in-context learners for arbitrary-size contexts. We focus on attention-alone universality. Our proofs show that even a single or two-layer softmax-attention mechanism can learn continuous sequence-to-sequence or in-context functions.

All these works use interpolation as a unifying theme, either via MLPs, attention, or trainable activations. Some rely on argmax (hard selection), others on softmax (continuous weighting). We extend this line by showing that a minimal attention-only setup suffices for universal approximation. Our method needs no deep stacks or feed-forward blocks, and it extends naturally to in-context learning.

**Remark E.1** (Head Complexity and Paramerters Complexity). *In Appendix G.6, we use $2dg^{dn} + 1$ heads to achieve sequence-to-scalar universal approximation. The first term is because we need $2d$ heads per grid point and there are $|G_D| = g^{dn}$ grid points, see (G.46). Also note that $p = |G_D|$ in those attentions. The extra one head in second term select $f(v)$, see (G.50) . **For sequence-to-sequence universal approximation**, the first layer is the same as multi-head attention that quantizes the input as in Appendix G.6, but increase the selection heads to $dn$. The total number of heads is therefore $2dg^{dn} + dn$. If we represent the head complexity with grid width $\delta_{\mathrm{grid}} \coloneqq 2B/g$ as in Yun et al. (2019), **the head complexity satisfies** $H = O(d(1/\delta_{\mathrm{grid}})^{dn})$. Next we derive the parameters complexity. In the multi-head construction (G.47) for grid-point approximation and in the single-head selection attention (G.50), each head has constant dimension, so the parameters of each head are $O(1)$. The linear maps $A_0$ and $A_*$ in the proof of Lemma G.14 (and $A_1$ in Lemma 3.12) are shared by all $2dg^{dn}$ heads and do not scale with the number of heads. The only component scale with the $2dg^{dn}$ head is $W_0$. That one contribute to $O(ng^{dn})$ parameters per head, hence $O(dng^{2dn})$ parameters in total. For $dn$ single-head attention $\mathrm{Attn}_s^{ij}$ in sequence-to-sequence approximation (G.57), each head has dimension $O(dn)$ and contributes $O((dn)^2)$ parameters. The map $A_2$ is shared by all $dn$ single-head $\mathrm{Attn}_s^{ij}$, so its parameter count does not scale with $dn$, but its dimension is $O(dng^{dn})$. Hence the total parameters complexity is $O(dng^{2dn} + (dn)^2) = O(dn(1/\delta_{\mathrm{grid}}^2)^{dn})$. For Theorem I.2, the sequence-to-squence approximation result based on Theorem 3.9, the head complexity is $H = O(N)$, where $N$ is the number of neuron of ReLU feed-forward network it aim to approximate. By classical FFN universal approximation results (Pinkus, 1999) we have $N = O((1/\delta_{\mathrm{grid}})^{dn})$, hence $H = O((1/\delta_{\mathrm{grid}})^{dn})$. The parameters complexity is dominated by the first $O(N(d^2 + dn))$ and the third layer $O(n^2N^2)$ of attention, and it's clear $O(n^2N^2) = O(n^2(1/\delta_{\mathrm{grid}}^2)^{dn})$ dominate since $N \gg d$. Finally, as noted also in Yun et al. (2019), this exponential dependence cannot be avoided when approximating arbitrary continuous functions, since in the worst case the model must memorize an independent output for each of the $g^{dn}$ grid.*

**Remark E.2.** *One may reduce the parameter complexity above by:*

- *Replacing or improving the sequence-wise linear transformation, for example by using an additional attention layer to replace the sequence-wise layer.*

- *Only calculating the learnable parameters (e.g., $w, t$) attribute to universal approximation ability. There is large amount of entries in the constructed matrices are fixed as $0$ and $1$. One can also optimize the constructive proof to optimize the use of those entries.*

*We leave this direction to future work.*

# F    ADDITIONAL THEORETICAL RESULTS

## F.1    APPROXIMATING HARDMAX WITH FINITE TEMPERATURE SOFTMAX

A central step in our proofs is to replace a hard $\arg\max$ operation with a continuous softmax using a sufficiently large inverse temperature $\beta$. Intuitively, as $\beta \to \infty$, the softmax output approaches a one-hot vector that selects the largest entry of $x$. The following lemma provides a precise bound on how large $\beta$ must be to achieve a desired approximation error.

**Lemma F.1** (Approximating Hardmax with Finite-Temperature Softmax). *Let* $x = [x_1, x_2, \ldots, x_n] \in \mathbb{R}^n$, $\epsilon > 0$. *Define* $\mathsf{Softmax}_\beta(\cdot)$ *as*

$$\mathsf{Softmax}_\beta(x) := \left[ \frac{\exp(\beta x_1)}{\sum_{j=1}^n \exp(\beta x_j)}, \cdots, \frac{\exp(\beta x_n)}{\sum_{j=1}^n \exp(\beta x_j)} \right].$$

*The following statements hold:*

- ***Case of a Unique Largest Entry.*** *Assume* $x_1 = \max_{i \in [n]} x_i$ *is unique, and* $x_2 = \max_{i \in [n] \setminus \{1\}} x_i$. *Then, if* $\beta \geq (\ln(n-1) - \ln(\epsilon))/(x_1 - x_2)$, *we have*

$$\left\| \mathsf{Softmax}_\beta(x) - e_1 \right\|_\infty \leq \epsilon,$$

  *where* $e_1 \in \mathbb{R}^n$ *is the one-hot vector corresponding to to the maximal entry of* $x$ *(i.e.,* $x_1$.*)*

- ***Case of Two Largest Entries (Tied or Separated by $\delta$).*** *Assume* $x_1$ *and* $x_2$ *are the first and second largest entries, respectively, with* $\delta = x_1 - x_2 \geq 0$. *Let* $x_3$ *be the third largest entry and is smaller than* $x_1$ *by a constant* $\gamma > 0$ *irrelevant to the input. Then, if* $\beta \geq (\ln(n-2) - \ln \epsilon)/\gamma$, *we have*

$$\left\| \mathsf{Softmax}_\beta(x) - \frac{1}{1 + e^{-\beta\delta}} e_1 - \frac{e^{-\beta\delta}}{1 + e^{-\beta\delta}} e_2 \right\|_\infty \leq \epsilon.$$

*Proof.* Please see Appendix G.1 for a detailed proof. $\qquad\qquad\qquad\qquad\qquad\qquad\qquad\square$

## F.2    SEQUENCE-TO-SEQUENCE UNIVERSAL APPROXIMATION WITH $\infty$-NORM ERROR

Here, we present the result that a two-layer multi-head attention mechanism achieves sequence-to-sequence universal approximation with respect to the $\infty$-norm error.

We refine our sequence-to-sequence approximation result from Theorem 3.14 to an $\infty$-norm guarantee in Theorem F.2. We achieve this by combining the existing ReLU neural networks approximation result in $\infty$-norm with our attention-approximate-generalized-ReLU result from Theorem 3.4.

**Theorem F.2** (Sequence-to-Sequence Approximation in Infinity Norm). *For any continuous function* $f : \mathbb{R}^{d \times n} \to \mathbb{R}^{d \times n}$ *of compact support* $\mathcal{X}$, *and any* $\epsilon > 0$, *we prove that when attached with linear transformations, there exists a one layer multi-head attention* $\mathrm{Attn}_m$ *stacked with one layer multi-head attention* $\mathrm{Attn}_m$, *such that when the precision parameter in Theorem G.22 is* $p = \Omega(n^{5/2})$, *for any* $X \in \mathcal{X}$

$$\|f(X) - \mathrm{Attn}_m^{(2)} \circ A \circ \mathrm{Attn}_m^{(1)} \circ A(X)\|_\infty \leq \epsilon.$$

*Proof.* Please see Appendix G.8 for a detailed proof. $\qquad\qquad\qquad\qquad\qquad\qquad\qquad\square$

# G PROOFS OF MAIN TEXT

## G.1 PROOF OF LEMMA F.1

**Lemma G.1** (Lemma F.1 Restated: Approximating Hardmax with Finite-Temperature Softmax)**.**
*Let $x = [x_1, x_2, \ldots, x_n] \in \mathbb{R}^n$, $\epsilon > 0$. Define $\mathsf{Softmax}_\beta(\cdot)$ as*

$$\mathsf{Softmax}_\beta(x) := [\frac{\exp(\beta x_1)}{\sum_{j=1}^n \exp(\beta x_j)}, \cdots, \frac{\exp(\beta x_n)}{\sum_{j=1}^n \exp(\beta x_j)}].$$

*The following statements hold:*

- ***Case of a unique largest entry.*** *Assume $x_1 = \max_{i \in [n]} x_i$ is unique, and $x_2 = \max_{i \in [n] \setminus \{1\}} x_i$. Then, if $\beta \geq (\ln(n-1) - \ln(\epsilon))/(x_1 - x_2)$, we have*

$$\left\| \mathsf{Softmax}_\beta(x) - e_1 \right\|_\infty \leq \epsilon,$$

  *where $e_1 \in \mathbb{R}^n$ is the one-hot vector corresponding to to the maximal entry of $x$ (i.e., $x_1$.)*

- ***Case of two largest entries (tied or separated by $\delta$).*** *Assume $x_1$ and $x_2$ are the first and second largest entries, respectively, with $\delta = x_1 - x_2 \geq 0$. Let $x_3$ be the third largest entry and is smaller than $x_1$ by a constant $\gamma > 0$ irrelevant to the input. Then, if $\beta \geq (\ln(n-2) - \ln(\epsilon))/\gamma$, we have*

$$\left\| \mathsf{Softmax}_\beta(x) - \frac{1}{1 + e^{-\beta\delta}} e_1 - \frac{e^{-\beta\delta}}{1 + e^{-\beta\delta}} e_2 \right\|_\infty \leq \epsilon.$$

*Proof.* In the following proof, we denote $\mathsf{Softmax}(\cdot)$ function as $\sigma(\cdot)$ for simplicity.

For the first condition that $x$ with unique maximal entry $x_1$, denote $\exp(\beta x_i)/\sum_{j=1}^n \exp(\beta x_j)$ as $\sigma_\beta(x)_i$. we have:

$$
\begin{aligned}
& \|\sigma_\beta(x_1, x_2, x_3, \cdots, x_n)_1 - e_1\|_\infty \\
&= \max\{1 - \sigma(x)_1, \sigma(x)_2, \cdots, \sigma(x)_n\} \\
&= \max\{1 - \sigma(x)_1, 1 - \sigma(x)_1 - \sum_{i \neq 1,2} \sigma(x)_i, \cdots, 1 - \sigma(x)_1 - \sum_{i \neq 1,n} \sigma(x)_i\} \quad \text{(By } \sum_{i=1}^n \sigma(x)_i = 1) \\
&\leq 1 - \sigma(x)_1 \\
&= 1 - \frac{1}{1 + \sum_{j=2}^n e^{\beta(x_j - x_1)}} \quad \text{(By dividing } \sigma(x)_1 \text{ by } e^{\beta x_1}) \\
&= \frac{\sum_{j=2}^n e^{\beta(x_j - x_1)}}{1 + \sum_{j=2}^n e^{\beta(x_j - x_1)}} \\
&\leq \sum_{j=2}^n e^{\beta(x_j - x_1)} \\
&\leq (n-1) e^{\beta(x_2 - x_1)} \quad \text{(Since } x_2 \text{ is the second largest entry)} \\
&\leq \epsilon.
\end{aligned}
$$

For the second occasion, we have:

$$
\begin{aligned}
& \left\| \sigma_\beta(x) - \frac{1}{1 + e^{-\beta\delta}} e_1 - \frac{e^{-\beta\delta}}{1 + e^{-\beta\delta}} e_2 \right\|_\infty \\
& \leq \max\{ \frac{1}{1 + e^{-\beta\delta}} - \sigma_\beta(x)_1, \frac{e^{-\beta\delta}}{1 + e^{-\beta\delta}} - \sigma_\beta(x)_2, \sigma_\beta(x)_3, \cdots, \sigma_\beta(x)_n \}, \qquad \text{(G.1)}
\end{aligned}
$$

where the last inequality comes from the definition of infinity norm. Plug in $\delta = x_1 - x_2$ we calculate the first two term to be:

$$
\begin{aligned}
\frac{1}{1 + e^{-\beta\delta}} - \sigma_\beta(x)_1 &= \frac{1}{1 + e^{\beta(x_2 - x_1)}} - \frac{e^{\beta x_1}}{\sum_{i=1}^n e^{\beta x_i}} \\
&= e^{\beta x_1} \left( \frac{1}{e^{\beta x_1} + e^{\beta x_2}} - \frac{1}{\sum_{i=1}^n e^{\beta x_i}} \right) \qquad (\tfrac{1}{1 + e^{\beta(x_2 - x_1)}} = \tfrac{e^{\beta x_1}}{e^{\beta x_1} + e^{\beta x_2}}) \\
&= e^{\beta x_1} \left( \frac{\sum_{i=1}^n e^{\beta x_i} - (e^{\beta x_1} + e^{\beta x_2})}{(e^{\beta x_1} + e^{\beta x_2})(\sum_{i=1}^n e^{\beta x_i})} \right) = \frac{e^{\beta x_1}(\sum_{i=3}^n e^{\beta x_i})}{(e^{\beta x_1} + e^{\beta x_2})(\sum_{i=1}^n e^{\beta x_i})}.
\end{aligned}
$$

Follows the same calculation we get

$$
\frac{e^{-\beta\delta}}{1 + e^{-\beta\delta}} - \sigma_\beta(x)_2 = \frac{e^{\beta x_2}(\sum_{i=3}^n e^{\beta x_i})}{(e^{\beta x_1} + e^{\beta x_2})(\sum_{i=1}^n e^{\beta x_i})}.
$$

Hence we have

$$
\begin{aligned}
&\max\left\{ \frac{e^{\beta x_1}(\sum_{i=3}^n e^{\beta x_i})}{(e^{\beta x_1} + e^{\beta x_2})(\sum_{i=1}^n e^{\beta x_i})}, \frac{e^{\beta x_2}(\sum_{i=3}^n e^{\beta x_i})}{(e^{\beta x_1} + e^{\beta x_2})(\sum_{i=1}^n e^{\beta x_i})} \right\} \\
&\leq \frac{e^{\beta x_1}(\sum_{i=3}^n e^{\beta x_i})}{(e^{\beta x_1} + e^{\beta x_2})(\sum_{i=1}^n e^{\beta x_i})} \qquad (x_1 \geq x_2 \text{ by assumption}) \\
&\leq \sum_{i=3}^n e^{\beta(x_i - x_1)} \\
&\leq (n - 2) \cdot e^{\beta(x_3 - x_1)} \\
&\leq \epsilon. \qquad\qquad\qquad\qquad\qquad\qquad\qquad\qquad\qquad\qquad\qquad\qquad\qquad\text{(G.2)}
\end{aligned}
$$

Furthermore we have

$$
\begin{aligned}
|\sigma_\beta(x)_i| &\leq |\sigma_\beta(x)_3| \qquad\qquad (\text{By assumption } x_3 \text{ is the third largest elements}) \\
&= \frac{e^{\beta x_3}}{\sum_{j=1}^n e^{\beta x_j}} \\
&\leq \frac{e^{\beta x_3}}{e^{\beta x_1}} \\
&= e^{\beta(x_3 - x_1)} \\
&\leq e^{\frac{\ln(n-2) - \ln(\epsilon)}{x_1 - x_3}(x_3 - x_1)} \qquad (\text{By the assumption of } \beta \text{ in the main text}) \\
&\leq \frac{\epsilon}{n - 2}. \qquad\qquad\qquad\qquad\qquad\qquad\qquad\qquad\qquad\qquad\qquad\text{(G.3)}
\end{aligned}
$$

Combining (G.2) and (G.3) yields that (G.1) is

$$
\begin{aligned}
&\left\| \sigma_\beta(x) - \frac{1}{1 + \exp^{-\beta\delta}} e_1 - \frac{\exp^{-\beta\delta}}{1 + \exp^{-\beta\delta}} e_2 \right\|_\infty \\
&= \max\left\{ \sigma_\beta(x)_1 - \frac{1}{1 + \exp^{-\beta\delta}}, \frac{\exp^{-\beta\delta}}{1 + \exp^{-\beta\delta}} - \sigma_\beta(x)_2, \sigma_\beta(x)_3 \right\} \\
&\leq \max\left\{ \frac{e^{\beta x_1}(\sum_{i=3}^n e^{\beta x_i})}{(e^{\beta x_1} + e^{\beta x_2})(\sum_{i=1}^n e^{\beta x_i})}, \frac{e^{\beta x_2}(\sum_{i=3}^n e^{\beta x_i})}{(e^{\beta x_1} + e^{\beta x_2})(\sum_{i=1}^n e^{\beta x_i})}, \frac{e^{\beta x_3}}{\sum_{j=1}^n e^{\beta x_j}} \right\} \\
&\leq \max\left\{ \epsilon, \frac{\epsilon}{n - 2} \right\} \qquad\qquad\qquad\qquad\qquad (\text{By (G.2) and (G.3)}) \\
&\leq \epsilon.
\end{aligned}
$$

This completes the proof. □

## G.2 PROOF OF THEOREM 3.4

We first define $\delta$ used in this theorem. For $i$-th column of attention score matrix $K^\top Q$, let $x_{1,i}$ and $x_{2,i}$ be its largest and second-largest entries and define $\delta_i := x_{1,i} - x_{2,i}$, and denote $\delta = \min_{i \in [n]} \delta_i$ to be the smallest such gap over all columns.

**Theorem G.2** (Theorem 3.4 Restated: Single-Head Attention Approximates Many Truncated Linear Models). *Fix real $a < b$, and let $\mathrm{Range}_{[a,b]}(\cdot)$ be the truncation operator from Definition 3.1. Let $\epsilon_0 \geq 0$. For a precision parameter $p > n$, there exists a single-layer, single-head self-attention $\mathrm{Attn}$ with a linear transformation $A : \mathbb{R}^{d \times n} \to \mathbb{R}^{(2d+d_o+2) \times p}$, such that $\mathrm{Attn} \circ A : \mathbb{R}^{d \times n} \to \mathbb{R}^{d_o \times n}$ satisfies, for any $i \in [n]$,*

$$
\|\mathrm{Attn} \circ A(X)_{:,i} - \mathrm{Range}_{[a,b]}(w_i^\top x_i + t_i) e_{\widetilde{k}_i}\|_\infty \leq \underbrace{\max\{|a|, |b|\} \cdot \epsilon_0}_{\text{finite-}\beta \text{ softmax error}} + \underbrace{\frac{b-a}{p}}_{\text{interpolation error}} \quad .
$$

*Here $e_{\widetilde{k}_i}$ is a one-hot vector with a value of $1$ at the $\widetilde{k}_i$-th index and $0$ elsewhere, and*

$$
k_i := \operatorname*{argmin}_{k \in \{0,1,2,\cdots,p-1\}} |x_i^\top w + t - \widetilde{L}_k| \quad \text{where} \quad \widetilde{k}_i := G(k_i) \in [d_o].
$$

*Here $k_i \in \{0, ..., p-1\}$ is the index of the interpolation point closest to the $i$-th token ($i$-th truncated linear model). For all $i \in [n]$, $G : \{0, ..., p-1\} \to [d_o]$ denotes any set-to-set function sending the interpolation index $k \in \{0, ..., p-1\}$ into a position index $\widetilde{k} \in [d_o]$ specifying in the desired row index of the output. By setting $\beta \geq (\ln(n-1) - \ln \epsilon_0)/\delta$, we make $\epsilon_0$ arbitrarily small, though the theorem fails on a arbitrarily small volume in $\mathbb{R}^{d \times n}$. When $\widetilde{k}_i$ a constant for all $i \in [n]$, by setting $\beta \geq (\ln(n-2) - \ln \epsilon_0)/((\Delta L)^2/2)$, we achieve arbitrary small $\epsilon_0$ without any failure region.*

*Proof.* We provides two version of proofs:

- **Proof of Case (i).** The largest entry in $K^\top Q$ is unique. In case (i), $\beta$ scale with $O(1/\delta)$. These have two drawbacks: (i) $\beta$ depends on the input instead of the model architecture and (ii) to make the error $\epsilon_0$ arbitrarily small and when $\delta$ is close to zero, one needs very large $\beta$, and even then the guarantee excludes an arbitrarily small volume in $\mathbb{R}^{d \times n}$.

- **Proof of Case (ii).** The top two entries are either tied or separated by a small gap $\delta \geq 0$. By contrast, later in the proof we show that when applying case (ii) of Lemma F.1, $\beta$ scale with $O(1/\gamma)$ and $\gamma = O((\Delta L)^2)$, a constant for fixed model and irrelevant to the input. This better align with practices, so the theory statement emphasizes case (ii) in the main text.

We begin with the common setup used by both cases.

First we denote $\ell_k := k\widetilde{L}_k + k\widetilde{L}_0 - 2kt$ and $\widetilde{L}_k$ for $k = 0, \ldots, p-1$ following Definition 3.3.

Then, we specify the linear transformation $A$ prepended to attention layer $\mathrm{Attn}$

$$
A(X) = \underbrace{\begin{bmatrix} I_d \\ 0_{(d+d_o+2) \times d} \end{bmatrix}}_{(2d+d_0+2) \times d} X \underbrace{[I_n, 0_{n \times (p-n)}]}_{n \times p} + \underbrace{\begin{bmatrix} 0_d & 0_d & \cdots & 0_d & 0_d & \cdots & 0_d \\ 0_d & w & \cdots & (n-1)w & nw & \cdots & (p-1)w \\ 0 & \ell_1 & \cdots & \ell_{n-1} & \ell_n & \cdots & \ell_{p-1} \\ & & & \widetilde{L}_{d_o \times p} & & & \\ 1 & 1 & \cdots & 1 & 0 & \cdots & 0 \end{bmatrix}}_{(2d+d_o+2) \times p}
$$

$$
= \begin{bmatrix} x_1 & x_2 & \cdots & x_n & 0 & \cdots & 0 \\ 0_d & w & \cdots & (n-1)w & nw & \cdots & (p-1)w \\ 0 & \ell_1 & \cdots & \ell_{n-1} & \ell_n & \cdots & \ell_{p-1} \\ & & & \widetilde{L}_{d_o \times p} & & & \\ 1 & 1 & \cdots & 1 & 0 & \cdots & 0 \end{bmatrix} \in \mathbb{R}^{(2d+d_o+2) \times p}, \tag{G.4}
$$

where $\widetilde{L} = [\widetilde{L}_0 e_{\widetilde{0}}, \cdots, \widetilde{L}_j e_{\widetilde{j}}, \widetilde{L}_{p-1} e_{\widetilde{p-1}}] \in \mathbb{R}^{d_0 \times p}$. Here, $e_{\widetilde{j}} \in \mathbb{R}^{d_0}$ denotes a one-hot vector where only the $j$-th index has a value of 1.

Namely, before feeding the input token into the self-attention mechanism $\mathrm{Attn}$, we preprocess it with linear transformations $A : \mathbb{R}^{d \times n} \to \mathbb{R}^{(2d+d_0+2) \times p}$. Note that the precision parameter $p \in \mathbb{N}$, defined in Definition 3.3, is required to be larger than the input sequence length $n$.

Essentially, $A$ extends the input sequence with extra rows/columns for the latter use of interpolation approximation.

> **Remark G.3.** *The A here is a sequence-wise linear transformation for the simplicity of demonstrating our method. For a practical, token-wise implementation, see Theorem 3.9. As noted at Appendix B, one can interchange Theorem 3.4 and Theorem 3.9 in all subsequent proofs since both yield the same approximation result. We also note that eliminating the sequence-wise operator $\begin{bmatrix} I_n, 0_{n \times (p-n)} \end{bmatrix}_{n \times p}$ in linear transformation A (G.4) is doable. We achieve this by simply padding input sequence $X \in \mathbb{R}^{d \times n}$ to have sequence length $p$.*

For the attention matrices of the self-attention layer, we construct their parameters to be

$$
W_Q = -\beta \begin{bmatrix} I_d & 0_{d \times d} & 0_{d \times 1} & 0_{d \times d_0} & 0_{d \times 1} \\ 0_{1 \times d} & 0_{1 \times d} & 0 & 0_{1 \times d_0} & 1 \end{bmatrix} \in \mathbb{R}^{(d+1) \times (2d+d_0+2)},
$$

$$
W_K = \beta \begin{bmatrix} 0_{d \times d} & -2I_d & 0_d & 0_{d \times d_0} & 0_{d \times 1} \\ 0_{1 \times d} & 0_{1 \times d} & 1 & 0_{1 \times d_0} & 0 \end{bmatrix} \in \mathbb{R}^{(d+1) \times (2d+d_0+2)}.
$$

In this setting, we construct the query and key matrix $Q$, $K$ as

$$
Q = W_Q A(X) = -\beta \begin{bmatrix} x_1 & x_2 & \cdots & x_n & 0 & \cdots & 0 \\ 1 & 1 & \cdots & 1 & 0 & \cdots & 0 \end{bmatrix} \in \mathbb{R}^{(d+1) \times p},
$$

and

$$
K = W_K A(X)
$$
$$
= \beta \begin{bmatrix} 0 & -2w & \cdots & -2(p-1)w \\ 0 & \widetilde{L}_0 + \widetilde{L}_1 - 2t & \cdots & (p-1)\widetilde{L}_{p-1} + (p-1)\widetilde{L}_0 - 2(p-1)t \end{bmatrix} \in \mathbb{R}^{(d+1) \times p}.
$$

Thus for $K^\top Q$, we have

$$
K^\top Q
$$
$$
= -\beta^2 \underbrace{\begin{bmatrix} (-2x_1^\top w - 2t + \widetilde{L}_0 + \widetilde{L}_0) \cdot 0 & \cdots & (-2x_n^\top w - 2t + \widetilde{L}_0 + \widetilde{L}_0) \cdot 0 & 0 & \cdots \\ (-2x_1^\top w - 2t + \widetilde{L}_0 + \widetilde{L}_1) \cdot 1 & \cdots & (-2x_n^\top w - 2t + \widetilde{L}_0 + \widetilde{L}_1) \cdot 1 & 0 & \cdots \\ \vdots & \ddots & \vdots & \vdots & \ddots \\ (-2x_1^\top w - 2t + \widetilde{L}_0 + \widetilde{L}_p) \cdot (p-1) & \cdots & (-2x_n^\top w - 2t + \widetilde{L}_0 + \widetilde{L}_p) \cdot (p-1) & 0 & \cdots \end{bmatrix}}_{p \times p}.
$$

$$
\tag{G.5}
$$

Next, we use Lemma F.1 that softmax approximate hardmax to find the smallest entry in each column of $K^\top Q$. Then we find the closest interpolation point $\widetilde{L}_i$ to $w^\top x_i + t$.

**Proof of Case (i).** We consider the case when the largest entry in every column of $K^\top Q$ is unique and larger than the second largest entry by at least $\delta$. Using case (i) have no constraint on $\widetilde{k}_i$ but with the tradeoff that $\beta$ scale with $O(1/\delta)$ depending on the input, see Appendix G.1 for the detailed discussion.

By Lemma F.1, for arbitrary $\epsilon_0 > 0$, when every column has a unique minimum entry $u_1$ that's larger then the second largest $u_2$ for a constant at least $\delta$, and $\beta$ to be sufficiently large such that

$$\beta \geq \frac{\ln(n-1) - \ln \epsilon_0}{u_1 - u_2},$$

the following holds

$$\|\mathsf{Softmax}_\beta((K^\top Q)_{:,i}) - e_{k_i}\|_\infty \leq \epsilon_0, \tag{G.6}$$

where $k_i$ is defined as

$$k_i := \operatorname*{argmin}_{k \in \{0,1,\cdots,p-1\}} (-2x_i^\top w - 2t + \widetilde{L}_0 + \widetilde{L}_k) \cdot k.$$

The meaning of $k_i$ correspond to the interpolation point index $k$ that minimizes $|x_i^\top w + t - \widetilde{L}_k|$ for $k \in \{0, 1, \cdots, p-1\}$.

We further deduce this result as follows

$$\begin{aligned}
k_i &= \operatorname*{argmin}_{k \in \{0,1,2,\cdots,p-1\}} (-2x_i^\top w - 2t + \widetilde{L}_0 + \widetilde{L}_k) \cdot k \\
&= \operatorname*{argmin}_{k \in \{0,1,2,\cdots,p-1\}} (-2x_i^\top w - 2t + \widetilde{L}_0 + \widetilde{L}_k) \cdot k\Delta L \qquad \text{\small (Multiply a positive constant change nothing)} \\
&= \operatorname*{argmin}_{k \in \{0,1,2,\cdots,p-1\}} (-2x_i^\top w - 2t + \widetilde{L}_0 + \widetilde{L}_k) \cdot (\widetilde{L}_k - \widetilde{L}_0) \qquad \text{\small (By $k\Delta L = \widetilde{L}_k - \widetilde{L}_0$)} \\
&= \operatorname*{argmin}_{k \in \{0,1,2,\cdots,p-1\}} (-2x_i^\top w - 2t) \cdot (\widetilde{L}_k - \widetilde{L}_0) - (\widetilde{L}_0)^2 + (\widetilde{L}_k)^2 \qquad \text{\small (By distributive law)} \\
&= \operatorname*{argmin}_{k \in \{0,1,2,\cdots,p-1\}} (-2x_i^\top w - 2t) \cdot \widetilde{L}_k + (\widetilde{L}_k)^2 + (x_i^\top w + t)^2
\end{aligned}$$

$$\text{\small (here relative to the } \operatorname{argmax} (x_i^\top w + t) \text{ and } \widetilde{L}_0 \text{ are constant)}$$

$$\begin{aligned}
&= \operatorname*{argmin}_{k \in \{0,1,2,\cdots,p-1\}} (x_i^\top w + t - \widetilde{L}_k)^2 \tag{G.7} \\
&= \operatorname*{argmin}_{k \in \{0,1,2,\cdots,p-1\}} |x_i^\top w + t - \widetilde{L}_k|. \tag{G.8}
\end{aligned}$$

Until now we find the right interpolation point index $k$ that minimizes $|x_i^\top w + t - \widetilde{L}_k|$ for $k \in \{0, 1, \cdots, p-1\}$.

Next, we construct value matrix $V$ to map out the desired interpolation point $\widetilde{L}_{k_i}$ according to $\widetilde{k}_i$.

Define $W_V$ to pick up the matrix $\widetilde{L} = [\widetilde{L}_0 e_{\widetilde{0}}, \cdots, \widetilde{L}_j e_{\widetilde{j}}, \widetilde{L}_{p-1} e_{\widetilde{p-1}}] \in \mathbb{R}^{d_0 \times p}$

$$W_V = \begin{bmatrix} 0_{d_0 \times (2d+1)} & I_{d_0} & 0_{d_0 \times 1} \end{bmatrix} \in \mathbb{R}^{d_0 \times (2d+d_0+2)}.$$

This yields

$$V = W_V A(X) = \widetilde{L} = [\widetilde{L}_0 e_{\widetilde{0}}, \cdots, \widetilde{L}_j e_{\widetilde{j}}, \widetilde{L}_{p-1} e_{\widetilde{p-1}}] \in \mathbb{R}^{d_0 \times p} \tag{G.9}$$

Lastly, we use the linear transform $W_O$ to remove the unwanted columns in (G.5)

$$W_O = \begin{bmatrix} I_n \\ 0_{(p-n)\times n} \end{bmatrix} \in \mathbb{R}^{p\times n}.$$

Until now, we finish the construction of our attention layer

$$\text{Attn} \circ A(X) = \underbrace{V}_{d_o\times p}\underbrace{\text{Softmax}((W_K A(X))^\top W_Q A(X))}_{p\times p}\underbrace{W_O}_{p\times n} \in \mathbb{R}^{d_o\times n}.$$

Next, we derive the approximation error

$$\|\text{Attn} \circ A(X) - \underbrace{[\text{Range}_{[a,b]}(w_1^\top x_1 + t_1)e_{\widetilde{k}_1}, \cdots, \text{Range}_{[a,b]}(w_n^\top x_n + t_n)e_{\widetilde{k}_n}]}_{d_0\times n}\|_\infty < \epsilon.$$

Combining the column-wise results from (G.6) together with $V$ and $W_O$ matrices, we derive that for any $\epsilon_0 > 0$, if

$$\beta \geq \frac{\ln(n-1) - \ln\left(\frac{1}{2}\max\{|a|,|b|\}\epsilon_0\right)}{u_1 - u_2}, \tag{G.10}$$

where $u_1$ and $u_2$ are the largest and second-largest entries in each column of $K^\top Q$, the following holds

$$\|V\text{Softmax}(K^\top Q)W_O - V[e_{k_1}, e_{k_2}, \cdots, e_{k_n}]\|_\infty \tag{G.11}$$

$$=\|V\text{Softmax}(K^\top Q)W_O - [Ve_{k_1}, Ve_{k_2}, \cdots, Ve_{k_n}]\|_\infty \tag{G.12}$$

$$=\|V\text{Softmax}(K^\top Q)W_O - [\widetilde{L}_{k_1}e_{\widetilde{k}_1}, \widetilde{L}_{k_2}e_{\widetilde{k}_2}, \cdots, \widetilde{L}_{k_n}e_{\widetilde{k}_n}].\|_\infty$$
$$\text{\scriptsize (By (G.9) and that } V \text{ \scriptsize multiplied by one-hot vector } e_{k_i} \text{ \scriptsize returns its } k_i\text{\scriptsize -th column } V_{:,k_i}.)$$

$$< \max\{|a|,|b|\} \cdot \epsilon_0. \tag{G.13}$$

The softmax error in (G.6) is at most $\epsilon_0$ in infinity norm, but here scale by $V$ since $|\widetilde{L}_k|$ in $V$ is at most $max\{|a|,|b|\}$.

Note that (G.10) implies $\beta$ scale with $\delta \leq u_1 - u_2$ with $O(1/\delta)$. To avoid this input dependence, we now turn to case (ii) of Lemma F.1.

**Proof of Case (ii).** By Lemma F.1 case (ii), there are two top entries in $K^\top Q$, either tied or separated by a small gap $\delta \geq 0$. In Lemma F.1 we see that this case give better scaling for $\beta$ since it doesn't depend on $\delta$. However, $\widetilde{k}_i$ for all $i \in [n]$ should only be a constant as we state in the theory statement. This make sure when value matrix times the softmax matrix it compute the correct averaged between two interpolation points.

Let $\widetilde{k}_i$ be identical for all $i \in [n]$. According to Lemma F.1, the third largest $-(1/2\cdot(x_i^\top w+t-\widetilde{L}_k)^2)$ for all $k \in [n]$ is at least smaller than the largest $-(1/2 \cdot (x_i^\top w + t - \widetilde{L}_{k_i})^2)$ by (without a loss of generality, assume $x_i^\top w + t - \widetilde{L}_{k_i} > 0$)

$$-(\frac{1}{2}(x_i^\top w + t - \widetilde{L}_{k_i})^2) - [-(\frac{1}{2}(x_i^\top w + t - \widetilde{L}_{k_i} + \Delta L)^2)]$$

$$= \frac{1}{2}\Delta L[\Delta L + 2(x_i^\top w + t - \widetilde{L}_{k_i})]$$

$$\geq \frac{(\Delta L)^2}{2}. \qquad\qquad \text{\scriptsize ($(\Delta L)^2/2$ corresponds to the $\gamma$ in Lemma F.1)}$$

Therefore, we have

$$\left\|\mathsf{Softmax}_\beta(x) - \frac{1}{1 + e^{-\beta\delta}} e_{k_i} - \frac{e^{-\beta\delta}}{1 + e^{-\beta\delta}} e_{k_i'}\right\|_\infty \leq \frac{\epsilon_0}{2},$$

where $k_i'$ is the second largest entry.

Because

$$\begin{aligned}
\|V e_{k_i} - V e_{k_i'}\|_\infty &= \|L_{k_i} e_{\widetilde{k}_i} - L_{k_i'} e_{\widetilde{k}_i'}\|_\infty \\
&= \|L_{k_i} - L_{k_i'}\|_\infty &&\text{(By } e_{\widetilde{k}_i} = e_{\widetilde{k}_i'}) \\
&\leq \Delta L.
\end{aligned}$$

Thus for any $\epsilon_m > 0$ when $\Delta L \leq \epsilon_m$, we have

$$\begin{aligned}
&\left\|V\mathsf{Softmax}_\beta(x) - V e_{k_i}\right\|_\infty \\
&\leq \left\|V\mathsf{Softmax}_\beta(x) - V\frac{1}{1 + e^{-\beta\delta}} e_{k_i} - V\frac{e^{-\beta\delta}}{1 + e^{-\beta\delta}} e_{k_i'}\right\|_\infty + \epsilon_m \\
&\leq \frac{\epsilon_0}{2} + \epsilon_m.
\end{aligned}$$

Setting $\epsilon_m \leq \max(|a|, |b|)\epsilon_0/2$ yields (G.11).

We also remark that (G.11) is equivalent to

$$\|V\mathsf{Softmax}(K^\top Q)W_O - \begin{bmatrix} \widetilde{L}_{k_1} e_{\widetilde{k}_1} & \widetilde{L}_{k_2} e_{\widetilde{k}_2} & \cdots & \widetilde{L}_{k_n} e_{\widetilde{k}_n} \end{bmatrix}_{d_o \times n}\| \leq \max\{|a|, |b|\} \cdot \epsilon_0. \quad \text{(G.14)}$$

Until now, we finish the two-cases discussion of Lemma F.1. We now move to derive the interpolation error.

Lastly by the definition of $k_i$ and $\widetilde{L}_{k_i}$ we have:

$$|\widetilde{L}_{k_i} - \mathrm{Range}_{[a,b]}(w^\top x_i + t)| \leq \frac{b - a}{p}.$$

Thus

$$\begin{aligned}
&\|\mathrm{Attn}(X)_{:,i} - \mathrm{Range}_{[a,b]}(w^\top x_i + t) \cdot e_{\widetilde{k}_i}\|_\infty \\
&\leq \|\mathrm{Attn}(X)_{:,i} - \widetilde{L}_{\widetilde{k}_i} \cdot e_{\widetilde{k}_i}\|_\infty + \|\widetilde{L}_{\widetilde{k}_i} \cdot e_{\widetilde{k}_i} - \mathrm{Range}_{[a,b]}(w^\top x_i + t) \cdot e_{\widetilde{k}_i}\|_\infty &&\text{(By triangle inequality)} \\
&\leq \underbrace{\max\{|a|, |b|\} \cdot \epsilon_0}_{\text{finite-}\beta\text{ softmax error}} + \underbrace{\frac{b - a}{p}}_{\text{interpolation error}}, \quad \text{for} \quad i \in [n].
\end{aligned}$$

When $p$ goes to infinity and $\epsilon_0$ goes to 0, the total error is arbitrary small. Thus, we set

$$\max\{|a|, |b|\} \cdot \epsilon_0 + \frac{b - a}{p} \leq \epsilon.$$

This yields

$$\|\mathrm{Attn}(X) - [\mathrm{Range}_{[a,b]}(w^\top x_1 + t)e_{\widetilde{k}_1}, \cdots, \mathrm{Range}_{[a,b]}(w^\top x_n + t)e_{\widetilde{k}_n}]\|_\infty \leq \epsilon. \quad \text{(G.15)}$$

Next, we generalize the above result to the case where each token associates with different $w_i$ and $t_i$ for all $i \in [n]$.

Until now we have

$$\|\text{Attn}(X)_{:,i} - \text{Range}_{[a,b]}(w^\top x_i + t) \cdot e_{\widetilde{k}_i}\|_\infty \leq \max\{|a|, |b|\} \cdot \epsilon_0 + \frac{b-a}{p}, \quad i \in [n]. \quad \text{(G.16)}$$

First, we combine the bias term $t$ into $w$ by augmenting the input $x_i \in \mathbb{R}^d$ with 1 such that $x_i' := [x_i^\top; 1] \in \mathbb{R}^{d+1}$ and $w' := [w^\top; t] \in \mathbb{R}^{d+1}$. This ensures that $w'^\top x_i'$ absorbs the bias term $t$ for all $i \in [n]$.

Thus we have

$$\|\text{Attn}(X)_{:,i} - \text{Range}_{[a,b]}(w^\top x_i + t) \cdot e_{\widetilde{k}_i}\|_\infty = \|\text{Attn}(X)_{:,i} - \text{Range}_{[a,b]}(w'^\top x_i') \cdot e_{\widetilde{k}_i}\|_\infty,$$

where the equality is by absorbing $t$ into $w' = [w, t]$.

Then, we multiply each token $x_i'$ element-wise by a trainable vector $v_i'$, i.e., $x_i' \odot v_i' \in \mathbb{R}^{d+1}$. Effectively, since $w'^\top(x_i' \odot v_i') = w_i'^\top x_i'$ with $w_i' := w' \odot v_i'$, we have

$$\begin{aligned}
&\|\text{Attn}(X)_{:,i} - \text{Range}_{[a,b]}(w'^\top(x_i' \odot v_i')) \cdot e_{\widetilde{k}_i}\|_\infty \\
&= \|\text{Attn}(X)_{:,i} - \text{Range}_{[a,b]}((w' \odot v_i')^\top x_i') \cdot e_{\widetilde{k}_i}\|_\infty &&\text{(Reorder the element-wise multiplication)}\\
&= \|\text{Attn}(X)_{:,i} - \text{Range}_{[a,b]}(w_i'^\top x_i') \cdot e_{\widetilde{k}_i}\|_\infty &&\text{(By } w_i' := w' \odot v_i')\\
&= \|\text{Attn}(X)_{:,i} - \text{Range}_{[a,b]}(w_i^\top x_i + t_i) \cdot e_{\widetilde{k}_i}\|_\infty,
\end{aligned}$$

where the last line is by $w_i' = [w_i, t_i]$.

**Remark G.4.** *We remark that, this elementwise multiplication of trainable vector is only a technicality for keeping our result general. Specifically, this make each token have different truncated linear models.*

Thus (G.16) generalizes to the following equation when multiplying each $x_i'$ element-wise by a trainable $v_i'$

$$\|\text{Attn}(X)_{:,i} - \text{Range}_{[a,b]}(w_i^\top x_i + t_i) \cdot e_{\widetilde{k}_i}\|_\infty \leq \max\{|a|, |b|\} \cdot \epsilon_0 + \frac{b-a}{p}, \quad i \in [n].$$

This completes the proof. $\qquad\square$

**Remark G.5** (Explicit $O(1/p)$ Rate). *Let $M := \max\{|a|, |b|\}$. If we choose $\epsilon_0 = 1/p$ and plug $\Delta L = (b-a)/p$ into $\beta$, it suffices to take*

$$\beta \geq \frac{2p^2}{(b-a)^2} \ln(p(p-2)),$$

*which ensures $M\epsilon_0 = M/p$ and hence*

$$\|\text{Attn} \circ A(X)_{:,i} - \text{Range}_{[a,b]}(w_i^\top x_i + t_i)e_{\widetilde{k}_i}\|_\infty \leq \frac{M + (b-a)}{p}.$$

*Thus the total approximation error decays as $O(1/p)$ with an explicit constant $M + (b-a)$, and the required $\beta$ grows as $\beta(p) = O(p^2 \log p)$.*

*For later convenience, here we recast Theorem 3.4 into an "arbitrary precision" version.*

**Corollary G.6** (Arbitrary Precision with Explicit Parameters)**.** *Let* $a < b$ *and set* $M :=$ $\max\{|a|, |b|\}$. *For any* $\epsilon > 0$, *choose*

$$p \geq \max\left\{ n+1, \left\lceil \frac{2(b-a)}{\epsilon} \right\rceil \right\} \quad and \quad \beta \geq \frac{2p^2}{(b-a)^2}(\ln(p-2) + \ln\frac{2M}{\epsilon}).$$

*Then the single-layer, single-head self-attention construction in Theorem 3.4 satisfies*

$$\left\| \mathrm{Attn}(X) - [\,\mathrm{Range}_{[a,b]}(w^\top x_1 + t)e_{\widetilde{k}_1}, \ldots, \mathrm{Range}_{[a,b]}(w^\top x_n + t)e_{\widetilde{k}_n}\,] \right\|_\infty \leq \epsilon.$$

*Proof.* By Theorem 3.4 we know

$$\epsilon = \underbrace{M\,\epsilon_0}_{\text{softmax error}} + \underbrace{\frac{b-a}{p}}_{\text{interpolation error}}.$$

Choose $\epsilon_0 = \epsilon/(2M)$ so the softmax error is $\epsilon/2$. Plug it into $\beta$ together with $\Delta L = (b-a)/p$ we have

$$\beta \geq \frac{\ln(p-2) - \ln\epsilon_0}{(\Delta L)^2/2} = \frac{2p^2}{(b-a)^2}(\ln(p-2) + \ln\frac{2M}{\epsilon}).$$

This guarantees the softmax term is $\leq \epsilon/2$. For the interpolation error, require

$$\frac{b-a}{p} \leq \frac{\epsilon}{2},$$

which is equal to

$$p \geq \frac{2(b-a)}{\epsilon}.$$

Finally take $p \geq n+1$ to ensure $p > n$. Summing the two halves gives total error $\leq \epsilon$. $\square$

G.3    PROOF OF THEOREM 3.9

To approximate a truncated linear function using multi-head attention, we partition the interval $[\widetilde{L}_0, \widetilde{L}_{H(n-2)}]$ into $H$ sub-intervals, each head handles $n-2$ interpolation points. For any scalar value $a$, we need to know which heads are responsible for it, that is whose interpolation range contains $a$. The next lemma shows that at most two adjacent heads cover the same $a$. This lemma enables a simplified case analysis later in the main theorem's proof.

**Lemma G.7** (Cases of All Heads in $\mathrm{Attn}^H$)**.** *For* $a \in [\widetilde{L}_0, \widetilde{L}_{H(n-2)}]$. *For any* $h \in [H]$, *define three cases of the relationship between* $a$ *and* $h$

- **Case 1:** $a \in [\widetilde{L}_{(h-1)(n-2)}, \widetilde{L}_{h(n-2)-1}]$,

- **Case 2:** $a \notin [\widetilde{L}_{(h-1)(n-2)-1}, \widetilde{L}_{h(n-2)}]$.

- **Case 3:** $a \in [\widetilde{L}_{(h-1)(n-2)-1}, \widetilde{L}_{(h-1)(n-2)}] \cup [\widetilde{L}_{h(n-2)-1}, \widetilde{L}_{h(n-2)}]$.

*These cases includes all possible situation. Then for all* $h$, *only two cases exists*

- *$a$ falls in Case 1 for an $h$ and Case 2 for all others.*

- *$a$ falls in Case 3 for two adjacent $h$ and Case 2 for all others.*

*Proof.* Because $a \in [\widetilde{L}_0, \widetilde{L}_{H(n-2)}]$ and

$$[\widetilde{L}_0, \widetilde{L}_{H(n-2)}] = \cup_{h=1}^{H}[\widetilde{L}_{(h-1)(n-2)}, \widetilde{L}_{h(n-2)}].$$

Thus

$$a \in [\widetilde{L}_{(h_a-1)(n-2)}, \widetilde{L}_{h_a(n-2)}] \tag{G.17}$$

for an $h_a$. This leads to only two possible cases

- Case 1*: $a \in [\widetilde{L}_{(h_a-1)(n-2)}, \widetilde{L}_{h_a(n-2)-1}]$.

- Case 2*: $a \in [\widetilde{L}_{h_a(n-2)-1}, \widetilde{L}_{h_a(n-2)}]$.

**Case 1*:** $a \in [\widetilde{L}_{(h_a-1)(n-2)}, \widetilde{L}_{h_a(n-2)-1}]$. Because $a \in [\widetilde{L}_{(h_a-1)(n-2)}, \widetilde{L}_{h_a(n-2)-1}]$, thus for $h \neq h_a$, we have

$$\widetilde{L}_{h(n-2)-2}, \widetilde{L}_{h(n-2)} < \widetilde{L}_{(h_a-1)(n-2)}, \quad h < h_a$$
$$\widetilde{L}_{h(n-2)+1}, \widetilde{L}_{(h-1)(n-2)-1} \geq \widetilde{L}_{h_a(n-2)-1}, \quad h > h_a.$$

Thus

$$[\widetilde{L}_{(h_a-1)(n-2)}, \widetilde{L}_{h_a(n-2)-1}] \cap [\widetilde{L}_{(h-1)(n-2)-1}, \widetilde{L}_{h(n-2)}] = \emptyset$$
$$[\widetilde{L}_{(h_a-1)(n-2)}, \widetilde{L}_{h_a(n-2)-1}] \cap [\widetilde{L}_{(h-1)(n-2)-1}, \widetilde{L}_{h(n-2)}] = \emptyset$$

for all $h \neq h_a$.

This means that $a$ does not fall into Case 1 nor Case 3 for other $h \in [H]$. Thus $a$ has to fall into Case 2 for other $h$.

**Case 2*:** $a \in [\widetilde{L}_{(h_a-1)(n-2)}, \widetilde{L}_{(h_a-1)(n-2)+1}] \cup [\widetilde{L}_{h_a(n-2)-1}, \widetilde{L}_{h_a(n-2)}]$. Without loss of generality, assume $a$ to be in the left half $[\widetilde{L}_{(h_a-1)(n-2)}, \widetilde{L}_{(h_a-1)(n-2)+1}]$. Because

$$[\widetilde{L}_{(h_a-1)(n-2)}, \widetilde{L}_{(h_a-1)(n-2)+1}] = [\widetilde{L}_{(h_a-1)(n-2)-1}, \widetilde{L}_{(h_a-1)(n-2)}] \qquad \text{(Case 3 of } h_a - 1)$$
$$[\widetilde{L}_{(h_a-1)(n-2)}, \widetilde{L}_{(h_a-1)(n-2)+1}] = [\widetilde{L}_{(h_a-1)(n-2)-1}, \widetilde{L}_{(h_a-1)(n-2)}]. \qquad \text{(Case 3 of } h_a)$$

This means $a$ falls into Case 3 for $h_a$ and $h_a - 1$.

This completes the proof. $\qquad \square$

**Theorem G.8** (Theorem 3.9 Restated: Multi-Head Attention Approximate Truncated Linear Models). *Fix real numbers $a < b$, and let the truncation operator $\text{Range}_{[a,b]}(\cdot)$ follow Definition 3.1. For a precision parameter $p > n$ with $\epsilon = O(1/p)$, number of head $H = p/(n-2)$ there exists a single-layer, $H$-head self-attention $\text{Attn}^H$ with a linear transformation $A : \mathbb{R}^{d \times n} \to \mathbb{R}^{(d+n) \times n}$, such that $\text{Attn}^H \circ A : \mathbb{R}^{d \times n} \to \mathbb{R}^{d_o \times n}$ satisfies, for any $i \in [n]$,*

$$\|\text{Attn}^H \circ A(X)_{:,i} - \text{Range}_{[a,b]}(w_i^\top x_i + t_i)e_{\widetilde{k}_i}\|_\infty \leq \underbrace{\max\{|a|, |b|\} \cdot \epsilon_0}_{\text{finite-}\beta \text{ softmax error}} + \underbrace{\frac{b-a}{(n-2)H}}_{\text{interpolation error}} .$$

*Here $e_{\widetilde{k}_i}$ is a one-hot vector with a value of $1$ at the $\widetilde{k}_i$-th index and $0$ elsewhere, and*

$$k_i := \underset{k \in \{0,1,2,\cdots,p-1\}}{\arg\min} |x_i^\top w + t - \widetilde{L}_k| \quad \text{where} \quad \widetilde{k}_i := G(k_i) \in [d_o]. \tag{G.18}$$

*Here $k_i \in \{0, ..., p-1\}$ is the index of the interpolation point closest to the $i$-th token ($i$-th truncated linear model). For all $i \in [n]$, $G : \{0, ..., p-1\} \to [d_o]$ denotes any set-to-set function sending the interpolation index $k \in \{0, ..., p-1\}$ into a position index $\widetilde{k} \in [d_o]$ specifying in the desired row index of the output.*

*Proof.* Define $A : \mathbb{R}^{d \times n} \to \mathbb{R}^{(d+n) \times n}$ for the input sequence $X$ as

$$A(X) := \underbrace{\begin{bmatrix} I_d \\ 0_{n \times d} \end{bmatrix}}_{(d+n) \times d} X + \underbrace{\begin{bmatrix} 0_{d \times n} \\ I_n \end{bmatrix}}_{(d+n) \times n} = \begin{bmatrix} X \\ I_n \end{bmatrix} \in \mathbb{R}^{d+n}.$$

Thus, A is a token-wise linear layer augmented with positional encoding, as it applies a linear projection to each token and then adds a unique per-token bias.

Let $p$ be a precision parameter, without loss of generality, let it be divisible by $n-2$ and denote $p/(n-2)$ as $H$.

Now we define the multi-head attention $\mathrm{Attn}$ of $H$ heads. Denote $\ell_k := k(\widetilde{L}_k + \widetilde{L}_0) - 2kt$ as in Theorem 3.4. We denote the $h$-th head as $\mathrm{Attn}_h$, and define the weight matrices as

$$W_K^{(h)} = -\beta \begin{bmatrix} 0_{d \times d} & -2[(h-1)(n-2)-1]w & -2(h-1)(n-2)w & \cdots & -2h(n-2)w \\ 0_d^\top & \ell_{(h-1)(n-2)-1} & \ell_{(h-1)(n-2)} & \cdots & \ell_{h(n-2)} \end{bmatrix},$$

$$W_Q^{(h)} = \begin{bmatrix} I_d & 0_{d \times n} \\ 0_d^\top & 1_n^\top \end{bmatrix},$$

$$W_V^{(h)} = \begin{bmatrix} 0_{d_o \times (d+1)} & \widetilde{L}_{(h-1)(n-2)}e_{\widetilde{k}_{(h-1)(n-2)}} & \widetilde{L}_{(h-1)(n-2)+1}e_{\widetilde{k}_{(h-1)(n-2)+1}} & \cdots & \widetilde{L}_{h(n-2)-1}e_{\widetilde{k}_{h(n-2)-1}} & 0_{d_o} \end{bmatrix},$$

for every $h \in [H]$. Here $\beta > 0$ is a coefficient we use to control the precision of our approximation. The attention reaches higher precision as $\beta$ gets larger.

With the construction of weights, we are also able to calculate the $K$, $Q$, $V$ matrices in $\mathrm{Attn}$

$$K^{(h)} := W_K^{(h)} A(X) \tag{G.19}$$

$$= -\beta \begin{bmatrix} 0_{d \times d} & -2[(h-1)(n-2)-1]w & -2(h-1)(n-2)w & \cdots & -2h(n-2)w \\ 0_d^\top & \ell_{(h-1)(n-2)-1} & \ell_{(h-1)(n-2)} & \cdots & \ell_{h(n-2)} \end{bmatrix} \cdot \begin{bmatrix} X \\ I_n \end{bmatrix}$$

$$= -\beta \begin{bmatrix} -2[(h-1)(n-2)-1]w & -2(h-1)(n-2)w & \cdots & -2h(n-2)w \\ \ell_{(h-1)(n-2)-1} & \ell_{(h-1)(n-2)} & \cdots & \ell_{h(n-2)} \end{bmatrix} \in \mathbb{R}^{(d+1) \times n} \tag{G.20}$$

where the last equality comes from multiplying $X$ with 0, thus this is a extraction of non-zero entries in $W_K$.

For $Q$, we have

$$Q^{(h)} := W_Q^h A(X)$$

$$= \begin{bmatrix} I_d & 0_{d \times n} \\ 0_d^\top & 1_n^\top \end{bmatrix} \cdot \begin{bmatrix} X \\ I_n \end{bmatrix}$$

$$= \underbrace{\begin{bmatrix} I_d \cdot X + 0_{d \times n} \cdot I_n \\ 0_{1 \times d} \cdot X + 1_{1 \times n} \cdot I_n \end{bmatrix}}_{(d+1) \times n}$$

$$= \begin{bmatrix} X \\ 1_{1 \times n} \end{bmatrix}. \tag{G.21}$$

For $V$, we have

$$V^{(h)} := W_V^{(h)} A(X)$$

$$= \begin{bmatrix} 0_{d_o \times (d+1)} & \widetilde{L}_{(h-1)(n-2)} e_{\widetilde{k}_{(h-1)(n-2)}} & \cdots & \widetilde{L}_{h(n-2)-1} e_{\widetilde{k}_{h(n-2)-1}} & 0_{d_o} \end{bmatrix} \cdot \begin{bmatrix} X \\ I_n \end{bmatrix}$$

$$= \underbrace{0}_{d_o \times d} \cdot X + \underbrace{\begin{bmatrix} 0_{d_o} & \widetilde{L}_{(h-1)(n-2)} e_{\widetilde{k}_{(h-1)(n-2)}} & \cdots & \widetilde{L}_{h(n-2)-1} e_{\widetilde{k}_{h(n-2)-1}} & 0_{d_o} \end{bmatrix}}_{d_o \times n} \cdot I_n$$

$$= \begin{bmatrix} 0_{d_o} & \widetilde{L}_{(h-1)(n-2)} e_{\widetilde{k}_{(h-1)(n-2)}} & \widetilde{L}_{(h-1)(n-2)+1} e_{\widetilde{k}_{(h-1)(n-2)+1}} & \cdots & \widetilde{L}_{h(n-2)-1} e_{\widetilde{k}_{h(n-2)-1}} & 0_{d_o} \end{bmatrix},$$
$$\tag{G.22}$$

Given that all $\widetilde{k}_j$, for $j \in [p]$, share the same identical number in $[d_o]$, we denote this number by $k_G$.

> **Remark G.9.** *This theorem have all the $\widetilde{k}_j$ as the same for simplicity. This version of identical $\widetilde{k}_j$ is also what subsequent theorems on universal approximations use.*

Hence we rewrite $V^{(h)}$ as

$$V^{(h)} = \begin{bmatrix} 0_{d_o} & \widetilde{L}_{(h-1)(n-2)} e_{k_G} & \widetilde{L}_{(h-1)(n-2)+1} e_{k_G} & \cdots & \widetilde{L}_{h(n-2)-1} e_{k_G} & 0_{d_o} \end{bmatrix}.$$

We define $m_v$ as

$$m_v := \max\{|a|, |b|\}.$$

By the definition of $V^{(h)}$, we have

$$\|V\|_\infty \le \max_{i \in [P]} \{\widetilde{L}_i\} \le m_v. \tag{G.23}$$

> **Remark G.10** (Intuition of the Construction of $V^{(h)}$). *As previously mentioned, $\widetilde{L}_i$, for $i \in [p]$, are all the interpolations. In this context, $V^{(h)}$ encompasses the $(n-2)$ elements of these interpolations (i.e., $(h-1)(n-2)$ to $h(n-2)-1$). Meanwhile, the value on the two ends of $V^h$ are both set to $0_{d_o}$, because we suppress the head and let it output $0$ when the input $X$ is not close enough to the interpolations of the head.*

Now we are ready to calculate the output of each $\text{Attn}_h$

$$\text{Attn}_h(A(X))$$
$$= V^{(h)} \text{Softmax}((K^{(h)})^\top Q^{(h)})$$
$$= V \text{Softmax} \left( -\beta \begin{bmatrix} -2[(h-1)(n-2)-1]w & -2(h-1)(n-2)w & \cdots & -2h(n-2)w \\ \ell_{(h-1)(n-2)-1} & \ell_{(h-1)(n-2)} & \cdots & \ell_{h(n-2)} \end{bmatrix}^\top \begin{bmatrix} X \\ 1_{1 \times n} \end{bmatrix} \right),$$

where last line is by plug in (G.19) and (G.21). Note the $i$-th column of the attention score matrix (the Softmax nested expression) is equivalent to the following expressions

$$\text{Softmax}((K^{(h)})^\top Q^{(h)})_{:,i}$$
$$= \text{Softmax} \left( -\beta \begin{bmatrix} -2[(h-1)(n-2)-1]w & -2(h-1)(n-2)w & \cdots & -2h(n-2)w \\ \ell_{(h-1)(n-2)-1} & \ell_{(h-1)(n-2)} & \cdots & \ell_{h(n-2)} \end{bmatrix}^\top \begin{bmatrix} X \\ 1_{1 \times n} \end{bmatrix} \right)_{:,i}$$

$$
= \mathsf{Softmax}\left(-\beta \begin{bmatrix} -2[(h-1)(n-2)-1]w^\top x_i + \ell_{(h-1)(n-2)-1} \\ -2(h-1)(n-2)w^\top x_i + \ell_{(h-1)(n-2)} \\ \vdots \\ -2h(n-2)w^\top x_i + \ell_{h(n-2)} \end{bmatrix}\right) \qquad \text{(pick column } i)
$$

$$
= \mathsf{Softmax}\left(-\beta \begin{bmatrix} [(h-1)(n-2)-1](-2w^\top x_i + \widetilde{L}_{(h-1)(n-2)-1} + \widetilde{L}_0) - 2[(h-1)(n-2)-1]t \\ (h-1)(n-2)(-2w^\top x_i + \widetilde{L}_{(h-1)(n-2)} + \widetilde{L}_0) - 2(h-1)(n-2)t \\ \vdots \\ h(n-2)(-2w^\top x_i + \widetilde{L}_{h(n-2)} + \widetilde{L}_0) - 2h(n-2)t \end{bmatrix}\right)
$$
$$
\text{(By } \ell_k = k(\widetilde{L}_k + \widetilde{L}_0) - 2kt)
$$

$$
= \mathsf{Softmax}\left(-\frac{\beta}{\Delta L} \begin{bmatrix} (-2x_i^\top w - 2t + \widetilde{L}_0 + \widetilde{L}_{(h-1)(n-2)-1}) \cdot [(h-1)(n-2)-1]\Delta L \\ (-2x_i^\top w - 2t + \widetilde{L}_0 + \widetilde{L}_{(h-1)(n-2)}) \cdot (h-1)(n-2)\Delta L \\ \vdots \\ (-2x_i^\top w - 2t + \widetilde{L}_0 + \widetilde{L}_{h(n-2)}) \cdot h(n-2)\Delta L \end{bmatrix}\right)
$$
$$
\text{(By mutiplying and dividing by } \Delta L)
$$

$$
= \mathsf{Softmax}\left(-\frac{\beta}{\Delta L} \begin{bmatrix} (-2x_i^\top w - 2t + \widetilde{L}_0 + \widetilde{L}_{(h-1)(n-2)-1}) \cdot (\widetilde{L}_{(h-1)(n-2)-1} - \widetilde{L}_0) \\ (-2x_i^\top w - 2t + \widetilde{L}_0 + \widetilde{L}_{(h-1)(n-2)}) \cdot (\widetilde{L}_{(h-1)(n-2)} - \widetilde{L}_0) \\ \vdots \\ (-2x_i^\top w - 2t + \widetilde{L}_0 + \widetilde{L}_{h(n-2)}) \cdot (\widetilde{L}_{h(n-2)} - \widetilde{L}_0) \end{bmatrix}\right)
$$
$$
\text{(By } k\Delta L = \widetilde{L}_k - \widetilde{L}_0)
$$

$$
= \mathsf{Softmax}\left(-\frac{\beta}{\Delta L} \begin{bmatrix} (-2x_i^\top w - 2t) \cdot \widetilde{L}_{(h-1)(n-2)-1} + (\widetilde{L}_{(h-1)(n-2)-1})^2 + (x_i^\top w + t)^2 \\ (-2x_i^\top w - 2t) \cdot \widetilde{L}_{(h-1)(n-2)} + (\widetilde{L}_{(h-1)(n-2)})^2 + (x_i^\top w + t)^2 \\ \vdots \\ (-2x_i^\top w - 2t) \cdot \widetilde{L}_{h(n-2)} + (\widetilde{L}_{h(n-2)})^2 + (x_i^\top w + t)^2 \end{bmatrix}\right)
$$

$$
= \mathsf{Softmax}\left(-\frac{\beta}{\Delta L} \begin{bmatrix} (x_i^\top w + t - \widetilde{L}_{(h-1)(n-2)-1})^2 \\ (x_i^\top w + t - \widetilde{L}_{(h-1)(n-2)})^2 \\ \vdots \\ (x_i^\top w + t - \widetilde{L}_{h(n-2)})^2 \end{bmatrix}\right). \tag{G.24}
$$

Here, the last-second equality arises from the fact that the softmax function is shift-invariant, allowing us to subtract and add a constant across all coordinates. To be more precise, we first expand the product for $k$-th coordinate of the column vector

$$
(-2x_i^\top w - 2t + \widetilde{L}_0 + \widetilde{L}_k)(\widetilde{L}_k - \widetilde{L}_0)
$$
$$
= (-2x_i^\top w - 2t)L_k + L_0 L_k + L_k^2 - (-2x_i^\top w - 2t)L_0 - L_0^2 - L_0 L_k
$$
$$
= (-2x_i^\top w - 2t)L_k + L_k^2 - \underbrace{(-2x_i^\top w - 2t)L_0 - L_0^2}_{\text{constant across the column vector}}.
$$

Then, dropping the constant and adding another constant $(x_i^\top w + t)^2$ across all coordinates, above equation becomes

$$
(-2x_i^\top w - 2t)L_k + L_k^2 + (x_i^\top w + t)^2 = (x_i^\top w + t - L_k)^2.
$$

Hence we finish the derivation of (G.24). Thus we have

$$
\text{Attn}_h(A(X))_{:,i} = V\text{Softmax}\left(-\frac{\beta}{\Delta L}\begin{bmatrix}(x_i^\top w + t - \widetilde{L}_{(h-1)(n-2)-1})^2\\ (x_i^\top w + t - \widetilde{L}_{(h-1)(n-2)})^2\\ \vdots\\ (x_i^\top w + t - \widetilde{L}_{h(n-2)})^2\end{bmatrix}\right). \tag{G.25}
$$

For a specific $h$, we calculate the result of (G.25) column by column. Let $X_i$ denote any column (token) of the matrix $X$. We partition the situation at each column (token) into three distinct cases:

- **Case 1:** $w^\top X_i + t$ is strictly within the interpolation range of $\text{Attn}_h$ ($X \in [\widetilde{L}_{(h-1)(n-2)}, \widetilde{L}_{h(n-2)-1}]$). This excludes the following range at the edge of the interpolation range of

$$
[\widetilde{L}_{(h-1)(n-2)-1}, \widetilde{L}_{(h-1)(n-2)}] \cup [\widetilde{L}_{h(n-2)-1}, \widetilde{L}_{h(n-2)}].
$$

- **Case 2:** $w^\top X_i + t$ is not within the interpolation range of $\text{Attn}_h$:

$$
w^\top X_i + t \notin [\widetilde{L}_{(h-1)(n-2)-1}, \widetilde{L}_{h(n-2)}].
$$

- **Case 3:** $w^\top X_i + t$ is on the edge (region) of the interpolation range of $\text{Attn}_h$:

$$
w^\top X_i + t \in [\widetilde{L}_{(h-1)(n-2)-1}, \widetilde{L}_{(h-1)(n-2)}] \cup [\widetilde{L}_{h(n-2)-1}, \widetilde{L}_{h(n-2)}].
$$

**Remark G.11** (Description of All Cases of a Single Head Attention). *The H heads equally split the task of approximating the truncated linear function. Namely and explicitly,*

$$
\|\text{Attn}_h(X) - \text{Range}_{[a+\frac{b-a}{p}((h-1)(n-2)-1),a+\frac{b-a}{p}h(n-2)]}(X)\|_\infty \leq \epsilon_1,
$$

*where $\epsilon > 0$ is arbitrarily small.*

*With this understanding, **Case 1**, **Case 2** and **Case 3** correspond to the different scenarios that may arise when approximating the expression*

$$
\text{Range}_{[a+\frac{b-a}{p}((h-1)(n-2)-1),a+\frac{b-a}{p}h(n-2)]}(\cdot).
$$

*Here, we provide an informal yet intuitive explanation of the three cases as follows:*

- ***Case 1:** $w^\top X_i + t$ falls in the interior of the interpolation range of the $h$-th head $\text{Attn}_h$, denoted as $\text{Range}_{[a+(b-a)((h-1)(n-2)-1)/p,a+(b-a)h(n-2)/p]}$.*

- ***Case 2:** $w^\top X_i + t$ is outside the the interpolation range of the $h$-th head $\text{Attn}_h$, which is $\text{Range}_{[a+(b-a)((h-1)(n-2)-1)/p,a+(b-a)h(n-2)/p]}$.*

- ***Case 3:** $w^\top X_i + t$ falls on the boundary of the interpolation range of the $h$-th head $\text{Attn}_h$.*

**Remark G.12** (Cases of All Attention Heads). *According to Lemma G.7, for all heads in $\text{Attn}^H$, there are two possible cases:*

- ***Case 1\*:** $x$ falls into Case 1 for a head, and Case 2 for all other heads.*

- **Case 2\*:** $x$ falls into Case 3 for two heads with adjacent interpolation ranges, and Case 2 for other heads.

  This also means that when Case 1 appears in $\mathrm{Attn}^H$, the situation of all head in $\mathrm{Attn}^H$ falls into Case 1\*. And when Case 3 appears in $\mathrm{Attn}^H$, the situation of all head in $\mathrm{Attn}^H$ falls into Case 2\*. Thus, We discuss Case 2\* in the discussion of Case 3.

**Case 1:** $X_i \in [\widetilde{L}_{(h-1)(n-2)}, \widetilde{L}_{h(n-2)-1}]$. In this case, our goal is to demonstrate this attention head outputs a value close to $\mathrm{Range}_{[a,b]}(w^\top X_i + t)$.

Let $\widetilde{L}_s$ and $\widetilde{L}_{s+1}$ be the two interpolants such that

$$w^\top X_i + t \in [\widetilde{L}_s, \widetilde{L}_{s+1}]. \tag{G.26}$$

Then, $s$ and $s+1$ are also the labels of the two largest entries in

$$-\frac{\beta}{\Delta L} \begin{bmatrix} (w^\top X_i + t - \widetilde{L}_{(h-1)(n-2)-1})^2 \\ (w^\top X_i + t - \widetilde{L}_{(h-1)(n-2)})^2 \\ \vdots \\ (w^\top X_i + t - \widetilde{L}_{h(n-2)})^2 \end{bmatrix},$$

since

$$\operatorname*{argmax}_{k \in \{(h-1)(n-2)-1, h(n-2)\}} -\frac{\beta}{\Delta L}(w^\top X_i + t - \widetilde{L}_k)^2$$

$$= \operatorname*{argmin}_{k \in \{(h-1)(n-2)-1, h(n-2)\}} (w^\top X_i + t - \widetilde{L}_k)^2$$

$$= \operatorname*{argmin}_{k \in \{(h-1)(n-2)-1, h(n-2)\}} |w^\top X_i + t - \widetilde{L}_k|.$$

We also note that the distance of $w^\top X_i + t$ to interpolants beside $\widetilde{L}_s$ and $\widetilde{L}_{s+1}$ differs from $w^\top X_i + t$ for at least $\widetilde{L}_s - \widetilde{L}_{s-1} = (b-a)/p$ or $\widetilde{L}_{s+1} - \widetilde{L}_s = (b-a)/p$.

This is equivalent to the occasion when $x_1 - x_3$ in Lemma F.1 is larger than

$$\max\left\{ \frac{\beta}{\Delta L}(w^\top X_i + t - \widetilde{L}_{s-1})^2 - (w^\top X_i + t - \widetilde{L}_s)^2, \frac{\beta}{\Delta L}(w^\top X_i + t - \widetilde{L}_{s+2})^2 - (w^\top X_i + t - \widetilde{L}_{s+1})^2 \right\}$$

$$\geq \frac{\beta}{\Delta L} \cdot (\frac{b-a}{p})^2,$$

which is invariant to $X_i$.

Thus according to Lemma F.1 and the fact that the $s$ and $s+1$ are the two largest entries in the $i$-th column of the attention score matrix, we have

$$\left\| \mathsf{Softmax}\left( -\frac{\beta}{\Delta L} \begin{bmatrix} (w^\top X_i + t - \widetilde{L}_{(h-1)(n-2)-1})^2 \\ (w^\top X_i + t - \widetilde{L}_{(h-1)(n-2)})^2 \\ \vdots \\ (w^\top X_i + t - \widetilde{L}_{h(n-2)})^2 \end{bmatrix} \right) - \frac{1}{1+e^{-\beta\delta}}\underbrace{e_s}_{n\times1} - \frac{e^{-\beta\delta}}{1+e^{-\beta\delta}}\underbrace{e_{s+1}}_{n\times1} \right\|_\infty \leq \epsilon_2,$$

for any $\epsilon_2 > 0$.

This yields that

$$\left\| V\mathsf{Softmax}\left( -\frac{\beta}{\Delta L}\begin{bmatrix} (w^\top X_i + t - \widetilde{L}_{(h-1)(n-2)-1})^2 \\ (w^\top X_i + t - \widetilde{L}_{(h-1)(n-2)})^2 \\ \vdots \\ (w^\top X_i + t - \widetilde{L}_{h(n-2)})^2 \end{bmatrix}\right) - V\frac{1}{1+e^{-\beta\delta}}e_s - V\frac{e^{-\beta\delta}}{1+e^{-\beta\delta}}e_{s+1}\right\|_\infty$$

$$\leq \left\| \mathsf{Softmax}\left( -\frac{\beta}{\Delta L}\begin{bmatrix} (w^\top X_i + t - \widetilde{L}_{(h-1)(n-2)-1})^2 \\ (w^\top X_i + t - \widetilde{L}_{(h-1)(n-2)})^2 \\ \vdots \\ (w^\top X_i + t - \widetilde{L}_{h(n-2)})^2 \end{bmatrix}\right) - \frac{1}{1+e^{-\beta\delta}}e_s - \frac{e^{-\beta\delta}}{1+e^{-\beta\delta}}e_{s+1}\right\|_\infty \cdot \|V\|_\infty$$

$$\leq \|V\|_\infty \epsilon_2.$$

This is equivalent to

$$\|V\mathsf{Softmax}(K^\top Q)_{:,i} - \frac{1}{1+e^{-\beta\delta}}\widetilde{L}_{(h-1)(n-2)+s-1}e_{k_G} - \frac{e^{-\beta\delta}}{1+e^{-\beta\delta}}\widetilde{L}_{(h-1)(n-2)+s}e_{k_G}\|_\infty$$

$$\leq \|V\|_\infty \cdot \epsilon_2 \qquad\qquad \text{(By } \|AB\| \leq \|A\| \cdot \|B\|)$$

$$\leq m_v\epsilon_2, \tag{G.27}$$

where the last line is by (G.23).

From (G.26), we derive that

$$\|\frac{1}{1+e^{-\beta\delta}}\widetilde{L}_{(h-1)(n-2)+s-1} + \frac{e^{-\beta\delta}}{1+e^{-\beta\delta}}\widetilde{L}_{(h-1)(n-2)+s} - (w^\top X_i + t)e_{k_G}\|_\infty$$

$$\leq \|\frac{1}{1+e^{-\beta\delta}}(\widetilde{L}_{(h-1)(n-2)+s-1} - (w^\top X_i + t)e_{k_G})\|_\infty + \|\frac{e^{-\beta\delta}}{1+e^{-\beta\delta}}(\widetilde{L}_{(h-1)(n-2)+s} - (w^\top X_i + t))\|_\infty$$

$$\text{(By convex combination of } (w^\top X_i + t) \text{ and triangle inequality)}$$

$$\leq \frac{1}{1+e^{-\beta\delta}} \cdot \frac{b-a}{p} + \frac{e^{-\beta\delta}}{1+e^{-\beta\delta}} \cdot \frac{b-a}{p} \qquad\qquad \text{(By (G.26))}$$

$$= \frac{b-a}{p}. \tag{G.28}$$

Combing (G.27) and (G.28) yields

$$\|V\mathsf{Softmax}(K^\top Q)_{:,i} - (w^\top X_i + t)\|_\infty$$

$$\leq \|V\mathsf{Softmax}(K^\top Q)_{:,i} - \frac{1}{1+e^{-\beta\delta}}\widetilde{L}_{(h-1)(n-2)+s-1} - \frac{e^{-\beta\delta}}{1+e^{-\beta\delta}}\widetilde{L}_{(h-1)(n-2)+s}\|_\infty$$

$$+ \|\frac{1}{1+e^{-\beta\delta}}\widetilde{L}_{(h-1)(n-2)+s-1} + \frac{e^{-\beta\delta}}{1+e^{-\beta\delta}}\widetilde{L}_{(h-1)(n-2)+s} - (w^\top X_i + t)e_{k_G}\|_\infty$$

$$\text{(By triangle inequality)}$$

$$\leq m_v\epsilon_2 + \frac{b-a}{p}, \tag{G.29}$$

where the first inequality comes from adding and subtracting the interpolation points' convex combination and then applying triangle inequality.

**Case 2:** $X \notin [\widetilde{L}_{(h-1)(n-2)-1}, \widetilde{L}_{h(n-2)}]$. In this case, $X_i$ falls out of the range of interpolation covered by $\mathsf{Attn}_h$.

Without loss of generality, suppose $w^\top X_i + t$ to lie left to the range of interpolation of $\mathsf{Attn}_h$.

This yields that $\widetilde{L}_{(h-1)(n-2)-1}$ is the closest interpolant within $\text{Attn}_h$ to $w^\top X_i + t$. Furthermore, the second closest interpolant $\widetilde{L}_{(h-1)(n-2)}$ is at least further for at least $(b-a)/p$, which is a constant irrelevant to $X_i$

Then by Lemma F.1, we have

$$\left\| \text{Softmax}\left( -\frac{\beta}{\Delta L} \begin{bmatrix} (w^\top X_i + t - \widetilde{L}_{(h-1)(n-2)-1})^2 \\ (w^\top X_i + t - \widetilde{L}_{(h-1)(n-2)})^2 \\ \vdots \\ (w^\top X_i + t - \widetilde{L}_{h(n-2)})^2 \end{bmatrix} \right) - \underbrace{e_1}_{n \times 1} \right\|_\infty \le \epsilon_3,$$

for any $\epsilon_3 > 0$.

This yields that

$$\left\| V\text{Softmax}\left( -\frac{\beta}{\Delta L} \begin{bmatrix} (w^\top X_i + t - \widetilde{L}_{(h-1)(n-2)-1})^2 \\ (w^\top X_i + t - \widetilde{L}_{(h-1)(n-2)})^2 \\ \vdots \\ (w^\top X_i + t - \widetilde{L}_{h(n-2)})^2 \end{bmatrix} \right) - V\underbrace{e_1}_{n \times 1} \right\|_\infty$$

$$\le \|V\|_\infty \cdot \epsilon_3 \qquad\qquad\qquad\qquad (\text{By } \|AB\| \le \|A\| \cdot \|B\|)$$

$$\le m_v \epsilon_3,$$

where the last line is by (G.23).

This is equivalent to

$$\left\| V\text{Softmax}\left( -\frac{\beta}{\Delta L} \begin{bmatrix} (w^\top X_i + t - \widetilde{L}_{(h-1)(n-2)-1})^2 \\ (w^\top X_i + t - \widetilde{L}_{(h-1)(n-2)})^2 \\ \vdots \\ (w^\top X_i + t - \widetilde{L}_{h(n-2)})^2 \end{bmatrix} \right) - 0_{d_o} \right\|_\infty \le m_v \epsilon_3. \qquad (\text{G.30})$$

**Case 1\*.** According to Lemma G.7, when Case 1 occurs for one head in the $H$ heads of $\text{Attn}^H$, all other head will be in Case 2.

Combining with the result in Case 2, we have the output of all heads as

$$\|\text{Attn}^H(A(X))_{:,i} - (w^\top X_i + t)e_{k_G}\|_\infty$$

$$= \| \sum_{h_0 \in [H]/\{h\}} \text{Attn}_{h_0} \circ A(X)_{:,i}\|_\infty + \|\text{Attn}_h \circ A(X)_{:,i} - (w^\top X_i + t)e_{k_G}\|_\infty$$

$$= (H-1)m_v\epsilon_3 + m_v\epsilon_2 + \frac{b-a}{p} \qquad\qquad (\text{By (G.29) and (G.30)})$$

$$= (H-1)m_v\epsilon_3 + m_v\epsilon_2 + \frac{b-a}{H(n-2)}.$$

Setting $\epsilon_2, \epsilon_3$ to be

$$\epsilon_2 = \frac{\epsilon_0}{2},$$

$$\epsilon_3 = \frac{\epsilon_0}{2(H-1)m},$$

yields the final result.

**Case 3 (and Case 2\*):** $X \in [\widetilde{L}_{(h-1)(n-2)-1}, \widetilde{L}_{(h-1)(n-2)}] \cup [\widetilde{L}_{h(n-2)-1}, \widetilde{L}_{h(n-2)}]$. In this case, $w^\top X_i + t$ is the boundary of the interpolation range of $\mathrm{Attn}_{h_0}$. By Lemma G.7, it should also fall on the boundary of a head with neighboring interpolation range. Without loss of generality, we set it to be $\mathrm{Attn}_{h_0-1}$. Furthermore, Lemma G.7 indicates that $w^\top X_i + t$ should fall on no other interpolation range of any heads beside $\mathrm{Attn}_{h_0}$ and $\mathrm{Attn}_{h_0-1}$.

Combining this with case 2, we have

$$\mathrm{Attn}^H(A(X))_{:,i} = \sum_{h=1}^{H} \mathrm{Attn}_h \circ A(X)_{:,i}$$
$$\in [(-(H-2)m_v\epsilon_3 + \mathrm{Attn}_{h_0} \circ A(X)_{:,i} + \mathrm{Attn}_{h_0-1} \circ A(X)_{:,i}),$$
$$((H-2)m_v\epsilon_3 + \mathrm{Attn}_{h_0} \circ A(X)_{:,i} + \mathrm{Attn}_{h_0-1} \circ A(X)_{:,i})]. \quad \text{(By (G.30))}$$

By Lemma F.1, let $\delta$ denote

$$\delta = \widetilde{L}_{(h-1)(n-2)+s} - (w^\top X_i + t)e_{k_G} - [\widetilde{L}_{(h-1)(n-2)+s} - (w^\top X_i + t)e_{k_G}],$$

we have

$$\|\mathsf{Softmax}((K^{(h)})^\top Q^{(h)}) - (\frac{1}{1+e^{-\beta\delta}}e_1 + \frac{e^{-\beta\delta}}{1+e^{-\beta\delta}}e_2)\| \le \epsilon_4,$$

and

$$\|\mathsf{Softmax}((K^{(h-1)})^\top Q^{(h-1)}) - (\frac{1}{1+e^{-\beta\delta}}e_{n-1} + \frac{e^{-\beta\delta}}{1+e^{-\beta\delta}}e_n)\| \le \epsilon_5,$$

for any $\epsilon_4, \epsilon_5 > 0$.

Thus we have

$$\|V^{(h)}\mathsf{Softmax}((K^{(h)})^\top Q^{(h)}) + V^{(h-1)}\mathsf{Softmax}((K^{(h-1)})^\top Q^{(h-1)})$$
$$- V(\frac{1}{1+e^{-\beta\delta}}e_1 + \frac{e^{-\beta\delta}}{1+e^{-\beta\delta}}e_2 + \frac{1}{1+e^{-\beta\delta}}e_{n-1} + \frac{e^{-\beta\delta}}{1+e^{-\beta\delta}}e_n)\|_\infty$$
$$\le \|V\|_\infty(\epsilon_4 + \epsilon_5).$$

This is equivalent to

$$\|V^{(h)}\mathsf{Softmax}((K^{(h)})^\top Q^{(h)}) + V^{(h-1)}\mathsf{Softmax}((K^{(h-1)})^\top Q^{(h-1)})$$
$$- (\frac{1}{1+e^{-\beta\delta}} \cdot 0 + \frac{e^{-\beta\delta}}{1+e^{-\beta\delta}}e_{k_G}\widetilde{L}_{(h-1)(n-2)+s} + \frac{1}{1+e^{-\beta\delta}}e_{k_G}\widetilde{L}_{(h-1)(n-2)+s-1} + \frac{e^{-\beta\delta}}{1+e^{-\beta\delta}}e_{k_G}) \cdot 0\|_\infty$$
$$\le \|V\|_\infty \cdot (\epsilon_4 + \epsilon_5).$$

Thus we have

$$\|V^{(h)}\mathsf{Softmax}((K^{(h)})^\top Q^{(h)}) + V^{(h-1)}\mathsf{Softmax}((K^{(h-1)})^\top Q^{(h-1)})$$
$$- (\frac{e^{-\beta\delta}}{1+e^{-\beta\delta}}e_{k_G}\widetilde{L}_{(h-1)(n-2)+s} + \frac{1}{1+e^{-\beta\delta}}e_{k_G}\widetilde{L}_{(h-1)(n-2)+s-1})\|_\infty$$
$$\le \|V\|_\infty(\epsilon_4 + \epsilon_5),$$

which implies

$$\|\sum_{h=1}^{H} \mathrm{Attn}_h(A(X))_{:,i} - (\frac{e^{-\beta\delta}}{1+e^{-\beta\delta}}e_{k_G}\widetilde{L}_{(h-1)(n-2)+s} + \frac{1}{1+e^{-\beta\delta}}e_{k_G}\widetilde{L}_{(h-1)(n-2)+s-1})\|_\infty$$

$$\leq (H-2)m_v\epsilon_3 + \|V\|_\infty(\epsilon_4 + \epsilon_5). \tag{G.31}$$

Finally, since

$$\|\frac{e^{-\beta\delta}}{1+e^{-\beta\delta}}e_{k_G}\widetilde{L}_{(h-1)(n-2)+s} + \frac{1}{1+e^{-\beta\delta}}e_{k_G}\widetilde{L}_{(h-1)(n-2)+s-1} - (w^\top X_i + t)e_{k_G}\|_\infty \leq \frac{b-a}{p},$$
$$\text{(By (G.28))}$$

combining with (G.31), we have

$$\|\sum_{h=1}^H \text{Attn}_h(A(X))_{:,i} - (w^\top X_i + t)e_{k_G}\|_\infty$$

$$\leq \|\sum_{h=1}^H \text{Attn}_h(A(X))_{:,i} - (\frac{e^{-\beta\delta}}{1+e^{-\beta\delta}}e_{k_G}\widetilde{L}_{(h-1)(n-2)+s} + \frac{1}{1+e^{-\beta\delta}}e_{k_G}\widetilde{L}_{(h-1)(n-2)+s-1})\|_\infty$$

$$+ \|(\frac{e^{-\beta\delta}}{1+e^{-\beta\delta}}e_{k_G}\widetilde{L}_{(h-1)(n-2)+s} + \frac{1}{1+e^{-\beta\delta}}e_{k_G}\widetilde{L}_{(h-1)(n-2)+s-1}) - (w^\top X_i + t)e_{k_G}\|_\infty$$
$$\text{(By triangle inequality)}$$

$$\leq \frac{b-a}{p} + (H-2)m_v\epsilon_3 + \|V\|_\infty(\epsilon_4 + \epsilon_5)$$

$$\leq \frac{b-a}{H(n-2)} + (H-2)\max\{|a|,|b|\}\epsilon_3 + \max\{|a|,|b|\}(\epsilon_4 + \epsilon_5)$$

Setting $\epsilon_3, \epsilon_4, \epsilon_5$ to be

$$\epsilon_3 = \frac{\epsilon_0}{3(H-2)}$$
$$\epsilon_4 = \epsilon_5 = \frac{\epsilon_0}{3}$$

yields the final result.

This completes the proof. $\qquad\square$

## G.4 PROOF OF LEMMA 3.11

**Theorem G.13** (Lemma 3.11 Restated: Explicit Construction of ReLU Neural Network as Universal Approximator). *Let $f : \mathcal{X} \to \mathbb{R}$ be a continuous function defined on a compact domain $\mathcal{X} \subset \mathbb{R}^N$ for some $N \in \mathbb{N}_+$. For any $\epsilon > 0$, there exists a two-layer feed-forward neural network FFN with ReLU activation functions such that for all $x \in \mathcal{X}$*

$$\|\text{FFN}(x) - f(x)\|_{L_p} \leq \epsilon. \tag{G.32}$$

*Proof Sketch.* First, we discretize the input domain into a grid of points $G_D$. Around each grid point $v \in G_D$, we construct a ReLU-based bump function $R_v(x) = \sum \text{ReLU}$ that equals 1 within a small region around $v$ and rapidly decays to 0 outside this region. Next, we define the feedforward network (FFN) as $\sum_{v \in G_D} f(v) \cdot \text{ReLU}(R_v(x) - N + 1)$, allowing us to approximate $f(x)$ as a weighted sum of function values evaluated on grid points $v$ near $x$. This process yields a piecewise linear approximation of $f$. $\qquad\square$

*Proof.* We first quantizes the domain into a grid, builds localized bump functions using ReLU, construct the FFN to combines the piecewise approximation in a weighted sum to approximate $f$, and analyze the approximation error.

**Construction of Bump Function $R_v(\cdot)$.** Let $x = [x_1, x_2, \cdots, x_N] \in \mathbb{R}^N$. The compactness of $\mathcal{X}$ means it lies within an $N$-dimensional cube $[-B, B]^N$. Quantize this domain into a grid $G_D$ with granularity $g$

$$G_D = \left\{ \frac{-B(g-1)}{g}, \frac{-B(g-3)}{g}, \cdots, \frac{B(g-1)}{g} \right\}^N,$$

which results in $g^N$ grid points across all dimensions.

For each point on the grid $v \in G_D$, we define a local bump function denoted as $R_v(x)$

$$
\begin{aligned}
R_v(x) &= \sum_{i=1}^N \phi(x_i, v_i) \\
&= \sum_{i=1}^N \left[ \text{ReLU}(\frac{1}{\delta}(\frac{g(x_i - v_i)}{B} + 1)) - \text{ReLU}(\frac{1}{\delta}(\frac{g(x_i - v_i)}{B} + 1 - \delta)) \right. \\
&\qquad\qquad \left. + \text{ReLU}(\frac{1}{\delta}(-\frac{g(x_i - v_i)}{B} + 1)) - \text{ReLU}(\frac{1}{\delta}(-\frac{g(x_i - v_i)}{B} + 1 - \delta)) - 1 \right].
\end{aligned}
$$

The function $\phi(x_i, v_i)$ behaves as

$$\phi(x_i, v_i) == \begin{cases} 0, & |x_i - v_i| \geq \dfrac{B}{g}, \\ -\dfrac{g}{\delta B}|x_i - v_i| + \dfrac{1}{\delta}, & (1-\delta)\dfrac{B}{g} < |x_i - v_i| < \dfrac{B}{g}, \\ 1, & |x_i - v_i| \leq (1-\delta)\dfrac{B}{g}. \end{cases} \tag{G.33}$$

for every $i \in [N]$.

Now we discuss the behavior of the bump function $R_v(x)$ for difference distance between the grid point $v$ and $x$.

As shown in (G.33), the value of bump function depends on three different distance between $v$ and $x$. We formally define them as follow.

First define $G_v$ as the region centered at $v$ with radius $B/g$ in the $\ell_\infty$ norm

$$G_v := \{x \in [-B, B]^N : \|x - v\|_\infty \leq \frac{B}{g}\}.$$

Second define $P_v$ as the core region of $G_v$ where the bump function $R_v(x)$ is fully on (equal to $N$)

$$P_v := \{x \in [-B, B]^N : \|x - v\|_\infty \leq (1 - \delta)\frac{B}{g}\}. \tag{G.34}$$

Third we define the shell region of $G_v$ denoted as $G_v \setminus P_v$

$$G_v \setminus P_v := \{x \in [-B, B]^N : (1 - \delta)\frac{B}{g} \leq \|x - v\|_\infty \leq \frac{B}{g}\}.$$

Now we discuss the behavior of $R_v(x)$ under this three situations.

For $x \notin G_v$, at least one dimension of $x$ satisfies $|x_i - v_i| \geq B/g$ for some $i \in [N]$. By examining the definition $\phi(x_i, v_i)$ in (G.33), since $R_v(x) = \sum_{i=1}^N \phi(x_i, v_i)$ and at least one $\phi(x_i, v_i) = 0$, we have

$$R_v(x) = \sum_{i=1}^N \phi(x_i, v_i) \leq N - 1 \quad \text{for} \quad x \notin G_v. \tag{G.35}$$

For $x \in P_v$, each coordinates $x_i$ satisfies $|x_i - v_i| \leq (1-\delta)B/g$. By (G.33), this implies $\phi(x_i, v_i) = 1$ for all $i$, we derive

$$R_v(x) = \sum_{i=1}^N \phi(x_i, v_i) = N \times 1 = N \quad \text{for} \quad x \in P_v. \tag{G.36}$$

For $x \in G_v \setminus P_v$, by (G.33), the corresponding $\phi(x_i, v_i) \leq 1$. Thus

$$R_v(x) \in [N - 1, N) \quad \text{for} \quad x \in G_v \setminus P_v.$$

Until now we finish the construction of $R_v(\cdot)$ and analysis its behavious. Next we move to the construction of FFN to approximate the target function $f$.

**Construction of FFN.** Following the above discussion, we construct the FFN to be:

$$\text{FFN}(x) = \sum_{v \in G_D} f(v) \cdot \text{ReLU}(R_v(x) - N + 1). \tag{G.37}$$

The behavior of $\text{ReLU}(R_v(x) - N + 1)$ is

$$\text{ReLU}(R_v(x) - N + 1) = \begin{cases} 0, & x \notin G_v, \\ 1, & x \in P_v, \\ \text{ReLU}(R_v(x) - N + 1), & x \in G_v \setminus P_v. \end{cases}$$

By this construction, the FFN approximate $f(x)$ by weighted sum over the grid points $v$ such that $x \in G_v$.

Now we move to analysis the approximation error of the constructed FFN.

**Approximation Error Analysis.** To approximate the continuous function $f$, we introduce the region $\mathcal{P} := \bigcup_{v \in G_D} P_v$.

We also denote $\mu$ as the Lebesgue measure in $N$-dimensional space for later use.

Using the uniform continuity of $f$ and the properties of the constructed FFN, we analyze the $L_p$-norm error by partitioning the input domain into $\mathcal{P}$ and its complement

$$\|\text{FFN}(x) - f(x)\|_{L_p} = \left( \int_{[-B,B]^N} (\text{FFN}(x) - f(x))^p \, dx \right)^{\frac{1}{p}}$$

$$= \left( \int_{[-B,B]^N / \mathcal{P}} (\text{FFN}(x) - f(x)dx)^p + \int_{\mathcal{P}} (\text{FFN}(x) - f(x))^p \, dx \right)^{\frac{1}{p}}.$$

Now we discuss the two situations in the following paragraph, and conclude our proof.

- **Case 1:** $x \in \mathcal{P}$.

  For an $x \in \mathcal{P}$, let $v_x$ denote the unique grid point such that $x \in \mathbb{P}_v$. By (G.35) and (G.36) we have

  $$R_v(x) = N, \quad \text{if } v = v_x, \quad \text{and} \quad R_v(x) \leq N - 1, \quad \text{if } v \neq v_x.$$

  Hence

  $$\text{FFN}(x) = \sum_{v \in G_D} f(v) \cdot \text{ReLU}(R_v(x) - N + 1) \quad \text{(By (G.37))}$$
  $$= f(v_x)\text{ReLU}(R_{v_x}(x) - N + 1) \quad \text{(By } R_v(x) \leq N - 1, \quad \text{if } v \neq v_x)$$
  $$= f(v_x) \times 1 = f(v_x).$$

  Since the $\text{FFN}(x)$ collapse to $f(v_x)$ when $x \in \mathcal{P}$, the error $|\text{FFN}(x) - f(x)|$ becomes to approximate $f(x)$ by the function value evaluate on the closest grid point $f(v_x)$.

  Because $f$ is continuous on a closed region, it is bounded and uniformly continuous. Thus there exists a $\Delta > 0$ such that for any $x_1, x_2 \in \mathbb{R}^N$ satisfying $\|x_1 - x_2\|_\infty \leq \Delta$, the following hold

  $$|f(x_1) - f(x_2)| \leq \frac{\epsilon}{2(2B)^{\frac{N}{p}}\mu(\mathcal{P})^{\frac{1}{p}}},$$

  where the term $2(2B)^{\frac{N}{p}}\mu(\mathcal{P})^{\frac{1}{p}}$ is a constant to help us normalize the final error bound.

  Set $g$ to be large enough such that $2B/g < \Delta$, and since $\|v_x - v\|_\infty$ is smaller than the grid length $2B/G$, it also smaller than $\Delta$. This yields that for $x \in \mathcal{P}$

  $$|\text{FFN}(x) - f(x)| = |f(v_x) - f(x)| \leq \frac{\epsilon}{2(2B)^{\frac{N}{p}}\mu(\mathcal{P})^{\frac{1}{p}}}. \quad \text{(G.38)}$$

- **Case 2:** $x \notin \mathcal{P}$.

  Now we turn to analyse the approximation error outside $\mathcal{P}$.

  First we know that

  $$|\text{FFN}(x) - f(x)| \leq |\text{FFN}(x)| + |f(x)|$$
  $$\leq \|f\|_{L_\infty} + \|f\|_{L_\infty}$$
  $$= 2\|f\|_{L_\infty},$$

where the second inequality coming from

$$\|f\|_{L_\infty} = \sup_{x \in [-B,B]^N} \|f(x)\|_\infty \geq f(x),$$

and by (G.37), $f(v) \leq \|f(x)\|_{L_\infty}$, and also the design of bump function make sure given $x$, only the one grid point closet to $x$ contribute, hence $|\text{FFN}(x)| \leq \|f(x)\|_{L_\infty}$.

Hence the approximation error outside $\mathcal{P}$ become

$$\int_{[-B,B]^N/\mathcal{P}} (\text{FFN}(x) - f(x))^p \, dx \leq \int_{[-B,B]^N/\mathcal{P}} (2\|f\|_{L_\infty})^p dx \qquad (\text{G.39})$$

$$= (2\|f\|_{L_\infty})^p \cdot \mu([-B,B]^N/\mathcal{P}), \quad {\scriptstyle ((2\|f\|_{L_\infty})^p \text{ is a constant.})}$$

where $\mu([-B,B]^N/\mathcal{P})$ is the volume of how much of the entire domain isn't covered by $\mathcal{P}$. We calculate it as

$$\mu([-B,B]^N/\mathcal{P})$$
$$= \mu([-B,B]^N) - \mu(\mathcal{P})$$
$$= B^N - (1-\delta)^N B^N \qquad {\scriptstyle (\text{By (G.34) and } \mathcal{P} := \bigcup_{v \in G_D} P_v)}$$
$$= (1 - (1-\delta)^N)B^N \qquad {\scriptstyle (\text{By associative property})}$$
$$= (\delta N + \mathcal{O}(\delta^2))B^N \qquad {\scriptstyle (\text{By binomial expansion on } (1-\delta)^N = 1 - N\delta + \mathcal{O}(\delta^2))}$$
$$= \delta N B^N + \mathcal{O}(\delta^2).$$

For any $\epsilon_1 > 0$, we select a small enough $\delta$ such that $\mu([-B,B]^N/\mathcal{P}) \leq \epsilon_1$, thus we can make the volumn outside $\mathcal{P}$ as small as desired by choosing $\delta$ sufficiently small.

Thus the approximation outside $\mathcal{P}$ in (G.39) become

$$\int_{[-B,B]^N/\mathcal{P}} (\text{FFN}(x) - f(x))^p \, dx \leq (2\|f\|_{L_\infty})^p \cdot \mu([-B,B]^N/\mathcal{P}) \leq (2\|f\|_{L_\infty})^p \cdot \epsilon_1.$$

$$(\text{G.40})$$

We set $\epsilon_1$ to be

$$\epsilon_1 = \frac{\epsilon^p}{2^{p+1}\|f\|_{L_\infty}^p}.$$

for the normalization of the final error bound.

Finally we combine (G.38) and (G.40), for any $p \in N_+$, the total approximation is

$$\|\text{FFN}(x) - f(x)\|_{L_p} = \left( \int_{[-B,B]^N} (\text{FFN}(x) - f(x))^p \, dx \right)^{\frac{1}{p}}$$

$$= \left( \int_{[-B,B]^N/\mathcal{P}} (\text{FFN}(x) - f(x))^p \, dx + \int_{\mathcal{P}} (\text{FFN}(x) - f(x))^p \, dx \right)^{\frac{1}{p}}$$

$$\leq \left( \epsilon_1 (2\|f\|_{L_\infty})^p + (\frac{\epsilon}{2(2B)^{\frac{N}{p}} \mu(\mathcal{P})^{\frac{1}{p}}})^p \times (2B)^N \right)^{\frac{1}{p}} \qquad {\scriptstyle (\text{By (G.38) and (G.40)})}$$

$$\leq \left( \frac{\epsilon^p}{2} + \frac{\epsilon^p}{2} \right)^{\frac{1}{p}}$$

$$= \epsilon.$$

This completes the proof. □

### G.5 PROOF OF LEMMA 3.12

We first present a auxiliary lemma deduced from Theorem 3.4.

**Lemma G.14.** *Fix real numbers $a < b$, and let the truncation operator $\mathrm{Range}_{[a,b]}(\cdot)$ follow Definition 3.1. Let $w \in \mathbb{R}^d$ and $t \in \mathbb{R}$ be such that $\|w\|_\infty \leq R_w$ and $|t| \leq R_t$. For a precision parameter $p \in \mathbb{N}_+$ satisfying $p > n$, let $\epsilon = O\left(\frac{1}{p}\right)$. Then, for any $\epsilon > 0$, there exists a single-layer, single-head self-attention $\mathrm{Attn} : \mathbb{R}^{d\times p} \to \mathbb{R}^{d\times p}$, and an layer of linear connections $A : \mathbb{R}^{d\times n} \to \mathbb{R}^{d\times p}$ free of activation function such that*

$$\|\mathrm{Attn} \circ A(X) - [\underbrace{0}_{d_o \times n_0}, \sum_{i=1}^n \mathrm{Range}_{[a,b]}(w^\top x_i + t_i), \underbrace{0}_{d_o \times (p-1-n_0)}]\|_\infty < \epsilon.$$

*Here $N, n_0 \in N_+$ are any integer satisfying $n_0 \leq p - n$,*

*Proof.* By Corollary G.6, let $\epsilon = \epsilon'/n$ with $\epsilon' > 0$, there exists a $\mathrm{Attn}^*$ and a $A^*$ that satisfy

$$\|\mathrm{Attn}^* \circ A(X)^*_{:,i} - \mathrm{Range}_{[a,b]}(w_i^\top x_i + t_i)e_{\widetilde{k}_i}\|_\infty < \frac{\epsilon'}{n}. \tag{G.41}$$

By setting $t = 0$, $d_o = 1$ and $e_{\widetilde{k}_i} = 1$, we have

$$\|\mathrm{Attn}^* \circ A^*(X) - \underbrace{\mathrm{Range}_{[a,b]}(w^\top X)}_{1\times n}\|_\infty < \frac{\epsilon'}{n}. \tag{G.42}$$

Define $A_0(Z)$

$$A_0(Z) := Z + \left[ \frac{t_1}{\|w\|_2^2}w \quad \frac{t_2}{\|w\|_2^2}w \quad \cdots \quad \frac{t_n}{\|w\|_2^2}w \right], \tag{G.43}$$

to insert the bias terms $\{t_i\}_{i\in[n]}$ by combining with $A^*$ and define $A := A^* \circ A_0$. The denominator $\|w\|_2^2$ is because in (G.42) every token is multiplied by $w$, and $\frac{\langle w,w\rangle}{\|w\|_2^2} = 1$ make sure we get $t_i$. Since a linear transformation followed by another linear transformation is still a linear transformation, $A$ is a linear transformation.

Multiply the $W_O$ in $\mathrm{Attn}^*$ with $W_0$ defined as

$$W_0 := \underbrace{\left[ 0_{n\times n_0}, 1_n, 0_{n\times(p-1-n_0)} \right]}_{n\times p}. \tag{G.44}$$

And define $\mathrm{Attn}$ as

$$\mathrm{Attn}(Z) = \mathrm{Attn}^*(Z) \cdot W_0.$$

Since $W_O$ in $\mathrm{Attn}^*$ multiplied with $W_0$ still outputs a matrix, $\mathrm{Attn}$ is still an attention module.

Now we calculate the difference between $\mathrm{Attn} \circ A(X)$ with target output

$$\|\mathrm{Attn} \circ A(X) - [\underbrace{0}_{1\times n_0}, \sum_{i=1}^n \mathrm{Range}_{[a,b]}(w^\top x_i + t_i), \underbrace{0}_{1\times(p-1-n_0)}]\|_\infty$$

$$= \|\mathrm{Attn}^* \circ A^*(A_0(X))W_0 - [\underbrace{0}_{1\times n_0}, \sum_{i=1}^n \mathrm{Range}_{[a,b]}(w^\top x_i + t_i), \underbrace{0}_{1\times(p-1-n_0)}]\|_\infty$$

$$\text{(By definition of } \mathrm{Attn}^* \text{ and } A^*)$$

$$= \|\text{Attn}^* \circ A^*(A_0(X))W_0 - [\underbrace{0}_{1 \times n_0}, \sum_{i=1}^{n} \text{Range}_{[a,b]}(w^\top(x_i + \frac{t_i}{\|w\|_2^2}w)), \underbrace{0}_{1 \times (p-1-n_0)}]\|_\infty$$

$$= \|\text{Attn}^* \circ A^*(A_0(X))W_0 - [\underbrace{0}_{1 \times n_0}, \sum_{i=1}^{n} \text{Range}_{[a,b]}(w^\top A_0(X)_{:,i}), \underbrace{0}_{1 \times (p-1-n_0)}]\|_\infty \qquad \text{(By (G.43))}$$

$$= \|\text{Attn}^* \circ A^*(A_0(X))W_0 - [\underbrace{0}_{1 \times n_0}, \underbrace{\text{Range}_{[a,b]}(w^\top A_0(X))}_{1 \times n} \cdot 1_n, \underbrace{0}_{1 \times (p-1-n_0)}]\|_\infty$$

$$= \|\text{Attn}^* \circ A^*(A_0(X))W_0 - \underbrace{\text{Range}_{[a,b]}(w^\top X)}_{1 \times n} W_0\|_\infty \qquad \text{(By (G.44))}$$

$$\leq \|\text{Attn}^* \circ A^*(A_0(X)) - \underbrace{\text{Range}_{[a,b]}(w^\top X)}_{1 \times n}\|_1 \cdot \|W_0\|_\infty \qquad \text{(Since } \|EW_0\|_\infty \leq \|E\|_1\|W_0\|_\infty)$$

$$\leq \|\text{Attn}^* \circ A^*(A_0(X)) - \underbrace{\text{Range}_{[a,b]}(w^\top X)}_{1 \times n}\|_\infty \cdot n\|W_0\|_\infty \qquad \text{(Since } \|E\|_1\|W_0\|_\infty \leq n\|E\|_\infty\|W_0\|_\infty)$$

$$= \frac{\epsilon'}{n} \cdot n$$
$$= \epsilon'.$$

Because $\epsilon'$ is arbitrary, we reset the notation and denote it as $\epsilon$ for simplicity of presentation. This completes the proof. $\qquad\square$

Then we state our proof of Lemma 3.12.

**Lemma G.15** (Lemma 3.12 Restated: Sequence-to-Scalar Universal Approximation of Attention). *For any continuous function $f : \mathbb{R}^{d \times n} \to \mathbb{R}$ of compact support $\mathcal{X}$, and any $\epsilon > 0$, we prove that when composed with linear transformations, there exists a one layer multi-head attention $\text{Attn}_m$ stacked with one layer single-head attention $\text{Attn}_s$ composed with linear connections $A_1$ and $A_2$, such that*

$$\|f - \text{Attn}_s \circ A_2 \circ \text{Attn}_m \circ A_1\|_{L_p} \leq \epsilon$$

*Proof.* Let $X := [X_1, X_2, \cdots, X_n] \in \mathbb{R}^{d \times n}$ denotes our input. Without loss of generality, assume our inputs to come from $[-B, B]^{d \times n}$, $B \in \mathbb{R}_+$ is their bound in every dimension.

We discretize the input domain into a set of grid points $G_D$ defined as follow.

**Definition G.16** (Grid Centers). *We define $G_D$ as the set of all grid centers in $\mathbb{R}^{d \times n}$. The corresponding grids consists a quantization on $[-B, B]^{dn}$ of granularity $g$(meaning each dimension is equally partitioned to $g$ intervals)*

$$G_D = \{\frac{-B(g-1)}{g}, \frac{-B(g-3)}{g}, \cdots, \frac{B(g-1)}{g}\}^{d \times n},$$

*and $|G_D| = g^{dn}$, since each of the $dn$ entries can chosen from $g$ values.*

**Remark G.17.** *For a grid point $v \in G_D$, we denote its columns as $v_i (i \in [n])$. The entry on the $j$-th row is denoted as $v_{i,j}$. We write it out explicitly as*

$$v := [v_1, \quad v_2, \quad \cdots, v_n,] \in \mathbb{R}^{d \times n},$$
$$v_i = [v_{i,1} \quad v_{i,2} \quad \cdots \quad v_{i,d}]^\top, \quad i \in [n], j \in [d],$$

*where each $v_{i,j} \in \mathbb{R}$.*

For a $v \in G_D$, define the corresponding $R_v$ similar to that in Lemma 3.11

$$R_v(X) := \sum_{i=1}^{d} \sum_{j=1}^{n} [\text{ReLU}(\frac{1}{\delta}(\frac{g(X_{i,j} - v_{i,j})}{B} + 1)) - \text{ReLU}(\frac{1}{\delta}(\frac{g(X_{i,j} - v_{i,j})}{B} + 1 - \delta))$$

$$+ \text{ReLU}(\frac{1}{\delta}(-\frac{g(X_{i,j} - v_{i,j})}{B} + 1)) - \text{ReLU}(\frac{1}{\delta}(-\frac{g(X_{i,j} - v_{i,j})}{B} + 1 - \delta))],$$

for any $v \in G_D$. We eliminate the "-1" term in the definition of $R_v(x)$ in Lemma 3.11. Here $\delta$ is a coefficient we use to control the precision of our approximation in later process. We use $v_i$ to denote its $i$-th column of $v \in G_D$ for $i \in [n]$, where $v_i \in [-B, B]^d$. We also label every $v$ in $G_D$ as $v^{(j)}$, $j \in [g^{dn}]$ and denote this label as $l(v)$ for every $v$.

Next we show single-layer attention approximate $R_v(X)$.

From Lemma G.14, by setting $w = g/\delta B e_i$ and $t_k = -g/\delta B v_{i,k}^{(j)}$, $k \in [n]$ there exists a single-head attention $\text{Attn}_{v^{(j)},+}^{(i)}$ attached with a linear transformation $A_1$ that satisfies

$$\|\text{Attn}_{v^{(j)},+}^{(i)} \circ A_1(X) - \begin{bmatrix} 0_{j-1}^{\top} & \text{Range}_{[0,1]}(\frac{g(e_i^{\top} X_1 - v_{i,1})}{\delta B}) + \cdots + \text{Range}_{[0,1]}(\frac{g(e_i^{\top} X_n - v_{i,n})}{\delta B}) & 0_{|G_D|-j}^{\top} \end{bmatrix}\|_{\infty}$$

$$\leq \epsilon_0,$$

for any $\epsilon_0 > 0$.

Also from Theorem 3.4, there should also exist a single-head attention $\text{Attn}_{v^{(j)},-}^{(i)}$ attached with the same linear transformation $A_1$ that satisfies

$$\|\text{Attn}_{v^{(j)},-}^{(i)} \circ A_1(X) - \begin{bmatrix} 0_{j-1}^{\top} & \text{Range}_{[0,1]}(\frac{g(v_{i,1} - e_i^{\top} X_1)}{\delta B}) + \cdots + \text{Range}_{[0,1]}(\frac{g(v_{i,n} - e_i^{\top} X_1)}{\delta B}) & 0_{|G_D|-j}^{\top} \end{bmatrix}\|_{\infty}$$

$$\leq \epsilon_0.$$

Summing these two kinds of single head attention across $d$ we approximate $R_{v^{(j)}}$ in the following remark.

**Remark G.18.** *For every $v^{(j)} \in [G_D]$, the aggregation of all $\text{Attn}_{v^{(j)},+}^{(i)}$ and all $\text{Attn}_{v^{(j)},-}^{(i)}$ for $i \in [d]$ outputs*

$$\|\sum_{i=1}^{d} (\text{Attn}_{v^{(j)},-}^{(i)}(X) + \text{Attn}_{v^{(j)},+}^{(i)}(X)) - \begin{bmatrix} 0_{1 \times (j-1)} & R_{v^{(j)}}(X) & 0_{1 \times (|G_D|-j)} \end{bmatrix}\|_{\infty} \leq d\epsilon_0.$$

$$(G.45)$$

Since we must do this for all $v \in G_D$, we use multiple heads in parallel. We construct the multi-head attention to be

$$\text{Attn}_m(X) := \sum_{j=1}^{|G_D|} \sum_{i=1}^{d} (\text{Attn}_{v^{(j)},-}^{(i)}(X) + \text{Attn}_{v^{(j)},+}^{(i)}(X)). \qquad (G.46)$$

Then by (G.45), the output of $\text{Attn}_m \circ A(X)$ satisfies

$$\|\text{Attn}_m \circ A(X) - \sum_{j=1}^{|G_D|} \underbrace{\begin{bmatrix} 0_{1 \times (j-1)} & R_{v^{(j)}}(X) & 0_{1 \times (|G_D|-j)} \end{bmatrix}}_{1 \times |G_D|}\|_{\infty} \leq d\epsilon_0.$$

Thus

$$\|\text{Attn}_m \circ A(X) - [R_{v^{(1)}}(X) \quad R_{v^{(2)}}(X) \quad \cdots \quad R_{v^{(|G_D|)}}(X)]\|_\infty \le d\epsilon_0 \tag{G.47}$$

where as previously denoted, $|G_D|$ is the total number of all grid centers.

Now we construct the second layer of attention $\text{Attn}_s$ to pick the largest $R_{v^{(j)}}(X)$, and use a linear layer $A_2$ to encode the function value.

First, we construct $A_2$ to be

$$A_2(Z) := \begin{bmatrix} 0 & 0 & \cdots & 0 \\ f(v^{(1)}) & f(v^{(2)}) & \cdots & f(v^{(|G_D|)}) \end{bmatrix} + Z. \tag{G.48}$$

By (G.47), $A_2$ connected after $\text{Attn} \circ A_1$ has an output satisfying

$$\|A_2 \circ \text{Attn} \circ A_1(X) - \begin{bmatrix} R_{v^{(1)}}(X) & R_{v^{(2)}}(X) & \cdots & R_{v^{(|G_D|)}}(X) \\ f(v^{(1)}) & f(v^{(2)}) & \cdots & f(v^{(|G_D|)}) \end{bmatrix} \|_\infty \le d\epsilon_0. \tag{G.49}$$

For $\text{Attn}_s$, we construct a single-head attention $\text{Attn}_s$ each weight matrix to pick the desired row in $Z$

$$\text{Attn}_s(Z) := \underbrace{[0 \quad 1]}_{1\times 2} Z \text{Softmax}_\beta((\underbrace{[1 \quad 0]}_{1\times 2} Z)^\top \underbrace{[1 \quad 0]}_{1\times 2} Z) \mathbb{1}_{|G_D|}, \tag{G.50}$$

where $\beta$ is a parameter we use to control the precision.

Now we claim that the construct attention layer satisfies the following

$$\|\text{Attn}_s \circ A_2 \circ \text{Attn}_m \circ A_1(X) - [f(v^{(1)}) \quad f(v^{(2)}) \quad \cdots \quad f(v^{(|G_D|)})] \text{Softmax}_\beta(M)\|_\infty \le d\epsilon_0, \tag{G.51}$$

which we derive by plugging (G.48) , (G.49) and (G.50) into $\text{Attn}_s \circ A_2 \circ \text{Attn}_m \circ A_1$

$$\underbrace{[0 \quad 1]}_{1\times 2} \begin{bmatrix} R_{v^{(1)}}(X) & R_{v^{(2)}}(X) & \cdots & R_{v^{(|G_D|)}}(X) \\ f(v^{(1)}) & f(v^{(2)}) & \cdots & f(v^{(|G_D|)}) \end{bmatrix}$$

$$\cdot \text{Softmax}_\beta((\underbrace{[1 \quad 0]}_{1\times 2} \begin{bmatrix} R_{v^{(1)}}(X) & \cdots & R_{v^{(|G_D|)}}(X) \\ f(v^{(1)}) & \cdots & f(v^{(|G_D|)}) \end{bmatrix})^\top \underbrace{[1 \quad 0]}_{1\times 2} \begin{bmatrix} R_{v^{(1)}}(X) & \cdots & R_{v^{(|G_D|)}}(X) \\ f(v^{(1)}) & \cdots & f(v^{(|G_D|)}) \end{bmatrix})$$

$$= [f(v^{(1)}) \quad \cdots \quad f(v^{(|G_D|)})] \text{Softmax}_\beta([R_{v^{(1)}}(X) \quad \cdots \quad R_{v^{(|G_D|)}}(X)]^\top [R_{v^{(1)}}(X) \quad \cdots \quad R_{v^{(|G_D|)}}(X)]) \tag{G.52}$$

$$= [f(v^{(1)}) \quad f(v^{(2)}) \quad \cdots \quad f(v^{(|G_D|)})] \text{Softmax}_\beta(M),$$

where $M$ is given as

$$M_{i,j} := R_{v^{(i)}}(X) R_{v^{(j)}}(X), \quad i,j \in [|G_d|].$$

Until now we complete the proof of showing two attention layers with linear transform approximate $[f(v^{(1)}) \quad f(v^{(2)}) \quad \cdots \quad f(v^{(|G_D|)})] \text{Softmax}_\beta(M)$.

To further calculate the approximation error $\text{Softmax}_\beta(M)$, we need to review some key attributes of $R_v(X)$. Hence we recall some results from the proof of Lemma 3.11.

Here we use the following attribute of $R_v(X)$

- $R_v(X) \in \{dn, dn+1, \cdots, 2dn\}$ on $[-B, B]^{d \times n}$ except for a region no larger than $1 - (1-\delta)^{dn}$. Here $\delta > 0$ is an self-selected coefficient we defined in the construction of $A_1$ and $\text{Attn}_m$. Thus by setting $\delta$ to be sufficiently large, the region of exception can be arbitrarily small.

- Except for an arbitrarily small region, the maximal $R_{v^{(i)}}$ equals to $2dn$ and the second largest equals to $2dn - 1$.

Since Softmax is a column-wise operation, we calculate $\text{Softmax}_\beta(M)$ by column

$$
\text{Softmax}_\beta(M)_{:,i} = \text{Softmax}_\beta\left(\begin{bmatrix} R_{v^{(i)}}(X)R_{v^{(1)}}(X) \\ R_{v^{(i)}}(X)R_{v^{(2)}}(X) \\ \vdots \\ R_{v^{(i)}}(X)R_{v^{(|G_D|)}}(X) \end{bmatrix}\right).
$$

Then by Lemma F.1, when $\beta$ is sufficiently large

$$
\|\text{Softmax}_\beta(M)_{:,} - e_k\|_\infty \leq \epsilon_1,
$$

for any $\epsilon_1 > 0$. Here $k$ is defined as

$$
k := \underset{k \in [|G_D|]}{\text{argmax}}(R_{v^{(k)}}(X)).
$$

Thus

$$
\| \begin{bmatrix} f(v^{(1)}) & f(v^{(2)}) & \cdots & f(v^{(|G_D|)}) \end{bmatrix} \text{Softmax}_\beta(M) - f(v^{(k)})\|_\infty \leq \epsilon_1 \cdot \|f\|_{L_\infty}.
$$

This excludes an arbitrarily small region where at least two entries in $\text{Softmax}_\beta(M)_{:,i}$ are identical. We denote this region as $\Delta_0$.

Combine this with (G.51) yields

$$
\|\text{Attn}_s \circ A_2 \circ \text{Attn}_m \circ A_1(X) - f(v^{(k)})\|_\infty \leq \epsilon_1 \cdot \|f\|_{L_\infty} + d\epsilon_0. \tag{G.53}
$$

Finally we calculate the approximation error including the grid point approximation. For simplicity we denote the $\text{Attn}_s \circ A_2 \circ \text{Attn}_m \circ A_1(X) := \mathcal{N}(X)$. Then, we have

$$
\|f(X) - \mathcal{N}(X)\|_{L_p} = \left(\int_{X \in [-B,B]^{d \times n}} \|f(X) - \mathcal{N}(X)\|_p^p \, dX\right)^{1/p}.
$$

We split the domain into two parts as in Appendix G.4

$$
\|f(X) - \mathcal{N}(X)\|_{L_p}
$$
$$
= (\int_{X \in [-B,B]^{d \times n}} \|f(X) - f(v^{(k)})\|_p^p \, dX + \int_{X \in [-B,B]^{d \times n}} \|f(v^{(k)}) - \mathcal{N}(X)\|_p^p \, dX)^{\frac{1}{p}}
$$
$$
\text{(By triangle inequality)}
$$
$$
\leq (\int_{X \in [-B,B]^{d \times n}} \|f(X) - f(v^{(k)})\|_p^p \, dX + \int_{X \in [-B,B]^{d \times n} \setminus \Delta_0} \|f(v^{(k)}) - \mathcal{N}(X)\|_p^p \, dX
$$
$$
+ \int_{X \in \Delta_0} \|f(v^{(k)}) - \mathcal{N}(X)\|_p^p \, dX)^{\frac{1}{p}} \qquad \text{(Seperate } \Delta_0 \text{ out)}
$$
$$
\leq \left(\varepsilon + (2B)^{dn}(\epsilon_1 \cdot \|f\|_{L_\infty} + d\epsilon_0) + \mu(\Delta_0) \cdot (2dnM_{fN})^p\right)^{\frac{1}{p}}
$$
$$
\leq \varepsilon^{\frac{1}{p}} + 2B^{\frac{dn}{p}}(\epsilon_1 \cdot \|f\|_{L_\infty} + d\epsilon_0)^{\frac{1}{p}} + 2(\mu(\Delta_0))^{\frac{1}{p}} dnM_{fN},
$$

where $\mu$ denotes the Lebesgue measure on $\mathbb{R}^{d \times n}$. Here, the $\varepsilon$ in the third row inequality can be arbitrarily small when $g$ is large enough, according to the discussion to derive (G.38). The error $\epsilon_1$ is the softmax approximation error, and $\epsilon_0$ is coming from (G.45). The term $M_{fN}$ comes from

$$\|f(X) - \mathcal{N}(X)\|_{i,j} \leq \|f(X)\|_{L_\infty} + \|\mathcal{N}(X)\|_{L_\infty} \leq 2M_{fN},$$

where $M_{fN}$ is a mutual upper-bound of the value of $f$ and $\mathcal{N}$. Because both $f$ and $\mathcal{N}$ are continuous on a compact support, they are bounded in $\infty$ norm and hence have a mutual upper-bound.

Configure $\Delta_0$, $\epsilon_0$ and set $g$ large enough

$$\varepsilon^{\frac{1}{p}} + 2B^{\frac{dn}{p}}(\epsilon_1 \cdot \|f\|_{L_\infty} + d\epsilon_0)^{\frac{1}{p}} + 2(\mu(\Delta_0))^{\frac{1}{p}} dn\|f\|_\infty \leq \epsilon.$$

This completes the proof. $\qquad\square$

### G.6    PROOF OF LEMMA 3.13

**Lemma G.19** (Lemma 3.13 Restated: Single-Layer Attention Version of Lemma 3.12). *For any continuous function $f : \mathbb{R}^{d \times n} \to \mathbb{R}$ of compact support $\mathcal{X}$, and any $\epsilon > 0$, we prove that when attached with linear transformations, there exists a one layer multi-head attention $\mathrm{Attn}_m$ followed by a* Softmax *function and attached with linear connections $A_1$ and $A_2$, such that*

$$\|f - A_2 \circ \mathsf{Softmax} \circ \mathrm{Attn}_m \circ A_1\|_{L_p} \leq \epsilon.$$

*Proof.* Starting from (G.47) we have

$$\|\mathrm{Attn}_m \circ A(X) - [R_{v^{(1)}} \quad R_{v^{(2)}} \quad \cdots \quad R_{v^{(|G_D|)}}]\|_\infty \leq d\epsilon_0. \tag{G.54}$$

Applying $\mathsf{Softmax}_\beta$ to (G.54) yields

$$\|\mathsf{Softmax}_\beta(\mathrm{Attn}_m \circ A(X)) - \mathsf{Softmax}_\beta([R_{v^{(1)}} \quad R_{v^{(2)}} \quad \cdots \quad R_{v^{(|G_D|)}}])\|_\infty \leq d\epsilon_0. \tag{G.55}$$

Define a linear map $A_2$ as

$$A_2(Z) := Z[f(v_1), f(v_2), \cdots, f(v_{|G_D|})]^\top,$$

and apply $A_2$ on (G.55) we have

$$\|A_2(\mathsf{Softmax}_\beta(\mathrm{Attn}_m \circ A(X))) - A_2(\mathsf{Softmax}_\beta([R_{v^{(1)}} \quad R_{v^{(2)}} \quad \cdots \quad R_{v^{(|G_D|)}}]))\|_\infty$$
$$= \|\mathsf{Softmax}_\beta(\mathrm{Attn}_m \circ A(X)) [f(v_1) \quad f(v_2) \quad \cdots \quad f(v_{|G_D|}]^\top$$
$$- \mathsf{Softmax}_\beta([R_{v^{(1)}} \quad R_{v^{(2)}} \quad \cdots \quad R_{v^{(|G_D|)}}]) [f(v_1) \quad f(v_2) \quad \cdots \quad f(v_{|G_D|}]^\top\|_\infty$$
$$\leq B_0 d\epsilon_0,$$

where $B_0$ denotes $\|f\|_{L_\infty}$. It is bounded in $\infty$ norm since $f$ is continuous on a compact domain. Thus

$$\|A_2(\mathsf{Softmax}_\beta(\mathrm{Attn}_m \circ A(X))) - \mathsf{Softmax}_\beta([R_{v^{(1)}} \quad \cdots \quad R_{v^{(|G_D|)}}]) [f(v^{(1)}) \quad \cdots \quad f(v^{|G_D|}]^\top\|_\infty$$
$$\leq B_0 d\epsilon_0.$$

When $\beta$ is sufficiently large, except for an arbitrarily small region $\Delta$ of measure $\mu(\Delta)$ (the region in which $2 R_v^{(i)}$ are nearly identical), by Lemma F.1, the following equation

$$\mathsf{Softmax}_\beta([R_{v^{(1)}} \quad R_{v^{(2)}} \quad \cdots \quad R_{v^{(|G_D|)}}]) [f(v^{(1)}) \quad \cdots \quad f(v^{|G_D|}]^\top,$$

approximates

$$e^{\top}_{\mathrm{argmax}_{i \in [|G_D|]} R_{v^{(i)}}} \cdot \left[ f(v^{(1)}) \quad f(v^{(2)}) \quad \cdots \quad f(v^{|G_D|}) \right]^{\top} = f(v^{\mathrm{argmax}_i R_{v^{(i)}}}),$$

by an arbitrarily small error, we set this to be $\epsilon_1$.

Since the maximal $R_{v^{(i)}}(X)$ corresponds to the $v^{(i)}$ whose corresponding grid encapsulates $X$.

Thus $X$ differs from $v^{(i)}$ on each dimension by a difference no larger than the grid length $B/g$.

When $g$ is sufficiently large, $\|X - v^{(i)}\|_{\infty}$ is sufficiently small such that by the uniform continuity of $f$, we have

$$\|f(X) - f(v^{(i)})\|_{\infty} \le \epsilon_1.$$

Thus

$$\begin{aligned}
&\|f(X) - A_2(\mathsf{Softmax}_{\beta}(\mathrm{Attn}_m \circ A(X)))\|_{\infty} \\
&\le \|f(X) - f(v^{(i)})\|_{\infty} + \|f(v^{(i)}) - A_2(\mathsf{Softmax}_{\beta}(\mathrm{Attn}_m \circ A(X)))\|_{\infty} \quad \text{(By triangle ineqality)} \\
&\le 2\epsilon_1,
\end{aligned}$$

where the second line is by the triangle inequality.

This yields that

$$\begin{aligned}
&\|f - A_2 \circ \mathsf{Softmax}_{\beta} \circ \mathrm{Attn}_m \circ A\|_{L_p} \\
&\le \Big( \int_{X \in [-B,B]^{d \times n} \backslash \Delta} \|f(X) - A_2(\mathsf{Softmax}_{\beta}(\mathrm{Attn}_m \circ A(X)))\|_p^p \, \mathrm{d}X \\
&\quad + \int_{X \in \Delta} \|f(X) - A_2(\mathsf{Softmax}_{\beta}(\mathrm{Attn}_m \circ A(X)))\|_p^p \, \mathrm{d}X \Big)^{\frac{1}{p}} \\
&\le \big( (2B)^{dn} \cdot 2\epsilon_1 + dn \cdot (2\|f\|_{\infty})^p \mu(\Delta) \big)^{\frac{1}{p}}.
\end{aligned}$$

Set $\epsilon_1$ and $\mu(\Delta)$ to satisfy that

$$\big( (2B)^{dn} \cdot 2\epsilon_1 + dn \cdot (2\|f\|_{\infty})^p \mu(\Delta) \big)^{\frac{1}{p}} \le \epsilon.$$

We have

$$\|f - A_2 \circ \mathsf{Softmax}_{\beta} \circ \mathrm{Attn}_m \circ A\|_{L_p} \le \epsilon.$$

This completes the proof. $\qquad\qquad\qquad\qquad\qquad\qquad\qquad\qquad\qquad\qquad\qquad \square$

### G.7 PROOF OF THEOREM 3.14

**Theorem G.20** (Theorem 3.14 Restated: Sequence-to-Sequence Approximation of Universal Approximation of Attention). *For any continuous function $f : \mathbb{R}^{d \times n} \to \mathbb{R}^{d \times n}$ of compact support $\mathcal{X}$, and any $\epsilon > 0$, we prove that when attached with linear transformations, there exists a two layer multi-head attention $\mathrm{Attn}_m$ stacked with one layer multi-head attention $\mathrm{Attn}_m$, attatched with linear connection $A_1$ and $A_2$, such that*

$$\|f - \mathrm{Attn}_m^{(2)} \circ A_2 \circ \mathrm{Attn}_m^{(1)} \circ A_1\|_{L_p} \le \epsilon.$$

*Proof.* Given $f : \mathbb{R}^{d \times n} \to \mathbb{R}^{d \times n}$, we decompose $f$ into $f_{ij} : \mathbb{R}^{d \times n} \to \mathbb{R}$, where $i \in [d], j \in [n]$ denote the entry on the $i$-th row and the $j$-th column $f$. Thus

$$f(X) = \begin{bmatrix} f_{11}(X) & \cdots & f_{1n}(X) \\ \vdots & \ddots & \vdots \\ f_{d1}(X) & \cdots & f_{dn}(X) \end{bmatrix}.$$

By Lemma 3.12, we approximate each $f_{ij}$ by a multi-head attention stacked with a single-head attention in the following way

$$\|f_{ij}(X) - \mathrm{Attn}_s^{ij} \circ A_2 \circ \mathrm{Attn}_m \circ A_1(X)\|_p \le \epsilon_{\text{scaler}}. \tag{G.56}$$

Recall that the goal of multi-head attention $\mathrm{Attn}_m$ in Lemma 3.12 is to approximate the bump function $R_{v^{(j)}}$ on all the grid point $v^{(j)} \in G_D$ , hence it's irrelevant to the function $f_{ij}$ we aim to approximate. The follow-up single-head attention $\mathrm{Attn}_s^{ij}$ is responsible to map out the function output, hence depends on the $i, j$.

One thing need to modify is the definition of $A_2$ in (G.48), we need to append $dn$ rows of different function value for $f_{ij}$

$$A_2(Z) := \begin{bmatrix} 0 & 0 & \cdots & 0 \\ f_{11}(v^{(1)}) & f_{11}(v^{(2)}) & \cdots & f_{11}(v^{(|G_D|)}) \\ f_{12}(v^{(1)}) & f_{12}(v^{(2)}) & \cdots & f_{12}(v^{(|G_D|)}) \\ \vdots & \vdots & \ddots & \vdots \\ f_{dn}(v^{(1)}) & f_{dn}(v^{(2)}) & \cdots & f_{dn}(v^{(|G_D|)}) \end{bmatrix} + Z.$$

Also the second single layer attention need slight modification to pick out function $f_{ij}$ among $dn$ rows

$$\mathrm{Attn}_s^{ij}(Z) = \underbrace{[0 \quad e_k]}_{1 \times (1+dn)} Z \mathrm{Softmax}((R \underbrace{[1 \quad 0_{1 \times dn}]}_{1 \times (1+dn)} Z)^\top \underbrace{[1 \quad 0_{1 \times dn}]}_{1 \times (1+dn)} Z) \mathbb{1}_{|G_D|}, \tag{G.57}$$

where $k = (i-1)n + j$, and one-hot vector $e_k \in \mathbb{R}^{dn}$ is used to pick out the corresponding row.

This modification doesn't change the output after (G.52), hence the approximation error remain the same.

What remain is to combine this $d \times n$ approximations into one output matrix. To combine the scalar approximations back into a $\mathbb{R}^{d \times n}$ map, we use the matrices $E^{ij} \in \mathbb{R}^{d \times n}$ whose entries is zero everywhere except for value 1 on the $i$-th row and the $j$-th column.

Combining $E^{ij}$ with $\mathrm{Attn}_s^{ij}$ we construct a new one-layer multi-head attention $\mathrm{Attn}_m^{(2)}$ defined as

$$\mathrm{Attn}_m^{(2)} = \sum_{i \in [d], j \in [n]} E^{ij} \mathrm{Attn}_s^{ij}, \tag{G.58}$$

Then by stacking with the same $\text{Attn}_m$ denoted as $\text{Attn}_m^{(1)}$, we construct the sequence-to-sequence approximation to be $\text{Attn}_m^{(2)} \circ \text{Attn}_m^{(1)}$.

The error of approximation $\|f(X) - \text{Attn}_m^{(2)} \circ A_2 \circ \text{Attn}_m^{(1)} \circ A_1(X)\|_p$, when requiring $\epsilon_{\text{scaler}} = \epsilon/((dn)^{1/p})$ is

$$
\begin{aligned}
&\|f(X) - \text{Attn}_m^{(2)} \circ A_2 \circ \text{Attn}_m^{(1)} \circ A_1(X)\|_p \\
&= \Big( \sum_{i\in[d],j\in[n]} |f_{ij}(X) - (\text{Attn}_m^{(2)} \circ A_2 \circ \text{Attn}_m^{(1)} \circ A_1(X))_{ij}|^p \Big)^{\frac{1}{p}} \\
&= \Big( \sum_{i\in[d],j\in[n]} |f_{ij}(X) - \text{Attn}_s^{ij} \circ A_2 \circ \text{Attn}_m^{(1)} \circ A_1(X)|^p \Big)^{\frac{1}{p}} &\text{(By (G.58))} \\
&\leq (dn\epsilon_{\text{scaler}}{}^p)^{\frac{1}{p}} &\text{(By (G.56))} \\
&= (dn)^{\frac{1}{p}} \epsilon_{\text{scaler}} \\
&= \epsilon. &\text{(By } \epsilon_{\text{scaler}} = \frac{\epsilon}{(dn)^{1/p}} \text{)}
\end{aligned}
$$

For the case of using one attention layer following by a softmax function, by Lemma 3.13 we know

$$
\|f_{ij}(X) - A_2 \circ \text{Softmax} \circ \text{Attn}_m \circ A_1(X)\|_p \leq \epsilon_{\text{scaler}}.
$$

Again by modifying $A_2$ to $A_2^{ij}$

$$
A_2^{ij}(Z) := E^{ij} Z \underbrace{\begin{bmatrix} f_{11}(v^{(1)}) & f_{12}(v^{(1)}) & \cdots & f_{dn}(v^{(1)}) \\ f_{11}(v^{(2)}) & f_{12}(v^{(2)}) & \cdots & f_{dn}(v^{(2)}) \\ \vdots & \vdots & \ddots & \vdots \\ f_{11}(v^{(|G_D|)}) & f_{12}(v^{(|G_D|)}) & \cdots & f_{dn}(v^{(|G_D|)}) \end{bmatrix}}_{v^{|G_D| \times dn}} e_{(i-1)n+j},
$$

and since the modification doesn't change the error analysis in Appendix G.6 , we have

$$
\|f_{ij}(X) - A_2^{ij} \circ \text{Softmax} \circ \text{Attn}_m \circ A_1(X)\|_p \leq \epsilon_{\text{scaler}}.
$$

We have

$$
\begin{aligned}
&\|f(X) - \sum_{i\in[d],j\in[n]} A_2^{ij} \circ \text{Softmax} \circ \text{Attn}_m^{(1)} \circ A_1(X)\|_p \\
&\leq \Big( \sum_{i\in[d],j\in[n]} |f_{ij}(X) - A_2^{ij} \circ \text{Softmax} \circ \text{Attn}_m \circ A_1(X)|^p \Big)^{\frac{1}{p}} \\
&\leq (dn\epsilon_{\text{scaler}}{}^p)^{\frac{1}{p}} &\text{(By (G.56))} \\
&= (dn)^{\frac{1}{p}} \epsilon_{\text{scaler}} \\
&= \epsilon. &\text{(By } \epsilon_{\text{scaler}} = \frac{\epsilon}{(dn)^{1/p}} \text{)}
\end{aligned}
$$

This completes the proof. $\qquad\square$

## G.8 PROOF OF THEOREM F.2

In this section, we prove the sequence-to-sequence universal approximation of a two-layer attention mechanism in the $\ell_\infty$ norm.

We first introduce a lemma modified from (Pinkus, 1999, Theorem 3.1) and show the universal approximation theory of one layer feed-forward neural network

**Lemma G.21** (Theorem 3.1 from Pinkus (1999), Universal Approximation Of One Layer Feed-Forward Neural Network). *Let $\sigma : \mathbb{R} \to \mathbb{R}$ be a continuous function. The space of functions defined by single-layer neural networks*

$$M(\sigma) = \left\{ g(x) = \sum_{i=1}^{N} \eta_i \sigma(w_i \cdot x + t_i) \mid N \in \mathbb{N}, w_i \in \mathbb{R}^d, \eta_i, t_i \in \mathbb{R} \right\},$$

*is dense in $C(K)$. Here, $C(K)$ represents the space of continuous functions on any compact domain $K \subset \mathbb{R}^d$, if and only if $\sigma$ is not a polynomial. In other words, for any continuous function $f \in C(K)$ and any small error tolerance $\varepsilon > 0$, there exists a function $g \in M(\sigma)$ such that the maximum difference between $f$ and $g$ over $K$ is less than $\varepsilon$ (i.e., $\|f - g\|_\infty < \varepsilon$).*

The ReLU activation function, $\sigma(x) = \max(0, x)$, satisfies the conditions of above lemma because it is continuous and not a polynomial. Therefore, single-layer neural networks with ReLU activations form a dense subset of $C(K)$, meaning that they approximate any continuous function on a compact set $K$ to arbitrary precision in the infinity norm.

Next we introduce a simplified version of Theorem 3.4, where the only difference is we force the mapping function $G$ maps to a constant $r$ instead of $\widetilde{k}_i \in [d_o]$ no matter what input $k_i$ is.

**Theorem G.22** (Single-Head Attention Approximates Many Truncated Linear Models). *Fix real $a < b$, and let $\mathrm{Range}_{[a,b]}(\cdot)$ be the truncation operator from Definition 3.1. For a precision parameter $p > n$ with $\epsilon = O(1/p)$, there exists a single-layer, single-head self-attention $\mathrm{Attn}$ with a linear transformation $A : \mathbb{R}^{d \times n} \to \mathbb{R}^{(2d+d_o+2) \times p}$, such that $\mathrm{Attn} \circ A : \mathbb{R}^{d \times n} \to \mathbb{R}^{d_o \times n}$ satisfies, for any $i \in [n]$,*

$$\|\mathrm{Attn} \circ A(X)_{:,i} - \mathrm{Range}_{[a,b]}(w_i^\top x_i + t_i)e_r\|_\infty$$
$$\leq \underbrace{\max\{|a|, |b|\} \cdot \epsilon_0}_{\textit{finite-}\beta \textit{ softmax error}} + \underbrace{(b - a)/p}_{\textit{interpolation error}}, \quad \textit{for} \quad i \in [n],$$

*where $e_r$ is a one-hot vector with a value of $1$ at the $r$-th index and $0$ elsewhere, and $r \in [d_o]$ is defined as*

$$k_i := \operatorname*{argmin}_{k \in \{0,1,\ldots,p-1\}} (-2x_i^\top w_i - 2t_i + \widetilde{L}_0 + \widetilde{L}_k) \cdot k$$
$$r := G(k_i),$$

*where $r$ is any positive integer.*

Now we are ready to prove attention approximate sequence-to-sequence function with a bounded error in the infinity norm.

**Theorem G.23** (Theorem F.2 Restated: Sequence-to-Sequence Approximation in Infinity Norm). *For any continuous function $f : \mathbb{R}^{d \times n} \to \mathbb{R}^{d \times n}$ of compact support $\mathcal{X}$, and any $\epsilon > 0$, we prove that when attached with linear transformations, there exists a one layer multi-head attention $\mathrm{Attn}_m$ stacked with one layer multi-head attention $\mathrm{Attn}_m$, such that when the precision parameter in Theorem G.22 is $p = \Omega(n^{5/2})$, for any $X \in \mathcal{X}$*

$$\|f(X) - \mathrm{Attn}_m^{(2)} \circ A \circ \mathrm{Attn}_m^{(1)} \circ A(X)\|_\infty \leq \epsilon.$$

*Proof.* Given $f : \mathbb{R}^{d \times n} \to \mathbb{R}^{d \times n}$, we decompose $f$ into $f_{ij} : \mathbb{R}^{d \times n} \to \mathbb{R}$, where $i \in [d], j \in [n]$ denote the entry on the $i$-th row and the $j$-th column $f$. Thus

$$f(X) = \begin{bmatrix} f_{11}(X) & \cdots & f_{1n}(X) \\ \vdots & \ddots & \vdots \\ f_{d1}(X) & \cdots & f_{dn}(X) \end{bmatrix}.$$

We aim to construct attention layer to approximate function $f_{ij}$ in the form of

$$\text{FFN}(\text{vec}(X)) = \sum_{i=1}^{N} \eta_i \text{ReLU}(w_i^\top \text{vec}(X) + t_i),$$

where $\text{vec}(X) \in \mathbb{R}^{dn}$ is the flatten operation.

We achieve this by modifying the proof of Theorem G.22 and sum over the multi-head attention output to make each entry of the multi-head attention output is in the form of

$$\sum_{i=1}^{N} \eta_i \text{ReLU}(w_i^\top X + t_i).$$

First, we set the mapping function $G$ to map each $k_i$ to the same row $r$, that is $G(k_i) = r$. Thus the value matrix $V$ become

$$V = \begin{bmatrix} 0_{(r-1) \times p} \\ \widetilde{L}^\top \\ 0_{(d_0 - r) \times p} \end{bmatrix} \in \mathbb{R}^{d_0 \times p}, \tag{G.59}$$

Then we modify the $W_O$ matrix from having an identity matrix on the upper $n \times n$ block, to having a $n \times n$ matrix with the $c$-th column is $\eta \in \mathbb{R}^n$ entry and other entry is $0$

$$W_O = \begin{bmatrix} \eta e_c^\top \\ 0_{(p-n) \times n} \end{bmatrix} \in \mathbb{R}^{p \times n}, \tag{G.60}$$

where $e_c \in \mathbb{R}^n$, then

$$V\text{Softmax}(K^\top Q)W_O = \underbrace{\begin{bmatrix} 0_{(r-1) \times p} \\ \widetilde{L}^\top \\ 0_{(d_0 - r) \times p} \end{bmatrix}}_{d_0 \times p} \underbrace{\begin{bmatrix} \alpha_1 & \alpha_2 & \cdots & \alpha_n & 0_{p \times (p-n)} \end{bmatrix}}_{p \times p} \underbrace{\begin{bmatrix} \eta e_c^\top \\ 0_{(p-n) \times n} \end{bmatrix}}_{p \times n},$$

then the following approximation error

$$\|V\text{Softmax}(K^\top Q)W_O - V[e_{k_1}, e_{k_2}, \cdots, e_{k_n}]\|_\infty < |b| \cdot \epsilon_0,$$

should become

$$\|V\text{Softmax}(K^\top Q)W_O - \sum_{i=1}^{n} \eta_i \widetilde{L}_{k_i} e_r e_c^\top\|_\infty < \|\eta\|_\infty \cdot n \cdot |b| \cdot \epsilon_0,$$

where the outer product $e_r e_c^\top$ create a matrix with $1$ at $(r, c)$ and $0$ elsewhere. We denote the attention with modifications in (G.59) and (G.60) as $\text{Attn}_{r,c}$.

Lastly, the error of the interpolation point is

$$\left| \sum_{i=1}^{n} \eta_i \widetilde{L}_{k_i} - \sum_{i=1}^{n} \eta_i \text{Range}_{[a,b]}(w_i^\top x_i + t_i) \right| \leq \sum_{i=1}^{n} \eta_i \left| \widetilde{L}_{k_i} - \text{Range}_{[a,b]}(w_i^\top x_i + t_i) \right|$$

(By triangle inequality)

$$= \|\eta\|_\infty \cdot n \cdot \frac{b-a}{p}.$$

Thus we have

$$\|\text{Attn}_{r,c}(X) - \cdot(\sum_{i=1}^{n} \eta_i \text{Range}_{[a,b]}(w_i^\top x_i + t_i))e_r e_c^\top\|_\infty \leq \|\eta\|_\infty \cdot n \cdot (|b| \cdot \epsilon_0 + \frac{b-a}{p}). \quad \text{(G.61)}$$

In fact, if we assume for every $i$ we have $a \leq (w_i^\top x_i + t_i) \leq b$ and $\eta_i = 1$ for $i \in [n]$, the term $\sum_{i=1}^{n} \text{Range}_{[a,b]}(w_i^\top x_i + t_i)$ become

$$\sum_{i=1}^{n} \text{Range}_{[a,b]}(w_i^\top x_i + t_i) = \sum_{i=1}^{n} (w_i^\top x_i + t_i) = \widetilde{w}^\top \widetilde{x} + \widetilde{t}, \quad \text{(G.62)}$$

where $\widetilde{x} \in \mathbb{R}^{dn}$ is the flatten vector of input sequence $X$, with $\widetilde{w} = [w_1^\top, \cdots, w_n^\top] \in \mathbb{R}^{dn}$ and $\widetilde{t} = [t_1^\top, \cdots, t_n^\top] \in \mathbb{R}^{dn}$.

Hence (G.61) become

$$\|\text{Attn}_{r,c}(X) - \cdot(\widetilde{w}^\top \widetilde{x} + \widetilde{t})e_r e_c^\top\|_\infty \leq n \cdot (|b| \cdot \epsilon_0 + \frac{b-a}{p}). \quad \text{(G.63)}$$

Recall that we aim to approximate $f_{rc}(\cdot) : \mathbb{R}^{d \times n} \to \mathbb{R}$ by showing attention mechanism approximate FFN with $N$ neurons $\sum_{i=1}^{N} \eta_i \text{ReLU}(\widetilde{w}_{n(r'-1)+c',i}^\top \widetilde{x} + \widetilde{t}_i)$ in the $(r, c)$-th entry of the attention output. Until now we success to construct one-layer single-head attention layer $\text{Attn}_{r,c}(\cdot)$ whose output $(r, c)$ entry is a linear model on the whole sequence $\eta(\widetilde{w}^\top \widetilde{x} + \widetilde{t})$ by (G.63).

What left is to use a second attention layer in Theorem G.22 to create the ReLU function and sum them up. We know by Theorem G.22, the attention layer take one input token into truncated linear model.

We construct the first layer multi-head attention with $d_o n$ head, each head is $\text{Attn}_{r,c}^{(1)}(\cdot)$ for $r \in [d_o], c \in [n]$.

Pass it to the second layer of another $\text{Attn}_{r',c'}^{(2)}(\cdot)$ where $r' \in [d_o], c' \in [n]$, and $k \in [d_o]$ for later use, we have

$$\text{Attn}_{r',c'}^{(2)}(\sum_{r \in [d_o], c \in [n]} \text{Attn}_{r,c}^{(1)}(X)) \approx \text{Attn}_{r',c'}^{(2)}(\underbrace{\begin{bmatrix} \widetilde{w}_{1,1}^\top \widetilde{x} + \widetilde{t}_{1,1} & \cdots & \widetilde{w}_{1,n}^\top \widetilde{x} + \widetilde{t}_{1,n} \\ \vdots & \vdots & \vdots \\ \widetilde{w}_{d_o,1}^\top \widetilde{x} + \widetilde{t}_{d_o,1} & \cdots & \widetilde{w}_{d_o,n}^\top \widetilde{x} + \widetilde{t}_{d_o,n} \end{bmatrix}}_{d_o \times n})$$

(By (G.63))

$$\approx \sum_{i=1}^{n} \eta_i \text{Range}_{[a,b]}(w_k^\top \begin{bmatrix} \widetilde{w}_{1,i}^\top \widetilde{x} + \widetilde{t}_{1,i} \\ \widetilde{w}_{2,i}^\top \widetilde{x} + \widetilde{t}_{2,i} \\ \vdots \\ \widetilde{w}_{d_o,i}^\top \widetilde{x} + \widetilde{t}_{d_o,i} \end{bmatrix} + t_k))e_{r'}e_{c'}^\top$$

(By (G.61))

$$= \sum_{i=1}^{n} \eta_i \text{ReLU}(\widetilde{w}_{n(r'-1)+c',i}^{\top}\widetilde{x} + \widetilde{t}_{k,i})e_{r'}e_{c'}^{\top}.$$

$$\text{(By letting } w_k = e_k, t_k = 0, a = 0, \text{ and } \widetilde{w}_{k,i}^{\top}\widetilde{x} + \widetilde{t}_{k,i} \leq b)$$

Denote $M := \sum_{r \in [d_o], c \in [n]} \text{Attn}_{r,c}^{(1)}(X)$ and $Y := \begin{bmatrix} \widetilde{w}_{1,1}^{\top}\widetilde{x} + \widetilde{t}_{1,1} & \cdots & \widetilde{w}_{1,n}^{\top}\widetilde{x} + \widetilde{t}_{1,n} \\ \vdots & \vdots & \vdots \\ \widetilde{w}_{d_o,1}^{\top}\widetilde{x} + \widetilde{t}_{d_o,1} & \cdots & \widetilde{w}_{d_o,n}^{\top}\widetilde{x} + \widetilde{t}_{d_o,n} \end{bmatrix}.$

The approximation error is

$$\|\text{Attn}_{r',c'}^{(2)}(M) - \sum_{i=1}^{n} \eta_i \text{ReLU}(\widetilde{w}_{n(r'-1)+c',i}^{\top}\widetilde{x} + \widetilde{t}_i)e_r e_c^{\top}\|_{\infty}$$

$$\leq \|\text{Attn}_{r',c'}^{(2)}(M) - \text{Attn}_{r',c'}^{(2)}(Y)\|_{\infty} + \|\text{Attn}_{r',c'}^{(2)}(Y) - \sum_{i=1}^{n} \eta_i \text{ReLU}(\widetilde{w}_{n(r'-1)+c',i}^{\top}\widetilde{x} + \widetilde{t}_i)e_r e_c^{\top}\|_{\infty}$$

$$\leq \|\text{Attn}_{r',c'}^{(2)}(M) - \text{Attn}_{r',c'}^{(2)}(Y)\|_{2,\infty} + \|\text{Attn}_{r',c'}^{(2)}(Y) - \sum_{i=1}^{n} \eta_i \text{ReLU}(\widetilde{w}_{n(r'-1)+c',i}^{\top}\widetilde{x} + \widetilde{t}_i)e_r e_c^{\top}\|_{\infty}$$

$$\text{(By } \|A\|_{\infty,\infty} \leq \|A\|_{2,\infty})$$

$$\leq \|\text{Attn}_{r',c'}^{(2)}(M) - \text{Attn}_{r',c'}^{(2)}(Y)\|_{2,\infty} + n(|b| \cdot \epsilon_0 + \frac{b}{p}) \qquad \text{(By (G.61))}$$

$$\leq \|W_O\|_{\infty}\|W_V^{\top}\|_2 \left(1 + 4\|W_K^{\top}W_Q\|_2\right)\|A(M) - A(Y))\|_{2,\infty} + n(|b| \cdot \epsilon_0 + \frac{b}{p})$$

$$\text{(By lipschitzness of attention modifying from (Edelman et al., 2022, Lemma A.14))}$$

$$= \|W_O\|_{\infty}\|W_V^{\top}\|_2 \left(1 + 4\|W_K^{\top}W_Q\|_2\right)\|M - Y\|_{2,\infty} + n(|b| \cdot \epsilon_0 + \frac{b}{p}) \qquad \text{($A$ preserves column norm)}$$

$$\leq \|W_O\|_{\infty}\|W_V^{\top}\|_2 \left(1 + 4\|W_K^{\top}W_Q\|_2\right)\sqrt{n}\|M - Y\|_{\infty} + n(|b| \cdot \epsilon_0 + \frac{b}{p}) \qquad \text{(By } \|A\|_{2,\infty} \leq \|A\|_{\infty})$$

$$\leq \|W_O\|_{\infty}\|W_V^{\top}\|_2 \left(1 + 4\|W_K^{\top}W_Q\|_2\right) \cdot d_o n^{\frac{5}{2}}(|b| \cdot \epsilon_0 + \frac{b}{p}) + n(|b| \cdot \epsilon_0 + \frac{b}{p})$$

$$= (|b| \cdot \epsilon_0 + \frac{b}{p}) \cdot (\|W_O\|_{\infty}\|W_V^{\top}\|_2 \left(1 + 4\|W_K^{\top}W_Q\|_2\right) \cdot d_o n^{\frac{5}{2}} + n), \qquad \text{(G.64)}$$

where we show $A(\cdot)$ does not change the maximum column norm and the term $\|M - Y\|_{\infty}$ is bounded as follow. To see why $A(\cdot)$ preserves the maximum column norm, from (G.4) we know

$$A(X) = \underbrace{\begin{bmatrix} I_d \\ 0_{(d+d_o+2) \times d} \end{bmatrix}}_{L} X \underbrace{[I_n, 0_{n \times (p-n)}]}_{R} + \text{Constant}$$

$$= LXR + \text{Constant},$$

and we have

$$A(X_1) - A(X_2) = L(X_1 - X_2)R.$$

Let $X = X_1 - X_2$, we aim to show

$$\|LXR\|_{2,\infty} = \|X\|_{2,\infty}.$$

By definition we have

$$\|LXR\|_{2,\infty} = \max_{j \in [p]} \|(LXR)_{:,j}\|_2,$$

that is, the maximum Euclidean norm of column vector of $LXR$. However, we know the effect of $XR$ is to create $0$ column vector on the right of $X$, and $L(XR)$ just create zero row vector to $X$. Hence, the maximum Euclidean norm of column vector of $X$ is the same as that of $LXR$, that is $\|LXR\|_{2,\infty} = \|X\|_{2,\infty}$.

To bound the term $\|M - Y\|_\infty$, first denote $E_{r,c} = \text{Attn}^{(1)}_{r,c}(X) - \left(\widetilde{w}^\top_{r,c}\widetilde{x} + \widetilde{t}_{r,c}\right)e_r e_c^\top$, then $M - Y = \sum_{r,c} E_{r,c}$. We have

$$
\begin{aligned}
\|M - Y\|_\infty &= \|\sum_{r=1}^{d_o}\sum_{c=1}^n E_{r,c}\|_\infty \\
&\leq \sum_{r=1}^{d_o}\sum_{c=1}^n \|E_{r,c}\|_\infty && \text{(By triangle inequality)} \\
&\leq \sum_{r=1}^{d_o}\sum_{c=1}^n n(|b|\cdot\epsilon_0 + \frac{b}{p}) && \text{(By (G.63))} \\
&= d_o n^2(|b|\cdot\epsilon_0 + \frac{b}{p}).
\end{aligned}
$$

For the bound on (G.64), $\epsilon_0$ is arbitrarily small when $\beta$ in softmax function is sufficiently large. If we further set $p = \Omega(n^{5/2})$, (G.64) is bounded or be arbitrary small when $n$ increase.

Hence for now we construct a multihead attention whose output $(r', c')$ entry is an approximation of an FFN that is a universal approximator of every continuous function defined on compact domain $f_{r'c'}: \mathbb{R}^{dn} \to \mathbb{R}$.

Thus combine the error of attention approximate ReLU neural network and the error of ReLU network approximate target function we have

$$
\left|(\text{Attn}^{(2)}_{r',c'}(\sum_{r\in[d_o],c\in[n]}\text{Attn}^{(1)}_{r,c}(X)))_{r',c'} - f_{r'c'}(X)\right|
$$

$$
\leq \left|(\text{Attn}^{(2)}_{r',c'}(\sum_{r,c}\text{Attn}^{(1)}_{r,c}(X)))_{r',c'} - \sum_{i=1}^n \widetilde{w}^\top_{n(r'-1)+c',i}\widetilde{x} + \widetilde{t}_i\right| + \left|\sum_{i=1}^n \widetilde{w}^\top_{n(r'-1)+c',i}\widetilde{x} + \widetilde{t}_i - f_{r'c'}(X)\right|
$$

$$
\leq (|b|\cdot\epsilon_0 + \frac{b}{p})\cdot(\|W_O\|_\infty\|W_V^\top\|_2\left(1 + 4\|W_K^\top W_Q\|_2\right)\cdot d_o n^{\frac{5}{2}} + n) + \varepsilon \leq \epsilon, \quad \text{for} \quad p = \Omega(n^{5/2}).
$$
$$\tag{G.65}$$

where $\varepsilon$ is the approximation error of the ReLU network, and the second inequality comes from triangle inequality. The universal approximation theory of ReLU neural network in infinity norm is shown in (Pinkus, 1999, Theorem 3.1).

Note that we remove the restriction that the neurons of FFN we aim to approximate is restricted by $n$ by increasing the output sequence length of the first layer $\text{Attn}_{r,c}$. We achieve simply by increasing $n$ in the matrix $W_O$ of our attention to arbitrary positive integer $N$.

Finally, by constructing $dn$ head of this second layer attention $\text{Attn}^{(2)}_{r',c'}$, and set $d_o = dn$ in the first layer attention $\sum_{r\in[d_o],c\in[N]}\text{Attn}^{(1)}_{r,c}$ we get

$$
\begin{aligned}
&\|f(X) - \sum_{r'\in[d],c'\in[n]}\text{Attn}^{(2)}_{r',c'}(\sum_{r\in[d_o],c\in[N]}\text{Attn}^{(1)}_{r,c}(X))\|_\infty \\
&= \max_{r'\in[d],c'\in[n]}\left|f_{r'c'}(X) - (\text{Attn}^{(2)}_{r',c'}(\sum_{r\in[d_o],c\in[N]}\text{Attn}^{(1)}_{r,c}(X)))_{r',c'}\right| \\
&\leq \epsilon. && \text{\scriptsize(By (G.65) each $(r,c)$-th difference is at most $\epsilon$, the $\max_{r,c}$ is also most $\epsilon$)}
\end{aligned}
$$

This completes the proof. □

## G.9 PROOFS OF THEOREM D.1

**Remark G.24** (Key Technique).

$$\underset{k\in\{0,1,2,\cdots,p-1\}}{\mathrm{argmin}} \quad (w_i^\top x_i + t_i - \widetilde{L}_k)^2$$

$$= \underset{k\in\{0,1,2,\cdots,p-1\}}{\mathrm{argmin}} \quad (-2w_i^\top x_i - 2t_i)\cdot \widetilde{L}_k + \widetilde{L}_k^2 + (w_i^\top x_i + t_i)^2$$

$$= \underset{k\in\{0,1,2,\cdots,p-1\}}{\mathrm{argmin}} \quad (-2w_i^\top x_i - 2t_i)\cdot (\widetilde{L}_k - \widetilde{L}_0) - \widetilde{L}_0^2 + \widetilde{L}_k^2 \quad {\scriptstyle ((w_i^\top x_i + t_i) \text{ and } \widetilde{L}_0 \text{ are constant w.r.t. } k)}$$

$$= \underset{k\in\{0,1,2,\cdots,p-1\}}{\mathrm{argmin}} \quad (-2w_i^\top x_i - 2t_i + \widetilde{L}_0 + \widetilde{L}_k)\cdot (\widetilde{L}_k - \widetilde{L}_0)$$

$$= \underset{k\in\{0,1,2,\cdots,p-1\}}{\mathrm{argmin}} \quad (-2w_i^\top x_i - 2t_i + \widetilde{L}_0 + \widetilde{L}_k)\cdot k\Delta L$$

$$= \underset{k\in\{0,1,2,\cdots,p-1\}}{\mathrm{argmin}} \quad (-2w_i^\top x_i - 2t_i + \widetilde{L}_0 + \widetilde{L}_k)\cdot k. \quad {\scriptstyle (\text{Multiply a positive constant doesn't change } \mathrm{argmin}(\cdot))}$$

We first prove the in-context version of our main theorem.

**Theorem G.25** (Theorem D.1 Restated). *Fix real numbers $a < b$, and let the truncation operator* $\mathrm{Range}_{[a,b]}(\cdot)$ *follow Definition 3.1. Let*

$$X = \underbrace{\begin{bmatrix} x_1 & x_2 & \cdots & x_n \\ w & w & \cdots & w \\ t & t & \cdots & t \end{bmatrix}}_{2d+1\times n},$$

*where $w, x_i (i \in [n])$ are bounded. For a precision parameter $p \in \mathbb{N}_+$ satisfying $p > n$, let $\epsilon = O\left(\frac{1}{p}\right)$. Then, for any $\epsilon > 0$ and $d, d_0 \in \mathbb{N}_+$, there exists a single-layer, single-head self-attention with linear transformation $A$: $\mathrm{Attn} \circ A : \mathbb{R}^{d\times n} \to \mathbb{R}^{d_o \times n}$ both irrelevant to $w$ and $t$ such that*

$$\|\mathrm{Attn} \circ A(X)_{:,i} - \mathrm{Range}_{[a,b]}(w_i^\top x_i + t_i)e_{\widetilde{k}_i}\|_\infty \leq \underbrace{|b|\cdot \epsilon_0}_{\textit{finite-}\beta \textit{ softmax error}} + \underbrace{(b-a)/p}_{\textit{interpolation error}}, \quad \textit{for} \quad i \in [n],$$

*where $\widetilde{k}_i \in [d_o]$ is defined as*

$$k_i = \underset{k\in\{0,1,\cdots,p-1\}}{\mathrm{argmin}} \quad ((-2x_i^\top w - 2t + \widetilde{L}_0 + \widetilde{L}_k)\cdot k),$$

$$\widetilde{k}_i = G(k_i).$$

*Here $G : [p] \to [d_o]$ denotes any set-to-set function sending each integer $k_i$ into an appropriate interpolation index $\widetilde{k}_i \in [d_o]$ for $i \in [n]$, and $e_{\widetilde{k}_j} \in \mathbb{R}^{d_o}$ denotes a one-hot vector with a value of $1$ at the $\widetilde{k}_i$-th index and $0$ elsewhere.*

*Proof.* Before we plug the input token to the self-attention, we preprocess it with linear transformations $A : \mathbb{R}^{d\times n} \to \mathbb{R}^{(2d+d_0+2)\times p}$. Without loss of generality, we set the precision parameter $p \in \mathbb{N}$ defined in Definition 3.3 to be larger than input sequence length $n$.

First we denote $\ell_k := k\widetilde{L}_k + k\widetilde{L}_0 - 2kt, \widetilde{L}_k$ following Definition 3.3.

Define a linear transform $A$ such that

$$
A(X) = \underbrace{\begin{bmatrix} I_d & 0_{d\times d+1} \\ 0_{(d+d_o+2)\times d} & 0_{(d+d_o+2)\times(d+1)} \end{bmatrix}}_{(2d+d_0+2)\times(2d+1)} X \underbrace{\begin{bmatrix} I_n, 0_{n\times(p-n)} \end{bmatrix}}_{n\times p}
$$

$$
+ \underbrace{\begin{bmatrix} 0_{d\times d} & 0_{d\times d} & 0_d \\ 0_{d\times d} & I_d & 0_d \\ 0_{1\times d} & 0_{1\times d} & -1 \\ 0_{(d_o+1)\times d} & 0_{(d_o+1)\times d} & 0_{d_o+1} \end{bmatrix}}_{(2d+d_0+2)\times(2d+1)} X \underbrace{\begin{bmatrix} 0 & 1 & \cdots & (p-1) \\ 0_{n-1} & 0_{n-1} & \cdots & 0_{n-1} \end{bmatrix}}_{n\times p}
$$

$$
+ \underbrace{\begin{bmatrix} 0_d & 0_d & \cdots & 0_d & 0_d & \cdots & 0_d \\ 0_d & 0_d & \cdots & 0_d & 0_d & \cdots & 0_d \\ 0 & \widetilde{L}_1 + \widetilde{L}_0 & \cdots & (n-1)(\widetilde{L}_{n-1}+\widetilde{L}_0) & n(\widetilde{L}_n+\widetilde{L}_0) & \cdots & (p-1)(\widetilde{L}_{p-1}+\widetilde{L}_0) \\ & & & \widetilde{L}_{d_o\times p} & & & \\ 1 & 1 & \cdots & 1 & 0 & \cdots & 0 \end{bmatrix}}_{(2d+d_o+2)\times p}
$$

$$
= \underbrace{\begin{bmatrix} x_1 & x_2 & \cdots & x_n & 0 & \cdots & 0 \\ 0_d & w & \cdots & (n-1)w & nw & \cdots & (p-1)w \\ 0 & \ell_1 & \cdots & \ell_{n-1} & \ell_n & \cdots & \ell_{p-1} \\ & & & \widetilde{L}_{d_o\times p} & & & \\ 1 & 1 & \cdots & 1 & 0 & \cdots & 0 \end{bmatrix}}_{(2d+d_o+2)\times p},
$$

where $\widetilde{L} = [\widetilde{L}_0 e_{G(1)}, \cdots, \widetilde{L}_j e_{G(j)}, \widetilde{L}_{p-1} e_{G(p)}] \in \mathbb{R}^{d_0\times p}$.

The output of this linear mapping is the same as the linear mapping output (G.4) in Appendix G.2, hence the remaining proof is the same.

This completes the proof. $\qquad\square$

### G.10 PROOFS OF THEOREM D.4

**Theorem G.26** (Restate of Theorem D.4). *Let $l : \mathbb{R} \times \mathbb{R} \to \mathbb{R}$ be any $C^1$ convex loss function defined on $(w^\top x_i, y_i)$. With input $X$ in the form of Definition 4.1, when $X$ is bounded, there exists a multi-head self-attention $\mathrm{Attn}_m$ whose parameters are irrelevant $X$, with skip connections and each attached with a linear layer, such that for any $\epsilon > 0$, we have*

$$
\left\| \mathrm{Attn}_m \circ A(X) - \begin{bmatrix} x_1 & \cdots & x_n \\ y_1 & \cdots & y_n \\ w - \eta \nabla L(w) & \cdots & w - \eta \nabla L(w) \\ 1 & \cdots & 1 \end{bmatrix} \right\|_\infty \leq \epsilon,
$$

*where $\eta$ denotes the learning rate and $L(w) := (1/n) \sum_{i=1}^n l(w^\top x_i, y_i)$ is an empirical loss upon the given input-output pairs.*

*Proof.* The main goal of the proof is to show multihead attention approximate $\nabla L(w) = (1/n) \sum_{i=1}^n \frac{\partial}{\partial_w} l(w^\top x_i, y_i)$.

Our proof consists of the following steps:

1. Approximate the derivative of loss function $l_w(w^\top x_i, y_i) := \frac{\partial}{\partial_w} l(w^\top x_i, y_i)$ by classical ReLU neural network.

2. Use single-head attention to approximate a $\mathrm{ReLU}$ nested linear function by Theorem D.1.

3. Use multihead attention to aggregate the $\mathrm{ReLU}$ function to approximate ReLU neural network (NN).

4. Combine the error of multihead attention approximate ReLU NN and the error of ReLU NN approximate $\frac{\partial}{\partial_w} l(w^\top x_i, y_i)$.

5. Design $W_O^*$ matrix to sum over $\frac{\partial}{\partial_w} l(w^\top x_i, y_i)$ on different in-context example $(x_i, y_i)$ to get $\nabla L(w)$.

We now begin our proof.

Since $l$ is $C^1$, the derivative of $l$ to $w$ is continuous. By standard universal approximation results (Pinkus, 1999), there exists a set of ReLU neural network with parameter $a_h^{(r)}, b_h^{(r)}, c_h^{(r)} \in \mathbb{R}$ bounded by $B_R$, for all $h \in [H]$. The subscript $r \in [d]$ indicates the $r$-th coordinate of partial derivative $l_w(w^\top x_i, y_i)$, such that for any $\epsilon_0 > 0$

$$
\|(l_w(w^\top x_i, y_i))_{r,:} - \sum_{h=1}^H \mathrm{ReLU}(a_h^{(r)} w^\top x_i + b_h^{(r)} y_i + c_h^{(r)})\|_\infty \leq \epsilon_0. \tag{G.66}
$$

We begin to construct multihead attention with linear mapping to approximate $\sum_{h=1}^H \mathrm{ReLU}(a_h^{(r)} w^\top x_i + b_h^{(r)} y_i + c_h^{(r)})$.

Construct a linear transform $L_{h,r} \in \mathbb{R}^{(2d+4)\times(2d+2)}$ to be

$$
L_{h,r} := \begin{bmatrix} \overbrace{\mathrm{diag}(a_h^{(r)} 1_{1\times d}, b_h^{(r)})}^{(d+1)\times(d+1)} & 0_{d\times d} & 0_{(d+1)\times 1} \\ 0_{1\times(d+1)} & 0_{1\times d} & 1 \\ 0_{(d+1)\times(d+1)} & I_{d\times d} & 0_{d\times 1} \\ 0_{1\times(d+1)} & 0_{1\times d} & c_h^{(r)} \end{bmatrix}.
$$

$L_{h,r}(X)$ outputs

$$
L_{h,r}(X) = \left[\begin{array}{ccc}
\overbrace{\mathrm{diag}(a_h^{(r)}1_{1\times d}, b_h^{(r)})}^{(d+1)\times(d+1)} & 0_{d\times d} & 0_{(d+1)\times 1} \\
0_{1\times(d+1)} & 0_{1\times d} & 1 \\
0_{(d+1)\times(d+1)} & I_{d\times d} & 0_{d\times 1} \\
0_{1\times(d+1)} & 0_{1\times d} & c_h^{(r)}
\end{array}\right]
\underbrace{\left[\begin{array}{cccc}
x_1 & x_2 & \cdots & x_n \\
y_1 & y_2 & \cdots & y_n \\
w & w & \cdots & w \\
1 & 1 & \cdots & 1
\end{array}\right]}_{(2d+2)\times n}
$$

$$
= \underbrace{\left[\begin{array}{cccc}
a_h^{(r)}x_1 & a_h^{(r)}x_2 & \cdots & a_h^{(r)}x_n \\
b_h^{(r)}y_1 & b_h^{(r)}y_2 & \cdots & b_h^{(r)}y_n \\
1 & 1 & \cdots & 1 \\
w & w & \cdots & w \\
1 & 1 & \cdots & 1 \\
c_h^{(r)} & c_h^{(r)} & \cdots & c_h^{(r)}
\end{array}\right]}_{(2d+4)\times n}.
$$

View $[x_i^\top, y_i, 1]$ as a whole input vector corresponding to the $x_i$ in Theorem D.1, view $[w^\top, 1, c_h^{(r)}]$ as the $w$ in Theorem D.1, and set $t = 0$. Let $B_1$ denote the bound of $\|X\|_1 = \sum_{i,j}|X_{ij}|$, then according to Theorem D.1, there exists a $\mathrm{Attn}_h^*$ and $A_h^*$ such that the $i$-th column of output satisfy

$$
\|\mathrm{Attn}_{h,r}^* \circ A_{h,r}^*(L_{h,r}(X))_{:,i} - \mathrm{Range}_{[0,B_RB_1^2]}((w,1,c_h^{(r)})^\top \begin{bmatrix} a_h^{(r)}x_i \\ b_h^{(r)}y_i \\ 1 \end{bmatrix}) \underbrace{e_{d+1+r}}_{(2d+2)\times 1}\|_\infty
$$

$$
\text{(By selecting the output dimension in Theorem D.1 to be 1)}
$$

$$
\leq \epsilon_1, \quad \text{for} \quad i \in [n], \tag{G.67}
$$

for any $\epsilon_1 > 0$.

Notice that

$$
\left|(a_h^{(r)}w^\top x_i + b_h^{(r)}y_i + c_h^{(r)})\right| \leq |a_h^{(r)}|B_1^2 + |b_h^{(r)}|B_1 + |c_h^{(r)}| \quad \text{(By } \|x_i\|_1 \leq B_1, |y_i| \leq B_1 \text{ and } \|w\|_1 \leq B_1.\text{)}
$$

$$
\leq B_R B_1^2,
$$

the truncated linear model $\mathrm{Range}_{[0,B_RB_1^2]}\cdot$ reduce to $\mathrm{ReLU}(\cdot)$

$$
\mathrm{Range}_{[0,B_RB_1^2]}((w,1,c_h^{(r)})^\top \begin{bmatrix} a_h^{(r)}x_i \\ b_h^{(r)}y_i \\ 1 \end{bmatrix}) = \mathrm{ReLU}(a_h^{(r)}w^\top x_i + b_h^{(r)}y_i + c_h^{(r)}).
$$

Hence (G.67) become

$$
\|\mathrm{Attn}_{h,r}^* \circ A_{h,r}^*(L_{h,r}(X))_{:,i} - \mathrm{ReLU}(a_h^{(r)}w^\top x_i + b_h^{(r)}y_i + c_{h,r}^{(r)}) \underbrace{e_{d+1+r}}_{(2d+2)\times 1}\|_\infty \leq \epsilon_1, \quad i \in [d]
$$

Thus, summing the $H$ head output we get

$$
\|\sum_{h=1}^H \mathrm{Attn}_{h,r}^* \circ A_{h,r}^*(L_{h,r}(X))_{:,i} - \sum_{h=1}^H \mathrm{ReLU}(a_h^{(r)}w^\top x_i + b_h^{(r)}y_i + c_h^{(r)}) \underbrace{e_{d+1+r}}_{(2d+2)\times 1}\|_\infty \leq H\epsilon_1, \quad r \in [d].
$$

Until now we success to construct multihead attention $\sum_{h=1}^{H} \text{Attn}_{h,r}^* \circ A_{h,r}^*(L_{h,r}(X))$ to approximate $\sum_{h=1}^{H} \text{ReLU}(a_h^{(r)} w^\top x_i + b_h^{(r)} y_i + c_h^{(r)})$ on the $(d+1+r, i)$ entry.

Combine the above expression for all $r \in [d]$ we have

$$\|\sum_{h=1}^{H}\sum_{r=1}^{d} \text{Attn}_{h,r}^* \circ A_{h,r}^*(L_{h,r}(X))_{:,i} - \sum_{h=1}^{H}\sum_{r=1}^{d} \text{ReLU}(a_h^{(r)} w^\top x_i + b_h^{(r)} y_i + c_h^{(r)}) \underbrace{e_{d+1+r}}_{(2d+2)\times 1} \|_\infty \leq d \cdot H\epsilon_1$$

$$(G.68)$$

We first bound the error for ReLU neural network to approximate $l_w(w^\top x_i, y_i)$.

By (G.66) we derive

$$\|\sum_{h=1}^{H}\sum_{r=1}^{d} \text{ReLU}(a_h^{(r)} w^\top x_i + b_h^{(r)} y_i + c_h^{(r)}) \underbrace{e_{d+1+r}}_{(2d+2)\times 1} - \begin{bmatrix} \overbrace{0}^{(d+1)\times 1} \\ \underbrace{l_w(w^\top x_i, y_i)}_{d\times 1} \\ 0 \end{bmatrix} \|_\infty$$

$$\leq \sum_{r=1}^{d} \|\sum_{h=1}^{H} \text{ReLU}(a_h^{(r)} w^\top x_i + b_h^{(r)} y_i + c_h^{(r)}) \underbrace{e_{d+1+r}}_{(2d+2)\times 1} - \begin{bmatrix} \overbrace{0}^{(d+1)\times 1} \\ \underbrace{l_w(w^\top x_i, y_i)}_{d\times 1} \\ 0 \end{bmatrix} \|_\infty \quad \text{(By triangle inequality)}$$

$$\leq d\epsilon_0,$$

where the last line is from (G.66).

Combine this with (G.68), we have

$$\|\sum_{h=1}^{H}\sum_{r=1}^{d} \text{Attn}_{h,r}^* \circ A_{h,r}^*(L_{h,r}(X))_{:,i} - \begin{bmatrix} \overbrace{0}^{(d+1)\times 1} \\ \underbrace{l_w(w^\top x_i, y_i)}_{d\times 1} \\ 0 \end{bmatrix} \|_\infty$$

$$\leq \|\sum_{h=1}^{H}\sum_{r=1}^{d} \text{Attn}_{h,r}^* \circ A_{h,r}^*(L_{h,r}(X))_{:,i} - \sum_{h=1}^{H}\sum_{r=1}^{d} \text{ReLU}(a_h^{(r)} w^\top x_i + b_h^{(r)} y_i + c_h^{(r)}) \underbrace{e_{d+1+r}}_{(2d+2)\times 1} \|_\infty$$

$$+ \|\sum_{h=1}^{H}\sum_{r=1}^{d} \text{ReLU}(a_h^{(r)} w^\top x_i + b_h^{(r)} y_i + c_h^{(r)}) \underbrace{e_{d+1+r}}_{(2d+2)\times 1} - \begin{bmatrix} \overbrace{0}^{(d+1)\times 1} \\ \underbrace{l_w(w^\top x_i, y_i)}_{d\times 1} \\ 0 \end{bmatrix} \|_\infty$$

$$\leq dH\epsilon_1 + d\epsilon_0.$$

What left is to sum up the derivative of loss function $l_w(w^\top x_i, y_i)$ on $n$ in-context example $x_i, y_i$ for $i \in [n]$ to form $\nabla L(w) = (1/n)\sum_{i=1}^{n} l_w(w^\top x_i, y_i)$.

We construct $W_O^*$ and integrate it into the original $W_O^*$ of $\text{Attn}$ to turn the loss gradient into a step of gradient descent

$$W_O^* := \begin{bmatrix} -\frac{\eta}{n}1_n & -\frac{\eta}{n}1_n & \cdots & -\frac{\eta}{n}1_n \end{bmatrix}.$$

Now we define the final form of our network

$$\text{Attn}^*_{h,r} = \text{Attn}^*_{h,r}(Z)W^*_O,$$
$$A_h(Z) = A^*_{h,r} \circ L_{h,r}(Z)$$

Thus we have

$$\| \sum_{h=1}^{H} \sum_{r=1}^{d} \text{Attn}_{h,r} \circ A_{h,r}(X) - (-\eta \begin{bmatrix} \overbrace{0}^{(d+1)\times 1} \\ \underbrace{\nabla L(w)}_{d\times 1} \\ 0 \end{bmatrix}) \|_\infty$$

$$= \| \sum_{h=1}^{H} \sum_{r=1}^{d} \text{Attn}_{h,r} \circ A_{h,r}(X) - (-\frac{\eta}{n} \sum_{i=1}^{n} \begin{bmatrix} \overbrace{0}^{(d+1)\times 1} \\ \underbrace{l_w(w^\top x_i, y_i)}_{d\times 1} \\ 0 \end{bmatrix}) \|_\infty \qquad \text{(By } W^*_O)$$

$$\leq dH\epsilon_1 + d\epsilon_0.$$

With skip connections, we have

$$\| \sum_{h=1}^{H} \sum_{r=1}^{d} \text{Attn}_{h,r} \circ A_{h,r}(X) + X - (X - \frac{\eta}{n} \sum_{i=1}^{n} \begin{bmatrix} \overbrace{0}^{(d+1)\times 1} \\ \underbrace{l_w(w^\top x_i, y_i)}_{d\times 1} \\ 0 \end{bmatrix}) \|_\infty \leq dH\epsilon_1 + d\epsilon_0,$$

where

$$X - \frac{\eta}{n} \sum_{i=1}^{n} \begin{bmatrix} \overbrace{0}^{(d+1)\times 1} \\ \underbrace{l_w(w^\top x_i, y_i)}_{d\times 1} \\ 0 \end{bmatrix} = \begin{bmatrix} x_1 & x_2 & \cdots & x_n \\ y_1 & y_2 & \cdots & y_n \\ w - \eta\nabla L(w) & w - \eta\nabla L(w) & \cdots & w - \eta\nabla L(w) \\ 1 & 1 & \cdots & 1 \end{bmatrix}.$$

Setting $dH\epsilon_1 + d\epsilon_0 \leq \epsilon$ yields the final result.

This completes the proof. $\qquad\qquad\square$

## H   RELU, HARD TANH AND CLIPPED RELU ACTIVATION FUNCTIONS

**Example H.1** (Truncated Linear Model Subsumes ReLU)**.** *When $a = 0$ and $b \to \infty$, $\text{Range}_{[a,b]}(w^\top x + t)$ reduces to the standard ReLU (ramp function). Conversely, choosing finite values for $a$ and $b$ saturates the function on both ends, effectively making $\text{Range}_{[a,b]}(w^\top x + t)$ a double-sided ReLU. It retains the piecewise linearity essential for universal approximation while bounding the output values.*

**Example H.2** (Truncated Linear Model Subsumes Hard Tanh)**.** *Consider $a = -1$ and $b = +1$. Then $\text{Range}_{[a,b]}(x)$ becomes the hard tanh activation:*

$$HardTanh(x) = \begin{cases} -1, & x \leq -1, \\ x, & -1 < x < +1, \\ +1, & x \geq +1. \end{cases}$$

*Thus, truncated linear functions recover this bounded, piecewise-linear activation.*

**Example H.3** (Truncated Linear Model Subsumes Clipped ReLU). *When $a = 0$ and $b > 0$ is finite,* $\text{Range}_{[0,b]}(x)$ *matches a clipped ReLU. That is,*

$$ClippedReLU_{[0,b]}(x) = \max\{0, \min\{x, b\}\},$$

*which maintains linearity in the interval $[0, b]$ and saturates at both ends.*

# I  SEQUENCE-TO-SEQUENCE UNIVERSAL APPROXIMATION BASED ON THEOREM 3.9

This section extends the softmax attention sequence-to-sequence approximation result of Theorem 3.14 to a more Transformer-native setting.

**Lemma I.1** (Attention simulates column-wise linear transformations)**.** *Let $X \in \mathbb{R}^{d \times n}$ and let*

$$\ell(X) := AXB \in \mathbb{R}^{d_{\text{out}} \times n}, \quad A \in \mathbb{R}^{d_{\text{out}} \times d}, \; B \in \mathbb{R}^{n \times n}$$

*be a linear map that is token-wise in $A$ and sequence-wise in $B$. Assume that all entries of $B$ are strictly positive.[3] Consider the augmented input*

$$Z := \begin{bmatrix} X & 0_d \\ I_n & 0_n \\ 0_{1 \times n} & 1 \end{bmatrix} \in \mathbb{R}^{(d+n+1) \times (n+1)},$$

*where $0_d \in \mathbb{R}^{d \times 1}$ be the all-zeros vector. Then for any $\epsilon > 0$, there exists a single-head attention*

$$\mathrm{Attn}(Z) = W_V Z \cdot \mathsf{Softmax}\big((W_K Z)^\top (W_Q Z)\big)$$

*such that*

$$\left\| \mathrm{Attn}(Z) - [\ell(X) \quad 0_{d_{\text{out}}}] \right\|_\infty \le \epsilon.$$

*Proof Sketch.* The goal is to realize the linear map $X \mapsto AXB$ in the first $n$ output columns and to keep the last (padding) column close to $0$. The construction proceeds in four steps:

1. Choose $W_V$ so that the values store $3MAX$ for real tokens and $0$ for the padding token.

2. Choose $W_K$ and $W_Q$ so that the first $n$ attention columns implement mixing by $B/(3M)$.

3. Use a large parameter $T$ in $W_Q$ so that the last attention column concentrates on the padding token, which yields an output close to $0$ there.

$\square$

*Proof.* For each column $i \in [n]$ of $B$, set

$$s_i := \sum_{r=1}^n B_{ri}, \qquad S := (s_1, \ldots, s_n) \in \mathbb{R}^{1 \times n},$$

and let

$$M := \max_{i \in [n]} s_i.$$

Strict positivity of the entries implies $0 < B_{ri} \le M$ and

$$3M - s_i \ge 2M > 0$$

---

[3] Any matrix $B$ admits an decomposition $B^+ - B^-$ with $B^+, B^- \ge 0$. The attention construction for positive matrices applies separately to $B^+$ and $B^-$, and combine through multi-head architecture yields the general case.

for all $r, i$. The constant $M$ will serve as a common denominator for the softmax normalization.

**Step 1: Values $V$ store $AX$ and ignore the padding token.**

Define

$$W_V := 3M \begin{bmatrix} A & 0_{d_{\text{out}} \times (n+1)} \end{bmatrix} \in \mathbb{R}^{d_{\text{out}} \times (d+n+1)}.$$

Then

$$V := W_V Z = 3MA[X \quad 0_d] \in \mathbb{R}^{d_{\text{out}} \times (n+1)}.$$

**Step 2: Keys and queries implement mixing by $B$.**

Let $1_n \in \mathbb{R}^{n \times 1}$ be the all-ones vector and let $T > 0$ be a scalar parameter (chosen later). Set

$$W_Q := [0_{n \times d} \quad I_n \quad T1_n] \in \mathbb{R}^{n \times (d+n+1)}.$$

Writing $Z = [z_1, \ldots, z_{n+1}]$, one obtains

$$Q_{:,j} := W_Q z_j = \begin{cases} e_j, & j \leq n, \\ T1_n, & j = n+1, \end{cases}$$

so

$$Q = W_Q Z = [e_1, \ldots, e_n, T1_n] \in \mathbb{R}^{n \times (n+1)}.$$

For the keys, define

$$W_K := \begin{bmatrix} 0_{n \times d} & \ln(B^\top) & \ln(3M1_n - S^\top) \end{bmatrix} \in \mathbb{R}^{n \times (d+n+1)},$$

where the logarithm appears entrywise and $\ln(3M1_n - S^\top) \in \mathbb{R}^{n \times 1}$ has $k$-th entry $\ln(3M - s_k)$. Then $K := W_K Z \in \mathbb{R}^{n \times (n+1)}$ satisfies, for $i \leq n$,

$$K_{:,i} = \ln(B^\top)e_i,$$

whose $r$-th entry equals $(K_{:,i})_r = \ln B_{ir}$, and

$$K_{:,n+1} = \ln(3M1_n - S^\top),$$

whose $r$-th entry equals $(K_{:,n+1})_r = \ln(3M - s_r)$.

Now consider the score matrix

$$Y := K^\top Q \in \mathbb{R}^{(n+1) \times (n+1)},$$

with entries $Y_{ij} = \langle K_{:,i}, Q_{:,j} \rangle$.

For $i \leq n$ and $j \leq n$,

$$Y_{ij} = \langle \ln(B^\top)e_i, e_j \rangle = \ln B_{ij}.$$

For $i = n+1$ and $j \leq n$,

$$Y_{n+1,j} = \langle \ln(3M1_n - S^\top), e_j \rangle = \ln(3M - s_j).$$

Thus, for each $j \le n$,

$$Y_{:,j} = \begin{bmatrix} \ln B_{1j} \\ \vdots \\ \ln B_{nj} \\ \ln(3M - s_j) \end{bmatrix}.$$

Apply column-wise softmax and write

$$W_{ij} := \big(\mathsf{Softmax}(Y)\big)_{ij} = \frac{\exp(Y_{ij})}{\sum_{r=1}^{n+1} \exp(Y_{rj})}.$$

Then

$$\exp(Y_{ij}) = \begin{cases} B_{ij}, & i \le n, \\ 3M - s_j, & i = n+1, \end{cases}$$

and

$$\sum_{r=1}^{n+1} \exp(Y_{rj}) = s_j + (3M - s_j) = 3M.$$

Hence, for $j \le n$,

$$W_{ij} = \begin{cases} B_{ij}/(3M), & i \le n, \\ (3M - s_j)/(3M), & i = n+1, \end{cases}$$

or in block form,

$$W_{:,1:n} = \begin{bmatrix} B/(3M) \\ 1_{1 \times n} - S/(3M) \end{bmatrix}.$$

Combining this with $V = 3MA[X \ 0_d]$ yields for the first $n$ columns

$$\mathrm{Attn}(Z)_{:,1:n} = VW_{:,1:n} = 3MAX \cdot \frac{B}{3M} = AXB = \ell(X). \tag{I.1}$$

So the first $n$ tokens already match the desired output exactly.

**Step 3: The padding column stays close to zero.** The remaining task is to control the last column $j = n+1$. Here $q_{n+1} = T1_n$ enters, and one obtains

$$Y_{i,n+1} = \begin{cases} TH_1(i), & i \le n, \\ TH_2, & i = n+1, \end{cases}$$

where

$$H_1(i) := \sum_{r=1}^{n} \ln B_{ir}, \qquad H_2 := \sum_{r=1}^{n} \ln(3M - s_r).$$

Since $0 < B_{ir} \le M$ and $3M - s_r \ge 2M$,

$$H_1(i) \le n \ln M,$$
$$H_2 \ge n \ln(2M),$$

so $H_2 - H_1(i) \geq n \ln 2 > 0$ for all $i \leq n$. Thus the last column of $Y$ has a strictly larger entry at index $n + 1$ than at any index $i \leq n$. For any $\delta > 0$, a sufficiently large choice of $T$ yields

$$\max_{i \leq n} W_{i,n+1} \leq \delta, \qquad \left|W_{n+1,n+1} - 1\right| \leq \delta.$$

In words, the last attention column concentrates on the padding token.

Recall that $V_{:,n+1} = 0_{d_{\text{out}}}$. Writing the last column of $W$ as

$$W_{:,n+1} = \begin{bmatrix} a_0(T) \\ a_1(T) \end{bmatrix},$$

with $a_0(T) \in \mathbb{R}^{n \times 1}$ and $a_1(T) \in \mathbb{R}$, gives

$$\text{Attn}(Z)_{:,n+1} = VW_{:,n+1} = 3MAX a_0(T),$$

since the contribution from the padding token vanishes. The bounds on $W_{i,n+1}$ imply $\|a_0(T)\|_\infty \leq \delta$ and $\|a_0(T)\|_1 \leq n\delta$, so

$$\left\|\text{Attn}(Z)_{:,n+1}\right\|_\infty \leq 3M\|AX\|_\infty \|a_0(T)\|_1 \leq 3Mn\|AX\|_\infty \delta.$$

For a given $\epsilon > 0$, choose $\delta$ and then $T$ so that $3Mn\|AX\|_\infty \delta \leq \epsilon$.

**Step 4: Final error bound.** The first $n$ output columns equal $\ell(X)$ exactly according to (I.1), and the last column has infinity norm at most $\epsilon$ by the choice of $T$. Therefore

$$\left\|\text{Attn}(Z) - [\ell(X) \quad 0_{d_{\text{out}}}]\right\|_\infty \leq \epsilon,$$

which completes the proof. $\qquad\qquad\square$

**Theorem I.2** (Sequence-to-Sequence Universal Approximation of Multi-Head Softmax Attention)**.** *Let $1 \leq p < \infty$. Let $\mathcal{X} \subset \mathbb{R}^{d \times n}$ be a compact domain of input sequences. Let $f : \mathcal{X} \to \mathbb{R}^{d \times n}$ be a continuous sequence-to-sequence function For any $\epsilon > 0$, there exists a network $\Phi$ composed of three multi-head attention layers such that*

$$\|\Phi(X) - f(X)\|_{L_p} < \epsilon.$$

*Proof Sketch.* We devide the proof into three stage.

1. We first show that there exist a multi-head attention approximating the pre-activation of ReLU neural network.

2. We then construct the second attention layer to reorganize and share information across tokens through Lemma I.1.

3. Finally, we construct the third attention layer to approximate ReLU activation and the final linear combination to approximate ReLU neural network.

$\qquad\qquad\square$

*Proof.* Let $X = [x_1, x_2, \cdots, x_n] \in \mathbb{R}^{d \times n}$ denote the input sequence. We first establish the result for a function $f : \mathbb{R}^{d \times n} \to \mathbb{R}^{1 \times n}$ acting on a sequence with a single output dimension. Generalization to multiple output dimensions follows by stacking such constructions. By standard universal approximation theorems for Feed-Forward Networks (FFNs) (Pinkus, 1999), for any $\epsilon_{FFN} > 0$, there exists an approximation of $f$ taking the form of a sum of ReLUs (flatten input $R^{dn}$). Let

$\sum_{k=1}^{N} a_{i,k} \mathrm{ReLU}(\sum_{j=1}^{n} w_{i,k,j}^{\top} x_j), a_{i,k} \in \{-1, 1\}^4, w_{i,k,j} \in \mathbb{R}^d$ denote the FFN approximation of the $i$-th token of the target function $f_i$.

**Preprocessing.**

Before feeding the input to the network, we pad the input with a zero token and append a positional encoding at the bottom. We denote this augmented input as

$$X_p := \begin{bmatrix} X & 0_n \\ I_n & 0_n \\ 0_{1 \times n} & 1 \end{bmatrix}.$$

**First Layer: Token-Wise Linear Pre-Activations.**

In this layer, we first show that there exist multi-head attention $\mathrm{Attn}_1$ approximating the pre-activation linear model of ReLU neural network.

According to Theorem 3.9, for every $k \in [N]$, there exists a multi-head attention $\mathrm{Attn}_{1,k}^*$ that for any $\epsilon_0 > 0$ approximates

$$\| \mathrm{Attn}_{1,k}^*(X_p) - e_{(i-1)N+k}^{(nN+n+1)} \begin{bmatrix} w_{i,k,1}^{\top} x_1 & w_{i,k,2}^{\top} x_2 & \cdots & w_{i,k,n}^{\top} x_n & 0 \end{bmatrix} \|_\infty \le \epsilon_0.$$

Next we construct an attention head $\mathrm{Attn}_I$ to preserve the identity matrix at the bottom

$$\mathrm{Attn}_I(X_p) := \begin{bmatrix} 0_{nN \times (2n+1)} \\ S_I \end{bmatrix} X_p \mathsf{Softmax}(\beta(S_I X_p)^{\top} S_I X_p),$$

where

$$S_I := \begin{bmatrix} 0_{(n+1) \times n} & I_{n+1} \end{bmatrix}.$$

Multiplication by $S_I$ simply discards the top $n$ rows of $X_p$ and keeps the bottom $(n+1)$ rows, hence

$$S_I X_p = I_{n+1}.$$

The value, key, and query projections used in $\mathrm{Attn}_I$ are therefore

$$V = \begin{bmatrix} 0_{nN \times (2n+1)} \\ S_I \end{bmatrix} X_p = \begin{bmatrix} 0_{nN \times (n+1)} \\ I_{n+1} \end{bmatrix},$$

$$K = S_I X_p = I_{n+1}, \qquad Q = S_I X_p = I_{n+1}.$$

The score matrix is

$$\beta K^{\top} Q = \beta I_{n+1}.$$

Hence $\mathsf{Softmax}(\beta K^{\top} Q)$ becomes arbitrarily close to the identity when $\beta$ is large. As a result, the attention output $V \mathsf{Softmax}(\beta K^{\top} Q)$ nearly copies the bottom identity block unchanged while keeping the upper rows zero.

now we know $\mathrm{Attn}_I$ satisfies

$$\| \mathrm{Attn}_I(X_p) - \begin{bmatrix} 0_{nN \times (n+1)} \\ I_{n+1} \end{bmatrix} \|_\infty \le \epsilon_0.$$

---

[4]Because ReLU is positively homogeneous, we absorb $|a_{i,k}|$ into the weights inside the ReLU and keep only $\mathrm{sign}(a_{i,k}) \in \{-1, 1\}$ outside.

Summing these heads ($\mathrm{Attn}_{1,k}^*$ for $k \in [N]$ and $\mathrm{Attn}_I$) defines $\mathrm{Attn}_1$

$$\mathrm{Attn}_1 := \mathrm{Attn}_1^* + \mathrm{Attn}_I.$$

It satisfies

$$\|\mathrm{Attn}_1 - \begin{bmatrix} W_0 \\ I_{n+1} \end{bmatrix}\| = \|\mathrm{Attn}_1^*(X_p) + \mathrm{Attn}_I(X_p) - \begin{bmatrix} W_0 \\ I_{n+1} \end{bmatrix}\|_\infty \le \epsilon_0,$$

where $W_0$ stacks the pre-activation blocks

$$W_0 := \begin{bmatrix} W_1 \\ W_2 \\ \vdots \\ W_n \end{bmatrix},$$

with each $W_i \in \mathbb{R}^{N \times (n+1)}$ defined as

$$W_i := \begin{bmatrix} w_{i,1,1}^\top x_1 & \cdots & w_{i,1,n}^\top x_n & 0 \\ w_{i,2,1}^\top x_1 & \cdots & w_{i,2,n}^\top x_n & 0 \\ \vdots & \ddots & \vdots & \vdots \\ w_{i,N,1}^\top x_1 & \cdots & w_{i,N,n}^\top x_n & 0 \end{bmatrix}.$$

**Second Layer: Reorganization and Mix Information across Tokens.**

Now we construct the second layer.

Define $A_i, i \in [N]$ as

$$A_i := \begin{bmatrix} 0_{N \times (i-1)N} & I_N & 0_{N \times (n-i)N} & 0_{1 \times (n+1)} \\ 0_{N \times (i-1)N} & 0_{N \times N} & 0_{N \times (n-i)N} & I_{n+1} \end{bmatrix}.$$

This matrix will select the rows in the output of the first layer which are needed for the $i$-th head in the second layer.

According to Lemma I.1, for any $\epsilon_2 > 0$ there exists an attention $\mathrm{Attn}_{2,i}$ that satisfies

$$\|\mathrm{Attn}_{2,i}(\begin{bmatrix} W_i \\ I_{n+1} \end{bmatrix}) - (\sum_{s=0}^{N-1} \underbrace{e_{sn+i}^{(nN+n+1)}(e_{s+1}^{(N)})^\top}_{E_{sn+i,s+1}})W_i \begin{bmatrix} 0_{(n+1)\times(i-1)} & 1_{n+1} & 0_{(n+1)\times(n-i)} \end{bmatrix}\|_\infty \le \epsilon_2.$$

Here $W_i := (W_0)_{k,:}$ is the $i$-th row of $W_0$.

Since

$$A_i \cdot \begin{bmatrix} W_0 \\ I_{n+1} \end{bmatrix} = \begin{bmatrix} W_i \\ I_{n+1} \end{bmatrix},$$

we have

$$\|\mathrm{Attn}_{2,i} \circ A_i(\begin{bmatrix} W_0 \\ I_{n+1} \end{bmatrix}) - (\sum_{s=0}^{N-1} \underbrace{e_{sn+i}^{(nN+n+1)}(e_{s+1}^{(N)})^\top}_{E_{sn+i,s+1}})W_i \begin{bmatrix} 0_{(n+1)\times(i-1)} & 1_{n+1} & 0_{(n+1)\times(n-i)} \end{bmatrix}\|_\infty \le \epsilon_2.$$

We note that $\mathrm{Attn}_{2,i} \circ A_i$ is also an attention.

Summing the above constructed heads yields

$$\|\sum_{i=1}^{n}\text{Attn}_{2,i}\circ A_i(\begin{bmatrix}W_0\\I_{n+1}\end{bmatrix})-\begin{bmatrix}\text{diag}(w_{1,1}^\top\bar{X},w_{2,1}^\top\bar{X},\cdots,w_{n,1}^\top\bar{X}) & 0\\ \text{diag}(w_{1,2}^\top\bar{X},w_{2,2}^\top\bar{X},\cdots,w_{n,2}^\top\bar{X}) & 0\\ \vdots & \vdots\\ \text{diag}(w_{1,N}^\top\bar{X},w_{2,N}^\top\bar{X},\cdots,w_{n,N}^\top\bar{X}) & 0\\ 0_{(n+1)\times n} & 0_{n+1}\end{bmatrix}\|\le\epsilon_2,$$

in which

$$\bar{X}:=\begin{bmatrix}x_1\\x_2\\\vdots\\x_n\end{bmatrix}$$

and

$$w_{i,k}:=\begin{bmatrix}w_{i,k,1}\\w_{i,k,2}\\\vdots\\w_{i,k,n}\end{bmatrix},$$

where

$$w_{i,k}^\top\bar{X}=\sum_{j=1}^{n}w_{i,k,j}^\top x_j.$$

We also use an attention head to preserve the identity matrix, constructed like the previous head we used to preserve the identity matrix

$$\text{Attn}_I'(X_p):=\begin{bmatrix}0_{N\times(nN+n+1)}\\S_I'\end{bmatrix}X_p\text{Softmax}(\beta(S_I'X_p)^\top S_I'X_p),$$

in which $S_I'$ is

$$\begin{bmatrix}0_{(n+1)\times(nN)} & I_{n+1}\end{bmatrix}.$$

adding this head yields the final output of the second layer to satisfy

$$\|\sum_{i=1}^{n}\text{Attn}_{2,i}\circ A_i(\begin{bmatrix}W_0\\I_{n+1}\end{bmatrix})+\text{Attn}_I'(\begin{bmatrix}W_0\\I_{n+1}\end{bmatrix})-\begin{bmatrix}\text{diag}(w_{1,1}^\top\bar{X},w_{2,1}^\top\bar{X},\cdots,w_{n,1}^\top\bar{X}) & 0_n\\ \text{diag}(w_{1,2}^\top\bar{X},w_{2,2}^\top\bar{X},\cdots,w_{n,2}^\top\bar{X}) & 0_n\\ \vdots & \vdots\\ \text{diag}(w_{1,N}^\top\bar{X},w_{2,N}^\top\bar{X},\cdots,w_{n,N}^\top\bar{X}) & 0_n\\ I_n & 0_n\\ 0_{1\times n} & 1\end{bmatrix}\|_\infty\le\epsilon_2.$$

This defines the second layer $\text{Attn}_2$

$$\text{Attn}_2:=\sum_{i=1}^{n}\text{Attn}_{2,i}\circ A_i+\text{Attn}_I'.$$

**Last Layer: ReLU and Aggregation.**

We now show that the third attention layer implements the nonlinearity and the final signed aggregation of the FFN. For notation simplicity later, we redefine the output after $\text{Attn}_2$ we as

$$Z^{(2)} = \begin{bmatrix} \text{diag}(s_{1,1}, \ldots, s_{n,1}) & 0_n \\ \text{diag}(s_{1,2}, \ldots, s_{n,2}) & 0_n \\ \vdots & \vdots \\ \text{diag}(s_{1,N}, \ldots, s_{n,N}) & 0_n \\ I_n & 0_n \\ 0_{1 \times n} & 1 \end{bmatrix} \in \mathbb{R}^{(Nn+n+1) \times (n+1)},$$

where

$$s_{i,k} := w_{i,k}^\top \bar{X}, \qquad \bar{X} = \begin{bmatrix} x_1 \\ \vdots \\ x_n \end{bmatrix}.$$

Let's get some intuition of our final goal. Stacking these pre-activations as rows yields an $N \times n$ matrix

$$S := \begin{bmatrix} s_{1,1} & \cdots & s_{n,1} \\ \vdots & \ddots & \vdots \\ s_{1,N} & \cdots & s_{n,N} \end{bmatrix},$$

so that the $k$-th block on top of $Z^{(2)}$ is exactly $\text{diag}(S_{k,:})$.

The FFN approximation of the function with a single output dimension form

$$f(X) = [f_1(X) \quad \cdots \quad f_n(X)] \approx \sum_{k=1}^{N} a^{(k)} \odot \text{ReLU}(S_{k,:}),$$

where $a^{(k)} := (a_{1,k}, \ldots, a_{n,k}) \in \{\pm 1\}^n$ and $\odot$ denotes elementwise multiplication. Thus the third layer $\text{Attn}_3$ must map the block-diagonal structure in $Z^{(2)}$ to this signed ReLU combination.

Now let's start to construct the attention weight matrices to achieve this goal.

*Values, keys, and queries.* For a fixed $k \in [N]$, define the value projection

$$W_V^{(k)} := \begin{bmatrix} 0_{1 \times (k-1)n} & 1_{1 \times n} & 0_{1 \times ((N-k)n+n+1)} \end{bmatrix} \in \mathbb{R}^{1 \times (Nn+n+1)}.$$

By construction, $W_V^{(k)}$ picks out the $k$-th $n \times (n+1)$ block at the top of $Z^{(2)}$ and sums its rows. Since that block is diagonal, we obtain

$$V_k := S_k Z^{(2)} = [S_{k,1} \quad \cdots \quad S_{k,n} \quad 0] \in \mathbb{R}^{1 \times (n+1)},$$

that is, the $k$-th row of $S$ appears as the first $n$ entries of $V_k$, and the value at the padding token is 0. In the notation of the attention operator above, this means

The common key projection is chosen as

$$W_K := [0_{n \times nN} \quad I_n \quad 0_{n \times 1}] \in \mathbb{R}^{n \times (Nn+n+1)}.$$

Multiplying by $Z^{(2)}$ selects the identity block in the bottom $(n+1) \times (n+1)$ portion:

$$K := W_K Z^{(2)} = [I_n \quad 0_{n \times 1}] \in \mathbb{R}^{n \times (n+1)}.$$

*Query projections for the positive and negative parts.* For a fixed $k \in [N]$, define

$$C_1 := \mathrm{diag}(1_{a_{1,k}=1}, \ldots, 1_{a_{n,k}=1}),$$
$$C_2 := \mathrm{diag}(1_{a_{1,k}=-1}, \ldots, 1_{a_{n,k}=-1}),$$
$$D := -1_{n \times (n+1)} + [I_n \quad 1_n] \in \mathbb{R}^{n \times (n+1)}.$$

The matrices $C_1$ and $C_2$ select positions with positive and negative coefficients $a_{i,k}$, respectively, while $D$ compares with a reference (the padding token) as we describe below.

The query projections for the two heads corresponding to hidden unit $k$ are

$$W_Q^{(1,k)} := \begin{bmatrix} 0_{n \times (k-1)n} & C_1 & 0_{n \times (N-k)n} & D \end{bmatrix},$$
$$W_Q^{(2,k)} := \begin{bmatrix} 0_{n \times (k-1)n} & C_2 & 0_{n \times (N-k)n} & D \end{bmatrix}.$$

Writing $Z^{(2)}$ in block form as

$$Z^{(2)} = \begin{bmatrix} \mathrm{diag}(S_{1,:}) & 0 \\ \vdots & \vdots \\ \mathrm{diag}(S_{N,:}) & 0 \\ I_n & 0 \\ 0_{1 \times n} & 1 \end{bmatrix} = \begin{bmatrix} Z^{\mathrm{top}} \\ Z^{\mathrm{bottom}} \end{bmatrix},$$

with $Z^{\mathrm{top}} \in \mathbb{R}^{Nn \times (n+1)}$ and $Z^{\mathrm{bottom}} \in \mathbb{R}^{(n+1) \times (n+1)}$, we obtain

$$W_Q^{(1,k)} Z^{(2)} = C_1 \, \mathrm{diag}(S_{k,:}) + D \, Z^{\mathrm{bottom}},$$
$$W_Q^{(2,k)} Z^{(2)} = C_2 \, \mathrm{diag}(S_{k,:}) + D \, Z^{\mathrm{bottom}}.$$

Since $Z^{\mathrm{bottom}}$ is the $(n+1) \times (n+1)$ identity,

$$D \, Z^{\mathrm{bottom}} = D,$$

and hence

$$Q_k^{(1)} := W_Q^{(1,k)} Z^{(2)} = C_1 \, \mathrm{diag}(S_{k,:}) + D,$$
$$Q_k^{(2)} := W_Q^{(2,k)} Z^{(2)} = C_2 \, \mathrm{diag}(S_{k,:}) + D.$$

*Score matrices and their structure.* The score matrices for the first heads associated with hidden unit $k$ are

$$Y_k^{(1)} := K^\top Q_k^{(1)} = K^\top C_1 \, \mathrm{diag}(S_{k,:}) + K^\top D.$$

Since

$$K^\top = \begin{bmatrix} I_n \\ 0_{1 \times n} \end{bmatrix},$$

we can compute the product with $D$ explicitly:

$$K^\top D = \begin{bmatrix} I_n \\ 0_{1 \times n} \end{bmatrix} \left( -1_{n \times (n+1)} + [I_n \ 1_n] \right) = E,$$

where

$$E = \begin{bmatrix} 0 & -1 & \dots & -1 & 0 \\ -1 & 0 & \dots & -1 & 0 \\ \vdots & \vdots & \ddots & \vdots & \vdots \\ -1 & -1 & \dots & 0 & 0 \\ 0 & 0 & \dots & 0 & 0 \end{bmatrix} \in \mathbb{R}^{(n+1)\times(n+1)}.$$

Thus

$$Y_k^{(1)} = \mathrm{diag}(1_{a_{1,k}=1}S_{k,1}, \dots, 1_{a_{n,k}=1}S_{k,n}) + E,$$

Equivalently, in full matrix form,

$$Y_k^{(1)} = \begin{bmatrix} 1_{a_{1,k}=1}S_{k,1} & -1 & \dots & -1 & 0 \\ -1 & 1_{a_{2,k}=1}S_{k,2} & \dots & -1 & 0 \\ \vdots & \vdots & \ddots & \vdots & \vdots \\ -1 & -1 & \dots & 1_{a_{n,k}=1}S_{k,n} & 0 \\ 0 & 0 & \dots & 0 & 0 \end{bmatrix},$$

and $Y_k^{(2)}$ is obtained by replacing $1_{a_{i,k}=1}$ with $1_{a_{i,k}=-1}$ on the diagonal.

*Softmax and signed ReLU.* For the first head associated with hidden unit $k$, define the output row vector

$$H_k^{(1)} := V_k \,\mathsf{Softmax}\big(\beta\, Y_k^{(1)}\big) \in \mathbb{R}^{1\times(n+1)},$$

where $V_k = \begin{bmatrix} S_{k,1} & \dots & S_{k,n} & 0 \end{bmatrix}$. Thus the first $n$ coordinates of $H_k^{(1)}$ can be written as

$$H_{k,1:n}^{(1)} = \begin{bmatrix} S_{k,1} & \dots & S_{k,n} & 0 \end{bmatrix} \mathsf{Softmax}\big(\beta\, Y_k^{(1)}\big)_{:,1:n}.$$

From the explicit form of $Y_k^{(1)}$ above, each column $j \le n$ has the structure

$$(Y_k^{(1)})_{:,j} = \begin{bmatrix} -1 \\ \vdots \\ -1 \\ 1_{a_{j,k}=1}S_{k,j} \\ -1 \\ \vdots \\ -1 \\ 0 \end{bmatrix},$$

that is, a diagonal entry $1_{a_{j,k}=1}S_{k,j}$ at position $i = j$, a baseline value $-1$ in all other rows $i \le n$, and 0 at the padding index $i = n + 1$. For large $\beta$, the column-wise Softmax therefore concentrates on *either* the diagonal entry $i = j$ (when $a_{j,k} = 1$ and $S_{k,j} > 0$) *or* on the padding index $i = n + 1$ (when $a_{j,k} = 1$ and $S_{k,j} \le 0$ or when $a_{j,k} = -1$)[5]. Formally, let $\widetilde{W}_k^{(1)} \in \mathbb{R}^{(n+1)\times(n+1)}$ denote the *ideal hardmax weight matrix* whose $j$-th column is

$$(\widetilde{W}_k^{(1)})_{:,j} := \begin{cases} e_j, & a_{j,k} = 1 \text{ and } S_{k,j} > 0, \\ e_{n+1}, & \text{otherwise,} \end{cases}$$

---

[5]We can utilize case (ii) of Lemma F.1 for the edge case $S_{k,j} = 0$.

so that $\mathsf{Softmax}\big(\beta\, Y_k^{(1)}\big)_{:,j} \to \big(\widetilde{W}_k^{(1)}\big)_{:,j}$ as $\beta \to \infty$. Define the corresponding "hard" output

$$\widetilde{H}_k^{(1)} := V_k\, \widetilde{W}_k^{(1)} \in \mathbb{R}^{1\times(n+1)}.$$

By the definition of $V_k = [\,S_{k,1}\ \ldots\ S_{k,n}\ 0\,]$ and the columns of $\widetilde{W}_k^{(1)}$, the $j$-th coordinate of $\widetilde{H}_k^{(1)}$ satisfies

$$\widetilde{H}_{k,j}^{(1)} = \begin{cases} S_{k,j}, & a_{j,k}=1 \text{ and } S_{k,j}>0, \\ 0, & \text{otherwise,} \end{cases} = 1_{a_{j,k}=1}\,\mathrm{ReLU}(S_{k,j}).$$

Hence, in vector form,

$$\widetilde{H}_{k,1:n}^{(1)} = \big(1_{a_{1,k}=1}\,\mathrm{ReLU}(S_{k,1}),\ldots,1_{a_{n,k}=1}\,\mathrm{ReLU}(S_{k,n})\big) \in \mathbb{R}^{1\times n}.$$

Since $\mathsf{Softmax}\big(\beta\, Y_k^{(1)}\big)$ converges column-wise to $\widetilde{W}_k^{(1)}$ as $\beta \to \infty$, we have

$$H_k^{(1)} = V_k\,\mathsf{Softmax}\big(\beta\, Y_k^{(1)}\big) \approx \widetilde{H}_k^{(1)},$$

and in particular

$$H_{k,1:n}^{(1)} \approx \big(1_{a_{1,k}=1}\,\mathrm{ReLU}(S_{k,1}),\ldots,1_{a_{n,k}=1}\,\mathrm{ReLU}(S_{k,n})\big).$$

The second head is treated analogously. Writing

$$H_k^{(2)} := V_k\,\mathsf{Softmax}\big(\beta\, Y_k^{(2)}\big),$$

the same reasoning applied to $Y_k^{(2)}$ yields

$$H_{k,1:n}^{(2)} \approx \big(1_{a_{1,k}=-1}\,\mathrm{ReLU}(S_{k,1}),\ldots,1_{a_{n,k}=-1}\,\mathrm{ReLU}(S_{k,n})\big).$$

The module $\mathrm{Attn}_{3,k}$ is defined as the difference of these two single-head attentions,

$$\mathrm{Attn}_{3,k}(Z^{(2)}) := V_k\mathsf{Softmax}\big(\beta\, Y_k^{(1)}\big) - V_k\mathsf{Softmax}\big(\beta\, Y_k^{(2)}\big) = H_k^{(1)} - H_k^{(2)},$$

so that its first $n$ coordinates satisfy

$$\big(\mathrm{Attn}_{3,k}(Z^{(2)})\big)_{1:n} \approx \big(a_{1,k}\,\mathrm{ReLU}(S_{k,1}),\ldots,a_{n,k}\,\mathrm{ReLU}(S_{k,n})\big) = a^{(k)} \odot \mathrm{ReLU}(S_{k,:}).$$

Summing over $k$ yields

$$\big(\mathrm{Attn}_3(Z^{(2)})\big)_{1:n} = \sum_{k=1}^{N}\big(\mathrm{Attn}_{3,k}(Z^{(2)})\big)_{1:n} \approx \sum_{k=1}^{N} a^{(k)} \odot \mathrm{ReLU}(S_{k,:}),$$

which exactly matches the length-$n$ output of the FFN approximation. Therefore there exists $\beta > 0$ such that

$$\left\|\mathrm{Attn}_3\left(\begin{bmatrix} \mathrm{diag}(s_{1,1},\ldots,s_{n,1}) & 0_n \\ \vdots & \vdots \\ \mathrm{diag}(s_{1,N},\ldots,s_{n,N}) & 0_n \\ I_n & 0_n \\ 0_{1\times n} & 1 \end{bmatrix}\right)_{:,1:n} - \Big[\sum_{k=1}^{N} a_{1,k}\mathrm{ReLU}(S_{k,1}) \quad \cdots \quad \sum_{k=1}^{N} a_{n,k}\mathrm{ReLU}(S_{k,n})\Big]\right\|_\infty \leq \epsilon_2.$$

Finally we truncate the padded token and define $\Phi$ as

$$T \circ \text{Attn}_3 \circ \text{Attn}_2 \circ \text{Attn}_1 \circ P$$

where $P$ is the preprocessing step that pads 1 zero token and $T$ is the truncation step that delete the last token.

**Error Analysis and Convergence in $L_p$.**

We now demonstrate that the accumulated error through the three layers can be bounded arbitrarily in the $L_p$ norm. Unlike analyses relying on Lipschitz constants, which may explode with large attention weights, we utilize the uniform continuity of the target operators on compact domains.

Let $L_1, L_2, L_3$ denote the ideal mathematical operators approximated by the three layers (component extraction, column summation, and ReLU aggregation, respectively). Let $H_0 = X_p$ be the input. We define the sequence of ideal feature maps as $H_1 = L_1(H_0)$, $H_2 = L_2(H_1)$, and $H_3 = L_3(H_2)$, where $H_3$ corresponds to the target FFN output. Conversely, let $\widetilde{H}_1 = \text{Attn}_1(H_0)$, $\widetilde{H}_2 = \text{Attn}_2(\widetilde{H}_1)$, and $\widetilde{H}_3 = \text{Attn}_3(\widetilde{H}_2)$ denote the actual outputs of the constructed layers.

Let $C_d$ be a constant such that $\|M\|_p \leq C_d \|M\|_\infty$ for matrices of the relevant dimensions. We seek to show that for any $\epsilon > 0$, the parameters of the attention layers can be chosen such that $\|\widetilde{H}_3 - H_3\|_p < \epsilon$.

We proceed via a backward induction argument. Consider the final layer. By the triangle inequality,

$$\|\widetilde{H}_3 - H_3\|_p = \|\text{Attn}_3(\widetilde{H}_2) - L_3(H_2)\|_p$$
$$\leq \|\text{Attn}_3(\widetilde{H}_2) - L_3(\widetilde{H}_2)\|_p + \|L_3(\widetilde{H}_2) - L_3(H_2)\|_p.$$

The operator $L_3$ involves ReLU functions and linear sums, which are continuous. Since the domain of valid feature maps is compact, $L_3$ is uniformly continuous. Therefore, there exists a $\delta_2 > 0$ such that for any inputs $Y, Y'$ satisfying $\|Y - Y'\|_p < \delta_2$, we have $\|L_3(Y) - L_3(Y')\|_p < \epsilon/2$. Furthermore, by the construction of the third layer, specifically by increasing the Softmax scaling parameter $\beta$, we can limit the approximation error such that $\|\text{Attn}_3(Z) - L_3(Z)\|_p < \epsilon/2$ for all $Z$ in the compact range. Thus, the total error is bounded by $\epsilon$ provided that $\|\widetilde{H}_2 - H_2\|_p < \delta_2$.

We apply the same logic to the second layer. We require $\|\widetilde{H}_2 - H_2\|_p < \delta_2$. Decomposing the error yields

$$\|\widetilde{H}_2 - H_2\|_p \leq \|\text{Attn}_2(\widetilde{H}_1) - L_2(\widetilde{H}_1)\|_p + \|L_2(\widetilde{H}_1) - L_2(H_1)\|_p.$$

The operator $L_2$ is linear and therefore uniformly continuous. Thus, there exists a $\delta_1 > 0$ such that $\|\widetilde{H}_1 - H_1\|_p < \delta_1$ implies $\|L_2(\widetilde{H}_1) - L_2(H_1)\|_p < \delta_2/2$. By Lemma I.1, we can choose the parameters of $\text{Attn}_2$ such that the approximation error $\|\text{Attn}_2(\widetilde{H}_1) - L_2(\widetilde{H}_1)\|_p$ is less than $\delta_2/2$. This condition holds if $\|\widetilde{H}_1 - H_1\|_p < \delta_1$.

Finally, for the first layer, we ensure $\|\widetilde{H}_1 - H_1\|_p < \delta_1$. The input $H_0$ is exact, so there is no propagated error. By Theorem 3.9, we can construct $\text{Attn}_1$ with precision $\epsilon_0 = \delta_1/C_d$ such that

$$\|\widetilde{H}_1 - H_1\|_p \leq C_d \|\text{Attn}_1(H_0) - L_1(H_0)\|_\infty \leq C_d \epsilon_0 = \delta_1.$$

By choosing the construction parameters corresponding to $\epsilon_0, \epsilon_2$ and $\beta$ derived from the moduli of continuity $\delta_1$ and $\delta_2$, we guarantee that the final output satisfies

$$\|\widetilde{H}_3 - H_3\|_p < \epsilon$$

for all $X \in \mathcal{X}$. Equivalently,

$$\sup_{X \in \mathcal{X}} \|\widetilde{H}_3 - H_3\|_p \leq \epsilon.$$

Since the target FFN $H_3$ can approximate the function $f$ to arbitrary precision, the pure attention network in the one-row case is a pointwise universal approximator on $\mathcal{X}$.

**Extension to $d$ output rows.**

The discussion above treats $f : \mathcal{X} \to \mathbb{R}^{1 \times n}$. For a general sequence-to-sequence map $f : \mathcal{X} \to \mathbb{R}^{d \times n}$, write

$$f^{(r)}(X) := f(X)_{r,:} \in \mathbb{R}^{1 \times n}, \qquad r \in [d],$$

and construct, for each $r$, a three-layer attention network $\Phi^{(r)}$ that approximates $f^{(r)}$ with

$$\|\Phi^{(r)}(X) - f^{(r)}(X)\|_p \leq \varepsilon_{\text{row}} \quad \text{for all } X \in \mathcal{X}.$$

Define the combined network

$$\Phi(X) := \begin{bmatrix} \Phi^{(1)}(X) \\ \vdots \\ \Phi^{(d)}(X) \end{bmatrix} \in \mathbb{R}^{d \times n},$$

which corresponds to placing the heads for all $\Phi^{(r)}$ inside the same three multi-head attention layers and concatenate their outputs in the feature dimension as usual. For the full error $E(X) := \Phi(X) - f(X)$, we have

$$\|E(X)\|_p^p = \sum_{r=1}^d \|\Phi^{(r)}(X) - f^{(r)}(X)\|_p^p \leq d\, \varepsilon_{\text{row}}^p,$$

so

$$\|E(X)\|_p \leq d^{1/p}\, \varepsilon_{\text{row}}.$$

Choosing $\varepsilon_{\text{row}} := \varepsilon/d^{1/p}$ yields

$$\|\Phi(X) - f(X)\|_p \leq \varepsilon \quad \text{for all } X \in \mathcal{X}.$$

By the definition of the $L_p$ norm in (2.1) (applied with $\Omega = \mathcal{X}$ and $f(X) = \|\Phi(X) - f(X)\|_p$), we have

$$\|\Phi - f\|_{L_p}^p = \int_{\mathcal{X}} \|\Phi(X) - f(X)\|_p^p \, dX$$

$$\leq \int_{\mathcal{X}} \varepsilon^p \, dX.$$

Since $\mathcal{X}$ is compact, the integral $\int_{\mathcal{X}} 1 \, dX$ is finite. Given any target $\epsilon > 0$, we can run the above construction with a pointwise tolerance $\varepsilon$ small enough so that the right-hand side is at most $\epsilon^p$, which yields

$$\|\Phi - f\|_{L_p} < \epsilon.$$

This complete the proof.

$\square$

