# OpenReview forum: "Universal Approximation with Softmax Attention"
_ICLR.cc/2026/Conference — ICLR 2026 Conference Withdrawn Submission_

### Official Review · Reviewer_dffi · 2025-10-30

**Soundness:** 3
**Presentation:** 1
**Contribution:** 3
**Rating:** 6
**Confidence:** 3

**Summary:**

The paper proves that even a single-layer transformer can be universal approximators for sequence-to-sequence functions, given certain architectural assumptions.

**Strengths:**

1. The paper does explore to what extent scaling the depth is needed for universal approximation in transformers, showing that one can in fact have constant depth of 1 and instead scale the sequence (and possibly hidden) dimensions instead. They also achieve that without using MLPs, solely by leveraging the attention mechanism.
2. They authors show that one can trade-off input length for the number of heads and extend their results to sequence-to-sequence modeling.

**Weaknesses:**

1. The fourth contribution point (“In-Context Approximation and Gradient Descent.”) is not at all discussed in the main text. I understand that it is difficult to fit a theoretical paper in the page limit but there is nevertheless the expectation that the main text is self-contained.
2. I didn’t see a clear note of the scaling properties of their construction, i.e., how do the model input length, hidden dimension and number of parameters scale with the precision?
3. Overall, the constructions were not particularly clearly explained. Still not entirely clear to me what the $G$, $\tilde{k}_i$ and $k_i$ are supposed to be. While Figure 1 was supposed to help, I can’t say I understood much of what it is supposed to illustrate. The paper can really benefit from rewriting Section 3 for clarity.
4. The paper appears to overclaim its novelty a bit. The authors say that prior works have “strong assumptions on data or architecture” but do not explain what these assumptions are. Furthermore, they don’t appear to clearly explain what their own assumptions are (e.g., expanding the input into a much longer input using the initial linear projections and possibly having unbounded activations to achieve “pickiness” of the softmax attention). A clear discussion of the limitations of the paper is missing.

**Questions:**

In Definition 2.1 $W_O$ has dimensions $n \times n_o$ which indicates it is mixing information across the sequence elements, when in practice it should operate element-wise. Is this a typo or a required property?

Typos:
- Line 117: Missing $X$ for $W_Q^\top X$?
- Line 197: these segments serves -> these segments serve
- Line 392: Possibly $R$ should be blackboard font?

---

> ### Author Response · Authors · 2025-11-25
> **Rebuttal 1: Weakness Part I**
>
> ### We thank the reviewer for the detailed review. In response, we have addressed all concerns and questions in the following replies and revisions in the updated draft. All changes from the originally submitted version are highlighted in blue in the revised PDF.
>
> ---
>
> > ### ***W1***: *The fourth contribution point (“In-Context Approximation and Gradient Descent.”) is not at all discussed in the main text. I understand that it is difficult to fit a theoretical paper in the page limit but there is nevertheless the expectation that the main text is self-contained.*
>
> **Response:**
> Thanks for pointing this out. We agree with the reviewer that the main contribution should be contained in the main text.
>
> **In response, we moved the ICL section to the 10th page of the revision as a new `Section 4` in our latest revision pdf.**
>
> To elaborate more on our ICGD results’ contribution, we also briefly highlight the two advanced applications:
>
> * **Task Composition.**
> Our results can be extended to task composition from subtasks. In fact, this is already an ongoing project within our group. Here is a short (non-rigorious) sketch:
>     * Suppose we have $N$ subtasks (e.g., gradient descent, lasso, linear regression, etc.). We want to use them to compose a task.
>     * For each subtask, our results already imply there exists an attention layer to approximate this task in-context (such as ${\rm Attn}_{\mathrm{GD}}$ for the gradient-descent subtask).
>     * We then use another frozen attention layer to approximate this “task-specific attention” in-context based on our UAP results.
>     * In this way, we can use 1 frozen attention layer to approximate  subtasks in-context
>     * Then, we can introduce another "routing" attention layer on top of the frozen layer to compose these subtasks into the task of our interest in-context.
> This way extends our techniques to meta-learning or task composition naturally.
>
> We also remark that “step 3-5” is not trivial.
>
> * **Simulation of Learning Algorithms In-Context.**
>
>     By multi-step in-context gradient descent, softmax attention models can also learn various ML algorithms such as ridge regression, generalized linear models (GLM), and lasso regression as done in [1]. We briefly introduce the proof idea here.
>     * In `Theorem 4.2` of the latest revision, we show single-layer softmax attention approximate one-step ICGD for any $C^1$ loss. This contains a wide range of loss functions, including ridge, GLM and lasso loss functions.
>     * By stacking the same attention layer and using the standard gradient descent convergence results, as described in Lemma 14, Lemma C.1, Proposition A.2 and Proposition A.3 of [1], we can arrive at the desired in-context approximation - simulate a wide range of learning algorithms from input-output examples
>
> We integrated this discussion into `Appendix D DETAILED RESULTS AND DISCUSSION OF IN-CONTEXT LEARNING` to make the ICGD contribution more complete. Please see our latest revision pdf for more details.
>
> [1] Bai, Yu, et al. "Transformers as statisticians: Provable in-context learning with in-context algorithm selection." NeurIPS (2023)
>
> > ### ***W2***: *I didn’t see a clear note of the scaling properties of their construction, i.e., how do the model input length, hidden dimension and number of parameters scale with the precision?*
>
> **Response:**
> Thanks for the question. We are sorry for any confusion caused. **The approximation error rate was discussed at `lines 346-350, line 357`, and `line 377` of the original submission.**
> It scales as $O(1/n)$ and $O(1/(nH))$, for single- and multi-head attention respectively. Thus the required $p$ ($p>n$ is the precision parameter in `Theorem 3.4, 3.9`) grows like $1/ \epsilon$ and the required $H$ like $1/(n\epsilon)$.
>
> In response to **other scalings**, **our seq-to-seq result requires the number of head growth as $O(dp)$, and the total parameters scale as $O(dnp^2)$.** Please see the new discussion on model complexity in `Remark E.1` for more details.
>
> Here we remark that, **the value of $p$ hinges on our desired accuracy by our `Theorem 3.4, 3.9`, and classical ReLU approximation result [1] without Lipschitz assumption (to keep the estimate general).** Its explicit dependence on the "target error" requires specifying the continuity properties of the target function. Please see the details of the `Remark E.1` in related work.
>
> [1] Pinkus, A. Approximation theory of the MLP model in neural networks. Acta numerica (1999)

---

> ### Author Response · Authors · 2025-11-25
> **Rebuttal 2: Weakness Part II**
>
> ---
>
> > ### ***W3***: *Overall, the constructions were not particularly clearly explained. Still not entirely clear to me what the G, \tilde{k}_i, \tilde{k}_i are supposed to be. While Figure 1 was supposed to help, I can’t say I understood much of what it is supposed to illustrate. The paper can really benefit from rewriting Section 3 for clarity.*
>
> **Response:**
>
> Thanks for bringing this up and sorry for the confusion. We agree the original explanation is not clear enough, hence we revised the section 3 as below:
>
> * We refined the `remark 3.7` about the meaning of $k\_i$, $\tilde{k}\_i$, and $G(\cdot)$. We quote here for your convenience:
>     > "Here, we clarify the distinction between $k\_i$ and $\tilde{k}\_i$. The difference lies in their roles within the interpolation and output spaces. Given the $i$-th token $x\_i$, **$k\_i \in \{0, \dots, p-1\}$ identifies the closest interpolation point $\tilde{L}\_{k\_i}$ to the target value $\mathrm{Range}\_{[a,b]}(w\_i^\top x_i + t\_i)$**. In contrast, **$\tilde{k}\_i \in [d\_o]$ is an output coordinate index**: it specifies in which coordinate of the $d_o$-dimensional output vector we place the selected point $\tilde{L}\_{k\_i}$ (grey dashed lines in Figure 1). The mapping $G : \{0,\dots,p-1\} \to [d_o]$ connects these two roles by assigning each interpolation index $k$ a coordinate $\tilde{k} := G(k)$, and for each token $i$ we then have $\tilde{k}_i = G(k\_i)$ (color purple in the font in figure 1). In the simplest case one take $G(k) \equiv 1$ for all $k \in \{1,...,p-1\}$, so that  every $\tilde{L}\_{k\_i}$ is placed in the first row of the output matrix. This flexibility allows $G$ to be tailored to the scenarios considered."
>
> * We updated the `Figure 1` to make it align with the proof sketch of `Theorem 3.4`. We hope this can make the explanation and figure echo with each other. We also update the caption of the figure to give more intuition on the interpolation scheme.
>
> ---
>
> > ### ***W4***: *The paper appears to overclaim its novelty a bit. The authors say that prior works have “strong assumptions on data or architecture” but do not explain what these assumptions are. Furthermore, they don’t appear to clearly explain what their own assumptions are (e.g., expanding the input into a much longer input using the initial linear projections and possibly having unbounded activations to achieve “pickiness” of the softmax attention). A clear discussion of the limitations of the paper is missing.*
>
> **Response:**
>
> Thank you for the comment. We clarify as following two points of the strong assumption we lift:
>
> * **Mild Data Assumption:** As in the original submission, Line 74-76, previous works utilize the concept of contextual mapping and assume input has token-wise separation [1,2]. Furthermore, [3] assumes input data lives on a hypersphere (fixed norm). In contrast, we only assume input within a compact domain.
>
> * **No Smoothness Assumption on Target Function:** Prior works consider target function with shift-equivariant α-smoothness ([4]) or finite complexity measure ([5]), resembling classical smoothness assumptions. In contrast, we only assume continuity on target function.
>
> [1] Yun et al. Are transformers universal approximators of sequence-to-sequence functions?. ICLR (2020)
>
> [2] Kajitsuka & Sato. Are Transformers with One Layer Self-Attention Using Low-Rank Weight Matrices Universal Approximators?. ICLR (2024)
>
> [3] Petrov et al. Prompting a pretrained transformer can be a universal approximator. ICML (2024)
>
> [4] Takakura & Suzuki. Approximation and estimation ability of transformers for sequence-to-sequence functions with infinite dimensional input. ICML (2023)
>
> [5] Jiang & Li (2024). Approximation rate of the transformer architecture for sequence modeling. NeurIPS (2024)
>
>
> In response to **"unbounded activations to achieve “pickiness” of the softmax attention,"** we apologize that, even after a few debates, we are still not sure what you are referring to. This is because **the only non-linear activation in our paper is softmax. And it's by definition bounded by 1 due to normalization.**
>
> So we respectfully ask for further clarifications if possible.

---

> > ### Author Response · Authors · 2025-11-25
> > **Rebuttal 2: Weakness Part III (W4 Continued)**
> >
> > In response to the "assumption" of the linear transformation expanding the input, we respectfully disagree this is an assumption.
> > It is the architecture required to facilitate approximation. There is no assumption on this archetecture (no infinite width, differentiblity/continuity of the network...etc) at all.
> >
> > However, we agree with reviewer that we should make this more clear in the paper. As a result, we
> >
> > - Added an explicit remark after `Theorem 3.4` that
> >     - states that the linear map A is sequence-wise and therefore not a standard layer in transformer, see `Remark G.3` for transformer alternative for this $A$, and
> >     - it serves as a simplest illustrative example to demonstrate our interpolation selection techniques,
> >     - points to `Theorem 3.9` as the version compatible with practical transformers
> >
> > - Added a short “theorem roadmap” at the end of `Section 1` / Modify the start of `Section 3` that explains the roles of `Theorem 3.4` and `Theorem 3.9`.
> >
> > We hope these revisions address your concerns, and welcome any further discussion.
> >
> > We also note that we provided a new seq-to-seq approximation result based on `Theorem 3.9` instead of `Theorem 3.4` in the `Appendix I` of the latest revision.

---

> ### Author Response · Authors · 2025-11-25
> **Rebuttal 3: Questions**
>
> > ### ***Q1***: *In Definition 2.1, W_O has dimensions n \times n_0 which indicates it is mixing information across the sequence elements, when in practice it should operate element-wise. Is this a typo or a required property?*
>
> **Response:**
>
> Thanks for pointing this out. **The UAP result doesn’t rely on this W_O matrix. We introduce it just for technical convenience.** In the proof of original submission `line 1674-1678`, we use $W_O$ as a fixed column selector. It keeps the first n columns and discards the last $p−n$ columns. **It does not mix tokens and it is not trainable.**
> This operation is easily replaced by two common operations in Transformer:
> * **Cross attention.**
>     Effect of $W_O$ is replaceable by cross attention. We build $K$, $V$ as the same, but build $Q$ only using the first n data tokens in $A(X)$:
>
>     In the proof we have $V\,\mathrm{Softmax}(K^\top Q)\, W_O.$ Since the softmax function acts columnwise we have:
>     \begin{align*}
>     \mathrm{Softmax}(K^\top Q) W_O
>     &= \mathrm{Softmax}(K^\top Q W_O) \\
>     &= \mathrm{Softmax}(K^\top Q_{:,n}),
>     \end{align*}
>     which in practice is implemented by cross attention.
>
> * **Self-attention with padding/slicing.**
>     Since the whole point of $W_O$ is to keep the first $n$ columns and discards the last p−n columns.
> $W_O$ operation is replaceable by commonly used slicing operation ($V{\rm Sofmatx}(K^\top Q))_{:n}$ or padding .
>
>
> > ### ***Q2***: *Typos:
> > * Line 117: Missing $X$ for $W^\top_{Q}X$?
> > * Line 197: these segments serves -> these segments serve
> > * Line 392: Possibly $R$ should be blackboard font?*
>
> **Response:**
>
> We thanks the reviewer for the careful reading. We have corrected the above typos and conducted another round of proofreading in our latest revision.
>
> ---
>
> Thank you again for the time and effort invested in our paper. We have addressed all your concerns and questions with utmost care. Please let us know if there is anything we can clarify further. We look forward to further discussion!

---

### Official Review · Reviewer_xyoo · 2025-11-01

**Soundness:** 3
**Presentation:** 3
**Contribution:** 3
**Rating:** 8
**Confidence:** 1

**Summary:**

This paper establishes universal approximation results for softmax attention mechanisms. The authors prove that (i) two-layer self-attention or (ii) one-layer self-attention followed by a softmax function can universally approximate continuous sequence-to-sequence functions on compact domains. They proposed an interpolation-based method showing that attention can approximate generalized ReLU functions with O(1/n) precision for single-head and O(1/(nH)) for H-head attention. The results are extended to in-context learning, demonstrating that attention can approximate gradient descent.

**Strengths:**

1. This paper provides novel theoretical insights. It’s the first work to discuss universal approximation for attention without FFN. The proposed interpolation-based technique is a unique view for the expressiveness of attention.
2. The theoretical foundation is solid, and the only assumption needed is the compactness. The detailed proofs and discussion of multiple cases makes the proof comprehensive and strong.

**Weaknesses:**

This paper is a theoretical paper which lack of real data experiments like language modeling to validate the predictions.

**Questions:**

How does the results change or extend to other attention like causal or masked attention?

---

> ### Author Response · Authors · 2025-11-25
> **Rebuttal 1**
>
> ### We thank the reviewer for the detailed review. In response, we have addressed all concerns and questions in the following replies and revisions in the updated draft. All changes from the originally submitted version are highlighted in blue in the revised PDF.
>
> ---
>
> > ### ***W1***: *This paper is a theoretical paper which lack of real data experiments like language modeling to validate the predictions.*
>
> **Response:**
> Thank you for the comment.
>
> To clarify, **in order to verify our theory, we should have a setting that we have absolute control.
> Using real-world data will add *unwanted confounding factors* that defeat the purpose of the validating our theory.**
>
>
> To elaborate more, we also wish to emphasize that the main goal of the paper is to investigate the fundamental capability of attention mechanism. Along this way, we theoretically extend the expressive power of attention by proposing novel and non-trivial proofing techniques. We believe the current proof-of-concept experiment setting is enough to support the paper’s result.
>
>
> ---
>
> > ### ***Q1***: *How does the results change or extend to other attention like causal or masked attention?*
>
> **Response:**
> Thank you for the question. It's possible to extend to those varants of attention. We imagine the technical difficulty will be that the masked out matrix will need to be compensated by the larger sequence length our linear layer or with extra linear or attention layer to ensure correct information routing.
>
> ---
>
> Thank you again for your review and positive endorsement. We have addressed all your concerns and questions with utmost care. We hope our rebuttal and revisions have increased your confidence in our work. Please let us know if there are any further questions. Thank you!

---

### Official Review · Reviewer_NV7d · 2025-11-01

**Soundness:** 2
**Presentation:** 1
**Contribution:** 2
**Rating:** 2
**Confidence:** 4

**Summary:**

The paper revolves around the proof of a Universal Approximation Property (UAP) for Attention. Crucially, it makes a point of not relying (for the most part) on the additional FFN layers one would normally find in a Transformer, thus departing from previous work, and highlighting the expressive power of the sole Attention layer. In a nutshell, the proof is built by showing that one single single-head attention layer, prepended by a linear operator, can represent a Truncated Linear Model (ie, a ramp) token-wise: that is, given an input X\in\mathbb{R}^{d x N}, attention can mimic Range(wT x_n + t). This result leverages an interpolation-based technique, which is at the core of the paper itself. Notably, this Truncated Linear Model is a basic building block which can be used for constructing piece-wise linear functions, which already hints at its approximation properties. In particular, its functional class includes many universal approximators, including ReLU: the proof then proceeds by leveraging the universality of ReLU approximators, and transferring it to Attention. The authors also provide some simplifications / recasting of the original proof: particularly, a similar result is recovered for a single multi-head attention layer (which simplifies the requirements on the prepending linear layer), and the final ReLU approximation can be done with either two attention layers, or one layer and a softmax nonlinearity.

**Strengths:**

- The approach of the paper is interesting: particularly, the first step of the proof and main contribution (the interpolation-based approximation of the Truncated Linear Model) is insightful
- Also the scope of the paper is relevant, as it claims to dramatically simplify the requirements for UAP for Transformers

**Weaknesses:**

- The paper would benefit from a strong revision to improve its structure and presentation
- Many relevant discussions and results (including mentioned core contributions) are relegated to the appendix
- Most importantly, while the overall proof strategy is clear, I found it difficult to verify its validity, also in light of the reduced clarity

**Questions:**

**The main proof**

There is a number of hidden details in the proof which I believe affect its overall validity, and should be spelled out more clearly, also to better highlight possible limitations in the approach proposed
- **Prepending linear layers** I believe this proof mainly aims at describing the properties of Attention, *as it appears inside a Transformers architecture* (otherwise its usefulness would be rather reduced). If this is the case, then the shape of the linear Layer A(X) used in Thm3.4, is not a valid one: by looking at Fig1 (and in general at its defined shape in L205, A:R^{d\times n}->R^{… \times p}), the layer expands the input tensor sequence-wise, which is not a valid operation in Transformers. In all fairness, you yourselves point this out in AppB and Remark G3, and highlight Thm3.9 (the multi-head version of UAP for attention) as the real “valid theorem” for the sake of proving UAP in a Transformer setting. While indeed the A in L1721 is a valid token-wise operator, due to its bias term it’s a rather unorthodox one—but I agree it can be seen as a special PE. In light of all of this, I understand why you decided to include Thm3.9 in the main (where at a first read it looked like a secondary extension to Thm3.4, which didn’t need to take up so much space, and could be put in the appendix), but if what I said above is correct, then it seems like Thm3.9 should be the *only* theorem we really care about. If not, then at the very least the limitations of Thm3.4 should be more clearly spelled out in the main text.
- **From token-wise to full sequence-to-sequence approximation** Step 2 in Sec3.3 is the one I’m struggling the most understanding. From Thm3.4/3.9 you managed to build a *token-wise* approximation to a ReLU, ie Att(xn)\sim ReLU(xn) \forall n=1…N. But here you need to represent a ReLU which depends on the *whole* input sequence at once. This requires at the very least to collapse information from the whole sequence into a single token. By skimming through AppG5, it seems like indeed this is the role of the interweaving linear layers A1, A2, but (as we established before) such sequence-wise linear operations are not “valid” in a Transformer setting (from G.42, it looks like A2 is expanding the sequence length). I admit due to time constraints I couldn’t go through the whole detailed proof in the appendix, so I might be missing something, in which case please do correct me. Nonetheless, I reckon Thm3.14 should provide the “valid” version, relying on the same trick of swapping single-head with multi-head (as in Thm3.4 vs Thm3.9), but then in L472 you need linear layers Eij that act as (tensor) one-hot selectors: these are full tensor operators (acting both sequence- and token-wise). I understand these are mainly technicalities, but again, if we’re trying to answer the question of “would a Transformer be able to implement this mechanism in practice?”, then the answer would remain “no”.
- **Limitations and comparison with previous work** In L92 and AppE you make a point of separating your result from previous work, highlighting in particular how, contrarily to [Yun et al., 19], you don’t require a (large) number of stacked layers to achieve UAP. However, in your approach you’re effectively swapping depth-complexity for head-complexity. Even from Thm3.9 it’s quick to realise how the number of attention heads necessary to achieve the desired level of accuracy directly grows with this accuracy. This is (potentially) worse than the logarithmic-power growth highlighted in [Takakura and Suzuki,23]. This is a limitation of your method, and should be highlighted. Moreover, from a practitioner’s point of view, rather than sheer number of layers / heads / feature components, a comparison of the total number of parameters required to assemble the architecture required by your approximation would be more useful.


**Clarity**

I have many reserves regarding the choices made in structuring the paper. As mentioned above, the focus should be on the most relevant Thm3.9 and Thm3.14, as they claim to pertain to the “real” Transformer architecture, rather than the “faulty” Thm3.4 single-head counterpart: including them, rather than simplifying the exposition, ends up confusions it and—most importantly—occupying relevant space in the main text. While I appreciate the attempt to provide high-level intuition behind the proofs—as in Fig1, and in the step-by-step explanations—the additional fleshed out technical details in the main don’t provide much more information, and could be directly incorporated in the step-by-step descriptions. The description of the core result is interrupted abruptly, without leaving room for a much-needed discussion of its applicability and limitations, nor the experimental results to help ground the theory, and relegating both to the appendix. Because of this, the overall impression is that this work has gone through a hastily revision, more concerned with meeting the page limit than with providing a cohesive, self-contained story, and the overall final result lacks in structure and clarity.


**The role of the Appendix**

Please don’t treat the Appendix as a free pass to extend the paper. This is ultimately unfair towards the other authors, who struggled to fit their content within the page limit. Particularly, I’ve never seen before a result being listed as core contribution and yet never mentioned in the main paper: if the application of your technique to in-context learning is a relevant contribution, as you report in Abstract and Introduction, then AppB should fit in the main.



**Minor**
- The Truncated Linear Function you introduce in Def3.1 is normally called a ramp function. Imho, this name is much more common than the one you provide
- L83: “Our theory only assumes the compactness of the target function’s domain”: more strictly, it relies on regularity of the target function (Lem3.11 requires f continuous, for example)
- L1091 “For any continuous sequence-to-sequence function, removing the factorial term n! in the denominator leaves the remaining term proportional to a power of n” If the goal of this comment is to highlight that this approximation requires a large number of layers, this is not a valid way to go on about it: the factorial term is itself proportional to a power of n (see Stirling’s approximation), and it helps counterbalancing the numerator. The main issue is the (1/\delta)^n term

**Details Of Ethics Concerns:**

//

---

> ### Author Response · Authors · 2025-11-25
> **Rebuttal 1: Clarity Concerns**
>
> ### We thank the reviewer for the detailed review. In response, we have addressed all concerns and questions in the following replies and revisions in the updated draft. All changes from the originally submitted version are highlighted in blue in the revised PDF.
>
> ---
>
> >### ***W1***: *The paper would benefit from a strong revision to improve its structure and presentation*
> >### ***W3***: *Most importantly, while the overall proof strategy is clear, I found it difficult to verify its validity, also in light of the reduced clarity.*
> >### ***Question about clarity***: *I have many reserves regarding the choices made in structuring the paper. As mentioned above, the focus should be on the most relevant Thm3.9 and Thm3.14, as they claim to pertain to the “real” Transformer architecture, rather than the “faulty” Thm3.4 single-head counterpart: including them, rather than simplifying the exposition, ends up confusions it and—most importantly—occupying relevant space in the main text. While I appreciate the attempt to provide high-level intuition behind the proofs—as in Fig1, and in the step-by-step explanations—the additional fleshed out technical details in the main don’t provide much more information, and could be directly incorporated in the step-by-step descriptions. The description of the core result is interrupted abruptly, without leaving room for a much-needed discussion of its applicability and limitations, nor the experimental results to help ground the theory, and relegating both to the appendix. Because of this, the overall impression is that this work has gone through a hastily revision, more concerned with meeting the page limit than with providing a cohesive, self-contained story, and the overall final result lacks in structure and clarity.*
>
> **Response:**
>
> Thank you for the suggestion. **Since we deem the interpolation is our main contribution, we believe it’s necessary to highlight them in the main text.** Moreover, it’s equally important to provide enough intuition to demonstrate our proof.
>
> **However, we acknowledge the reviewer's concern on clarity. To further improve, we have done several revisions accordingly.** These includes:
>
> * `line 95-100`: Adding a `Roadmap of Our Theoretical Results` paragraph in the introduction to show the relations between all core theoretical results.
>
> * `line 216-243`: Updating `Figure 1` and its caption to show a more illustrative visualization on the interpolation scheme. `Figure 1` now align directly to the 5 steps in the proof sketch of `Theorem 3.4`.
>
> * `line 277-286`: Rewrting the explanation of $k$, $\tilde{k}$ and $G$ in the theorem definition and `Remark 3.7` to make their relation clearer.
>
> * `line 131-142` and `line 270-276`: Adding clarifications on (1) the role of `Theorem 3.4` and its use of sequence-wise linear map, and (2) `Theorem 3.9` is a more  transformer-native setting, into the opening paragraph of `Section 3` and `Remark 3.6`. We hope this improves the discussion on limitation and applicaility.
>
> * `line 1584 - 1587:` Adding a discussion on the way to replace sequence-wise $A$ in `Theorem 3.4` in `Remark G.3` to improve the validity of `Theorem 3.4`.
>
> * `Appendix I:` Extending the seq-to-seq UAP result based on `Theorem 3.9` instead of `Theorem 3.4` in `Appendix I`, and adding it into the Contribution to echo reviewer's suggestion regarding applicaility. We respectfully remind the reviewer, as stated in both our abstract and intro, the focus of this paper is the expressiveness of "softmax attention", not "Transformer." Thus, after careful considerations, we choose to keep our main text attention-focus and defer the Transformer-native extension to appendix. This keeps the moral of our paper more consistent.
>
> *  `page 10`: Moving the main contribution ICL section to the 10th page of the revision as a new `Section 4`.
>
> * Conduct careful proofreading to fix typos/improve notation causing confusion/expand derivation to enhance clarity and validity
>
>
> We hope that these revisions make our paper more clear and address your concerns!

---

> ### Author Response · Authors · 2025-11-25
> **Rebuttal 2: Questions Part I**
>
> > ### ***Q1***: *Prepending linear layers. I believe this proof mainly aims at describing the properties of Attention, as it appears inside a Transformers architecture (otherwise its usefulness would be rather reduced). If this is the case, then the shape of the linear Layer A(X) used in Thm3.4, is not a valid one: by looking at Fig1 (and in general at its defined shape in L205, A:R^{d\times n}->R^{… \times p}), the layer expands the input tensor sequence-wise, which is not a valid operation in Transformers. In all fairness, you yourselves point this out in AppB and Remark G3, and highlight Thm3.9 (the multi-head version of UAP for attention) as the real “valid theorem” for the sake of proving UAP in a Transformer setting. While indeed the A in L1721 is a valid token-wise operator, due to its bias term it’s a rather unorthodox one—but I agree it can be seen as a special PE. In light of all of this, I understand why you decided to include Thm3.9 in the main (where at a first read it looked like a secondary extension to Thm3.4, which didn’t need to take up so much space, and could be put in the appendix), but if what I said above is correct, then it seems like Thm3.9 should be the only theorem we really care about. If not, then at the very least the limitations of Thm3.4 should be more clearly spelled out in the main text.*
>
> **Response:**
>
> Thank you for the detailed reading and the feedback. **Your understanding on the role of `Theorem 3.4` and `Theorem 3.9` is correct.** **`Theorem 3.4` serves as the simplest illustrative example to demonstrate our interpolation selection techniques.** It clearly demonstrates softmax attention to select interpolation point to facilitate universal approximation. **Such clarity is much much harder to achieve in the multi-head setting.** For example,multi-head version will complicate Figure 1 and is less likely to demonstrate our technique. To be concrete, `Theorem 3.9` require more complicated boundary discussion (`Lemma G.6, Remark G.10/G.11` in the revision).
>
> However, **we do agree the current main text should be clearer on the relation of two theories. As a result, we have included this discussion in additional few modifications in the main text**:
>
> - Add an explicit remark after `Theorem 3.4` that
>     - it serves as a simplest illustrative example to demonstrate our interpolation selection techniques,
>     - states that the linear map A is sequence-wise and therefore not a standard layer in transformer, and
>     - (points to `Theorem 3.9` as the version compatible with practical transformers
>
> - Add a short “theorem roadmap” at the end of `Section 1` / Modify the start of `Section 3` that explains the roles of `Theorem 3.4` and `Theorem 3.9`.
>
> We hope these revisions address your concerns about `Theorem 3.4`, and welcome any further discussion.
>
> We also provide a new seq-to-seq UAP result based on `Theorem 3.9`, please see below (and `Appendix I`) for more discussion.

---

> ### Author Response · Authors · 2025-11-25
> **Rebuttal 3: Questions Part II**
>
> ---
>
> > ### ***Q2***: *From token-wise to full sequence-to-sequence approximation Step 2 in Sec3.3 is the one I’m struggling the most understanding. From Thm3.4/3.9 you managed to build a token-wise approximation to a ReLU, ie Att(xn)\sim ReLU(xn) \forall n=1…N. But here you need to represent a ReLU which depends on the whole input sequence at once. This requires at the very least to collapse information from the whole sequence into a singletoken. By skimming through AppG5, it seems like indeed this is the role of the interweaving linear layers A1, A2, but (as we established before) such sequence-wise linear operations are not “valid” in a Transformer setting (from G.42, it looks like A2 is expanding the sequence length). I admit due to time constraints I couldn’t go through the whole detailed proof in the appendix, so I might be missing something, in which case please do correct me. Nonetheless, I reckon Thm3.14 should provide the “valid” version, relying on the same trick of swapping single-head with multi-head (as in Thm3.4 vs Thm3.9), but then in L472 you need linear layers Eij that act as (tensor) one-hot selectors: these are full tensor operators (acting both sequence- and token-wise). I understand these are mainly technicalities, but again, if we’re trying to answer the question of “would a Transformer be able to implement this mechanism in practice?”, then the answer would remain “no”.*
>
> **Response:**
>
> We thanks the reviewer for the detailed investigation and question! Indeed, $A_1$ is a sequence-wise operation that isn’t used in the standard transformer. However, we remark that, **our original goal is not investigate the transformer UAP ability, but the expressiveness of attention-only architecture.** The use of linear transformation is also clearly stated in the first sentence of abstract, and the beginning of the intro.
>
> We agree with reviewer that the result will have more impact if only consider the component in standard transformer. As you mention, that is the purpose of `Theorem 3.9` and the discussion section in original submission. Although we state
> > “*We remark that our sequence-to-sequence universal approximation in Section 3.3 uses the single-head result of Theorem 3.4 for simplicity of presentation. The same proofs hold if we replace it with the multi-head result of Theorem 3.9. That is, the two theorems are interchangeable for establishing universal sequence-to-sequence approximation.*”
>
> **We totally agree with reviewer this statement will be more solid if one can prove seq-to-seq UAP by using `Theorem 3.9` only.**
> In response, **we have extended `Section 3.3` on the seq-to-seq UAP result without the use of sequence-wise linear transform in our latest revision in Appendix I.**
>
> We sketch our proof here:
>
> * We first show attention approximate a sequence-wise operation (`Lemma I.1`).
> * We then show three-layer attention achieve seq-to-seq UAP by:
>     * first-layer attention approximate the pre-activation of ReLU neural network
>     * second-layer attention mix the information across token
>     * third-layer attention layer implement ReLU and the final linear combination to approximate ReLU neural network
>
>
> In response to your **concerns about $E^{ij}$ in `Theorem 3.14`**, you are absolutely correct. This is mainly for technicality. However, we remind the reviewer, throughout our paper (especially in abstract and intro), we clearly position this paper as a expressiveness theory paper of softmax attention. We did not ask the question "would a Transformer be able to implement this mechanism in practice?" We argee such questions are interesting and important. As highlighted above, **our new `Theorem I.2` serves to address this concern (the same result/proof via a more "Transformer-like" route).**

---

> ### Author Response · Authors · 2025-11-25
> **Rebuttal 4: Questions Part III**
>
> > ### ***Q3***: *Limitations and comparison with previous work In L92 and AppE you make a point of separating your result from previous work, highlighting in particular how, contrarily to [Yun et al., 19], you don’t require a (large) number of stacked layers to achieve UAP. However, in your approach you’re effectively swapping depth-complexity for head-complexity. Even from Thm3.9 it’s quick to realise how the number of attention heads necessary to achieve the desired level of accuracy directly grows with this accuracy. This is (potentially) worse than the logarithmic-power growth highlighted in [Takakura and Suzuki,23]. This is a limitation of your method, and should be highlighted. Moreover, from a practitioner’s point of view, rather than sheer number of layers / heads / feature components, a comparison of the total number of parameters required to assemble the architecture required by your approximation would be more useful.*
>
> **Response**:
> Thanks for the insightful suggestions. We agree that we should include the discussion on head complexity. **Our seq-to-seq result requires the number of head growth as $O(dp)$, and the total parameters scale as $O(dnp^2)$.**
>
> Here we remark that, **the value of $p$ hinges on our desired accuracy by our `Theorem 3.4/3.9` and classical ReLU approximation result without Lipschitz assumption (to keep the estimate general). Its explicit dependence on the "target error" requires specifying the continuity properties of the target function.** Please see the details of the new `Remark E.1` in related work.
>
>
> For the concern on the comparison in related work, we are sorry for any confusion caused. To clarify, as in `line 1210-1212` in the revision:
> > “*Our work is different from these papers by removing the need for FFN from transformer to demonstrate the first universal approximation result of attention mechanism.*”,
>
> **We achieve UAP without FFN. We don’t imply better model architecture complexity.** However, we also understand your concern. **To balance the narrative, as mentioned above, we now added our head-complexity discussion into `Remark E.1`.**
>
> To elaborate more, **we highlight that the exponential growth of neuron is well-known in depth-bound FFN UAP theory [1,2].** We prove attention achieves UAP for continuous seq-to-seq functions by approximating FFNs. As a result, the exponential dependence is unavoidable if we want to approximate such a wide range of function classes.
>
> There are papers on FFN UAP that try to trade the exponential dependence of width (number of neuron) with unbounded depth [3-5] and restrict the function class [6]. We leave the circumvention of exponential dependence for future work.
>
> [1] Cybenko, George. Approximation by superpositions of a sigmoidal function. Mathematics of control, signals and systems (1989)
>
> [2] Pinkus, A. Approximation theory of the MLP model in neural networks. Acta numerica (1999)
>
> [3] Lu, Z., Pu, H., Wang, F., Hu, Z., & Wang, L. The expressive power of neural networks: A view from the width. NeurIPS (2017)
>
> [4] Lin, H., & Jegelka, S.  Resnet with one-neuron hidden layers is a universal approximator.  NeurIPS (2018)
>
> [5] Hanin, B., & Sellke, M.  Approximating continuous functions by relu nets of minimal width. (2017).
>
> [6] Shen, Z., Yang, H., & Zhang, S. (2022). Optimal approximation rate of ReLU networks in terms of width and depth. Journal de Mathématiques Pures et Appliquées (2022)
>
>
>
> Finally, we also elaborate more on the transformer works we are comparing with. Besides removing FFN, our work relaxes prior assumptions in two aspects:
>
> * **Mild Data Assumption:** As in the original submission, Line 74-76, previous works utilize the concept of contextual mapping and assume input has token-wise separation [1,2]. Furthermore, [3] assumes input data lives on a hypersphere (fixed norm). In contrast, we only assume input within a compact domain.
>
> * **No Smoothness Assumption on Target Function:** Prior works consider target function with shift-equivariant α-smoothness ([4]) or finite complexity measure ([5]), resembling classical smoothness assumptions. In contrast, we only assume continuity on target function.
>
> [1] Yun et al. Are transformers universal approximators of sequence-to-sequence functions?. ICLR (2020)
>
> [2] Kajitsuka & Sato. Are Transformers with One Layer Self-Attention Using Low-Rank Weight Matrices Universal Approximators?. ICLR (2024)
>
> [3] Petrov et al. Prompting a pretrained transformer can be a universal approximator. ICML (2024)
>
> [4] Takakura & Suzuki. Approximation and estimation ability of transformers for sequence-to-sequence functions with infinite dimensional input. ICML (2023)
>
> [5] Jiang & Li (2024). Approximation rate of the transformer architecture for sequence modeling. NeurIPS (2024)

---

> ### Author Response · Authors · 2025-11-25
> **Rebuttal 5: Questions Part IV**
>
> > ### ***W2***: *Many relevant discussions and results (including mentioned core contributions) are relegated to the appendix.*
> > ### ***Q4***: *Please don’t treat the Appendix as a free pass to extend the paper. This is ultimately unfair towards the other authors, who struggled to fit their content within the page limit. Particularly, I’ve never seen before a result being listed as core contribution and yet never mentioned in the main paper: if the application of your technique to in-context learning is a relevant contribution, as you report in Abstract and Introduction, then AppB should fit in the main.*
>
> **Response:**
> Thanks for the suggestion, we agree with reviewer the main contribution should be in the main text. **We have moved the ICL part to the 10th page of the revision.**
>
> ---
>
> > ### ***Q5***: *L1091 “For any continuous sequence-to-sequence function, removing the factorial term n! in the denominator leaves the remaining term proportional to a power of n” If the goal of this comment is to highlight that this approximation requires a large number of layers, this is not a valid way to go on about it: the factorial term is itself proportional to a power of n (see Stirling’s approximation), and it helps counterbalancing the numerator. The main issue is the (1/\delta)^n term*
>
> **Response:**
> Thanks for pointing this out! **You are absolutely correct! The wording was confusion.** Instead of "proportional to a power of $n$", what we meant was "growing exponentially with $n$". We have changed it in the revision as below:
> > *"For any continuous sequence-to-sequence function, removing the factorial term $n!$ in the denominator leaves the remaining term growing exponentially with $n$."*
>
> **This sentence aims to report the complexity  of their construction ($O(n(1/\delta)^{dn})$ of layer) in the same setting as us (continuous seq-to-seq function).** This discussion can also be found at section `4.4 TIGHTNESS OF CONSTRUCTIONS` of [1].
>
> [1] Yun et al. Are transformers universal approximators of sequence-to-sequence functions?. ICLR (2020)
>
> ---
>
> > ### ***Q6***: *The Truncated Linear Function you introduce in Def3.1 is normally called a ramp function. Imho, this name is much more common than the one you provided*
>
> **Response:**
> Thanks for bringing this up. We survey the literature and find the definition of ramp function is the ReLU function in machine learning community. The ramp function is a special case of our definition of of truncated linear model (a = 0). We include this more general truncated linear model so it subsum several activation function, see appendix H for examples. However, we do agree ramp function is more common in mathematical community, hence we added it into `appendix H` to improve clarity. We thanks the reviewer for the suggestion.
>
> ---
>
> > ### ***Q7***: *L83:Our theory only assumes the compactness of the target function’s domain”: more strictly, it relies on regularity of the target function (Lem3.11 requires f continuous, for example)*
>
> **Response:**
> Thanks for pointing this out. To be more precise, we have change `line 83-84` to be
> > *"We highlight that our results are general and require minimal assumptions.
> Our theory assumes only the target function is continuous on the compact domain.""*
>
> ---
>
> Thank you for your review and your meticulous attention to detail (really helpful!). We have addressed all your concerns and questions with utmost care. We hope our revisions and clarifications meet your expectations. Please let us know if there is anything else we can clarify better! We will be more than happy to discuss more :)

---

### Official Review · Reviewer_Jq31 · 2025-11-06

**Soundness:** 3
**Presentation:** 3
**Contribution:** 3
**Rating:** 8
**Confidence:** 1

**Summary:**

The paper shows that attention-only architectures, with only linear transformations, achieve sequence-to-sequence universal approximation on compact domains.
The key technique is an interpolation-based construction in which the softmax serves as a near-argmax selector over a fixed grid of anchors. This converts self-attention into a piecewise-linear generalized ReLU approximator, establishing that attention alone suffices for universal approximation.
The paper also shows that multi-head attention improves the approximation rate to O(1/H) and extends the method to approximate in-context gradient descent.

**Strengths:**

1. This work isolates the expressive power of attention without feed-forward layers, a minimalistic configuration not covered by previous proofs.
2. This work formalizes "anchor selection" within attention, turning softmax into a continuous selection mechanism that generalizes ReLU behavior (Theorems 3.4 & 3.9).
3. Synthetic experiments confirm the predicted O(1/p) and O(1/H) error rates and visualize one-hot-like attention heatmaps.
4. The theoretical analysis in this paper is exceptionally solid with minimal reliance on unverifiable assumptions.
5. The boundary conditions for the softmax–hardmax approximation (Lemma F.1) are explicit, and the relationships among parameters p, H, and beta are carefully derived.
6.  The interpolation-anchor construction and separation of interpolation and softmax errors demonstrate a high level of precision.
7. The appendix provides complete supporting derivations.

**Weaknesses:**

1. Achieving small error requires large p (number of anchors) or large beta (inverse temperature). Constants in the asymptotic rates are unspecified, making practical implications unclear.
2. Experiments use synthetic data only.

Comment:
The construction (Theorem 3.4) rely on a sequence-wise linear map A that expands columns, which is the less practical form.
 I understand that this structure is primarily motivated by interpretability, so I do not consider it a serious limitation.
although a token-wise version (Theorem 3.9) exists, this is the less practical form.

**Questions:**

I understand that this paper achieves attention-only universal approximation through a research idea fundamentally different from previous studies on universal approximation for LLMs.
Although I am not sure how practically meaningful it is to prove universal approximation for an attention-only model, I believe the technical contribution is substantial. From that perspective, I would like to ask the following questions.

Q1.
I understand that demonstrating attention-only universal approximation is valuable because it reveals the intrinsic expressive power of the attention mechanism itself. However, if there are other intended meanings or motivations behind proving attention-only universal approximation, could you please elaborate on them?

Q2.
I understand well the general limitations of constructive universal approximation theorems, and this paper shares the same type of limitations as prior works, for example, being an existence proof that does not necessarily imply the model can be learned via gradient descent.  Also, in this paper, attention uses a temperature parameter to make softmax behave like a hard max; this approach is not unique to this work and appears in other studies as well. Nevertheless, hard-max operations are problematic when applying gradient-based optimization. Since some existing universal approximation theorems avoid using hard max, could you clarify the conceptual or technical significance of proposing a proof that explicitly depends on hard-max behavior?

Q3.
In the attention + feed-forward (FFN) configuration, attention acts as a context mapping, while the FFN plays the role of memory and recall. This decomposition is intuitive and is consistent with known interpretations that link the FFN to memory mechanisms like key-value memory, independent of universality. Demonstrating that attention alone can approximate ReLU networks clearly shows its expressive capacity. However, does your analysis uncover any new property or potential role of attention beyond its known expressive power?
Could you explain what new conceptual insight about attention arises from your construction?

Q4.
Improving approximation accuracy in your construction requires substantial growth of the key parameters, namely the inverse temperature beta, the number of interpolation anchors p, and the number of heads H. This scaling seems far from what realistic Transformers typically employ.
Could you provide at least a rough quantitative estimate or empirical sense of how large these parameters would need to be to achieve a reasonable approximation accuracy (e.g., a target error epsilon)? Even an order-of-magnitude discussion would help clarify how far the theoretical regime is from practical parameter ranges.

---

> ### Author Response · Authors · 2025-11-25
> **Rebuttal 1: Weakness**
>
> ### We thank the reviewer for the detailed review. In response, we have addressed all concerns and questions in the following replies and revisions in the updated draft. All changes from the originally submitted version are highlighted in blue in the revised PDF.
>
> > ### ***W1***: *Achieving small error requires large p (number of anchors) or large beta (inverse temperature). Constants in the asymptotic rates are unspecified, making practical implications unclear.*
>
> **Response:**
> We thank the reviewer for the comments. We address the mentioned weakness in several points:
>
> * **Large $p$ and $\beta$:**
> We emphasize that **the requried large model architecture/complexity to achieve arbitrarily small error for general function class (i.e, continuous function) is common in approximation theory.** For example, the UAP result of FFN requires the number of neurons to scale exponentially with $\epsilon$ to approximate any continuous function [1,2]. Second, in the first paper prove the UAP of transformer, their construction also shows that the number of layers is exponential in the worst-case (Section 4.4) [3].
>
> * **Constants in the asymptotic rates:**
> In `Theorem 3.4` and `Theorem 3.9`, the interpolation error term is bounded by $(b-a)/p$, so the asymptotic rate is $O(1/p)$ with constant $b-a$. The softmax error $\epsilon_0$ term can be arbitrarily small for large enough $\beta$. This is why we did not write it in big-O form in the main statement. For completeness, we derive an explicit $\beta$ at the end of the proof of `Theorem 3.4` as new `Remark G.4`, by making $\epsilon_0$ scale with $1/p$. In particular, if we choose $\beta \geq \frac{2p^2}{(b-a)^2}\ln(p(p-2))$, the total approximation error is bounded by $\big(\max{(∣a∣,∣b∣)}+(b−a)\big) \cdot 1/p = O(1/p)$, that is, it scales as $O(1/p)$ with all the constant written out.
>
>
>
> [1] Cybenko, George. Approximation by superpositions of a sigmoidal function. Mathematics of control, signals and systems (1989)
>
> [2] Pinkus, A. Approximation theory of the MLP model in neural networks. Acta numerica (1999)
>
> [3]  Yun et al. Are transformers universal approximators of sequence-to-sequence functions?. ICLR (2020)
>
> ---
>
> > ### ***W2***: *Experiments use synthetic data only.*
>
> **Response:**
> Thank you for the comments. **To verify our theory, we should have a setting in which we have absolute control. Using real-world data will introduce unwanted confounding factors that defeat the purpose of validating our theory.**
>
> We wish to emphasize that **the main goal of the paper is to investigate the fundamental capabilities of attention mechanism.** Along this way, we theoretically extend the expressive power of attention by proposing a novel non-trivial proofing technique. We believe the current proof-of-concept experiment is enough to support the paper’s result.
>
> We agree with the reviewer that empirical evidence can increase practical value. For that reason, we included `Figure.3`. It shows that with minimal anchor seeding, a standard optimizer sharpens the one-hot attention pattern predicted by the theory. Extending this sanity check to real-world tasks is an exciting next step, but we believe the present theoretical result already fills a clear gap by proving, for the first time, that **softmax attention alone is a universal sequence-to-sequence approximator**.

---

> ### Author Response · Authors · 2025-11-25
> **Rebuttal 2: Question Part I**
>
> > ### ***Q1***: *I understand that demonstrating attention-only universal approximation is valuable because it reveals the intrinsic expressive power of the attention mechanism itself. However, if there are other intended meanings or motivations behind proving attention-only universal approximation, could you please elaborate on them?*
>
> **Response:**
> Thanks for the great question. Besides UAP, one of the motivations is from the recent studies on the computational and statistical theory of generative AI. Modern generative models (diffusion and flow matching) still utilize Transformer-based architectures like DiT. Consequently, analyzing attention mechanisms directly impacts the theoretical understanding of generative AI.
>
>
>
> For instance, `Remark 3.4` of [1] implies that prior UAP results of the transformer are insufficient to obtain sharp statistical rates (sample complexity of DiT training). They pointed out that the fundamental barrier is the “contextual mapping” technique used by prior studies [2,3]. In these frameworks, softmax attention does not really contribute to the expressiveness of Transformer. The approximation ability is mainly contributed by FNN. The softmax attention mainly serves as a unique ID identifier rather than an approximator. This motivates us to develop attention-only UAP theory.
>
> [1] Hu, J. Y. C., Wu, W., Li, Z., Pi, S., Song, Z., & Liu, H. (2024). On statistical rates and provably efficient criteria of latent diffusion transformers (dits). NeurIPS (2024)
>
> [2]  Yun et al. (2019). Are transformers universal approximators of sequence-to-sequence functions?.
> ICLR (2020)
>
> [3] Kajitsuka & Sato (2023). Are Transformers with One Layer Self-Attention Using Low-Rank Weight Matrices Universal Approximators?.
> ICLR (2024)
>
> ---
> > ### ***Q2***: *I understand well the general limitations of constructive universal approximation theorems, and this paper shares the same type of limitations as prior works, for example, being an existence proof that does not necessarily imply the model can be learned via gradient descent. Also, in this paper, attention uses a temperature parameter to make softmax behave like a hard max; this approach is not unique to this work and appears in other studies as well. Nevertheless, hard-max operations are problematic when applying gradient-based optimization. Since some existing universal approximation theorems avoid using hard max, could you clarify the conceptual or technical significance of proposing a proof that explicitly depends on hard-max behavior?*
>
> **Response:**
> Thanks for the insightful question. First, in our interpolation scheme, intuitively we want to do some selection. We treat softmax as a selection mechanism to select the closest interpolation point from the KQ score matrix. Achieving the closest point selection accurately needs the hard-max behavior. We have reflected this intuition in our `Figure 1` accordingly. Also, to improve clarity, we have updated `Figure 1` and its caption in the latest revision.
>
> However, we remark that our theorem aims to approximate an **arbitrary continuous function** on a compact set. Such a function may have extremely sharp transitions, so any universal construction needs to approximate very “steep” behaviors somewhere. This implies we need a fine grid (more grid points) to capture these sharp changes, resulting to a sharp softmax (larger $\beta$) to distinguish neighboring grids, see `line 2862-2865` and `line 2925-2939` in our latest revision.
>
> In contrast, if the target function is smoother (small Lipschitzness), since the value of
> the function doesn't change much within a larger interval, we can work with a coarser grid. This implies:
> * fewer grid points $|G_D|$ (fewer $p$),  then softmax attention needs only a modest $\beta$
>
> In summary, the “almost hard‑max” regime in the theory **is not claiming that practical models must use a large $\beta$. It reflects the fact that we prove universality for a very broad function class, including functions with sharp local behavior.** For smoother target functions, the same construction can achieve a given approximation error with smaller $\beta$.

---

> ### Author Response · Authors · 2025-11-25
> **Rebuttal 3: Questions Part II**
>
> > ### ***Q3***: *In the attention + feed-forward (FFN) configuration, attention acts as a context mapping, while the FFN plays the role of memory and recall. This decomposition is intuitive and is consistent with known interpretations that link the FFN to memory mechanisms like key-value memory, independent of universality. Demonstrating that attention alone can approximate ReLU networks clearly shows its expressive capacity. However, does your analysis uncover any new property or potential role of attention beyond its known expressive power? Could you explain what new conceptual insight about attention arises from your construction?*
>
> **Response:** In response to a potential role beyond expressness, our analysis (`Theorem 3.4/Theorem 3.9`) implies attention’s mechanical operation rule as selecting and routing (${\rm Softmax}$ chooses an anchor, $V$ routes it).
>
> In response to new conceptual insights, we show that softmax attention itself is a powerful information processor through accurate information selection and routing. To see this, please see our `Figure 1` in the revision. In there, our goal was to use softmax attention mechanism to approximate $n$ truncated linear models $\mathrm{Range}\_{[a,b]}(w\_i^\top x\_i + t\_i)$ for $i \in [n]$, and hence establish universality. We first divide the output range $[a,b]$ into $p$ interpolation points, and encode them into the value matrix $V$. Then, we treat the attention score ${\rm Softmax}(K^\top Q)$ as a *selector* to select an interpolation point closest to the desired output from $V$. Specifically, each column of ${\rm Softmax}(K^\top Q)\_{:,i}$ (for $i \in [n]$) approximates an one-hot vector $e\_{k\_i}$, where $k\_i$ is the index of closest interpolation point to $\mathrm{Range}\_{[a,b]}(w\_i^\top x\_i + t\_i)$. Hence, when multiplying with $V$, ${\rm Softmax}(K^\top Q)$ selects out the closest interpolation points for every truncated linear model from $V$.
>
>
>
> This scheme gives us much flexibility beyond `Theorem 3.4` and `Theorem 3.9`. Here we use `Theorem D.1` as an example of the application of this sight.  By slightly modifying the proof of `Theorem 3.4`, one sees that we can change $(w,t)$ from model parameter into the input. Hence a single attention with fixed parameters can implement $(x,w,t) \to \text{Range}(w^\top x +t)$ for many different $(w,t)$ supplied in the input. Stacking these gives attention blocks simulating FFNs whose effective parameters are read from the input. This is the kind of computation required for in-context learning. Hence, the selection-and-routing mechanism not only builds the foundation of the expressive capacity of attention, as the reviewer mentions. It also shed light on the mechanism behind in-context learning.
>
>
>
>
> ---
>
> > ### ***Q4***: *Improving approximation accuracy in your construction requires substantial growth of the key parameters, namely the inverse temperature beta, the number of interpolation anchors p, and the number of heads H. This scaling seems far from what realistic Transformers typically employ. Could you provide at least a rough quantitative estimate or empirical sense of how large these parameters would need to be to achieve a reasonable approximation accuracy (e.g., a target error epsilon)? Even an order-of-magnitude discussion would help clarify how far the theoretical regime is from practical parameter ranges.*
>
> **Response:**
>
> Thanks for the insightful suggestions. We agree that we should include the model complexity in the main text. Our seq-to-seq result requires the number of head growth as $O(dp)$, and the total parameters scale as $O(dnp^2)$. We have added a new discussion about model complexity in `Remark E.1`.
>
> Here we remark that the value of $p$ hinges on our desired accuracy by our `Theorem 3.4/3.9` and classical ReLU approximation result without Lipschitz assumption (to keep the estimate general). Its explicit dependence on the "target error" requires specifying the continuity properties of the target function.
>
> Please see the details of the `Remark E.1` in the related work.
>
> ---
> Thank you again for the valuable feedback. We hope our clarification addresses your concerns and welcome further discussion.

---

### Author Response · Authors · 2025-11-25
**Global Rebuttal Response**

Dear Reviewers and Chairs:

We thank the reviewers for the helpful comments and detailed reviews. We have answered all the questions and addressed all the concerns in detail in rebuttal and revisions. Any changes or modifications made to the submitted version are highlighted in blue in this updated draft.

In response to the reviewers' suggestions, we have made revisions to improve the clarity of the paper.
These include additional explanations, remarks, paragraphs, and an updated figure/caption to help the reader build intuition on our interpolation scheme. We also added a more detailed discussion on model complexity and expanded the in-context learning result, reorganizing it into the main text. **Most importantly, a new section, `Appendix I`, has been added to present a more Transformer-naive softmax universal approximation result.**

---

## Revision Details

**Major revisions include:**

* **A New Section of seq-to-seq universal approximation (UAP) results based on Theorem 3.9.** [[NV7d-Q2](https://openreview.net/forum?id=8cj7ydwaaK&noteId=dolcwN4npo), [dffi-W4](https://openreview.net/forum?id=8cj7ydwaaK&noteId=yFn4ORo816)]
    * `Pages 72-84:` Besides the seq-to-seq UAP result `Theorem 3.14` based on `Theorem 3.4`, we deliver a new seq-to-seq UAP result in `Appendix I` based on `Theorem 3.9`.
    * We show that 3-layer softmax attention suffices to do this: one attention layer implement sequence-wise operation to mix information of token, two other layer construct the ReLU linear model and combined them properly.
    * `Page 24:` We calculate the construction complextiy in `Remark E.1` for fair comparision with the literature.

* **A New Discussion on Implications from our ICL Results.** [[NV7d-W2/Q4](https://openreview.net/forum?id=8cj7ydwaaK&noteId=ka86pjAWDg), [dffi-W1](https://openreview.net/forum?id=8cj7ydwaaK&noteId=a8QVrx2CaN)]
    * `Pages 10:` Move in-context learning result to the main text as new `Section 4`.
    * `Pages 21-22:` To make the ICL result more complete since it is one of our main contribution, we add a paragraph to discuss how our result lead to (i) simulation of learning algorithms in-context and (2) meta learning composited tasks in-context. These are important in understand the in-context learning theory to solve complicated task.

* **Improve Clarity of `Section 3` on Interpolation Scheme.** [[NV7d-Clarity](https://openreview.net/forum?id=8cj7ydwaaK&noteId=JAy7KmkRWV), [dffi-W3](https://openreview.net/forum?id=8cj7ydwaaK&noteId=Awog2JkHy3)]
    * `Pages 5:` Updating `Figure 1` and its caption to show a more illustrative visualization on the interpolation scheme. `Figure 1` now align directly to the 5 steps in the proof sketch of `Theorem 3.4`.

    * `Pages 6 and 14:` Rewrting the explanation of $k$, $\tilde{k}$ and $G$ in the theorem definition, `Remark 3.7` and `Table 1` to make their relation clearer.

* **Additional Discussion on the Role of `Theorem 3.4`.** [[NV7d-Q1](https://openreview.net/forum?id=8cj7ydwaaK&noteId=[FOJoV48Sp), [dffi-W4](https://openreview.net/forum?id=8cj7ydwaaK&noteId=Awog2JkHy3)]
    * `Pages 2, 3, 6:` Extend `Remark 3.6`, opening paragraph of `Section 3`, and a new `Roadmap of Theoretical Results` paragraph in `Section 1` to make relation between `Theorem 3.4` and `Theorem 3.9` clearer.
    * `Pages 30:` Extend `Remark G.3` to show elimination of the sequence-wise map $A$ is doable.


* `Pages 24: ` A new discussion in `Secion B` on **model complextiy** of the seq-to-seq UAP results (`Theorem 3.14, Theorem I.2`) to achieve completeness.
[[Jq31-Q4](https://openreview.net/forum?id=8cj7ydwaaK&noteId=6tLOU4IeTP), [NV7d-Q3](https://openreview.net/forum?id=8cj7ydwaaK&noteId=Va0Himue2Z), [dffi-W2](https://openreview.net/forum?id=8cj7ydwaaK&noteId=a8QVrx2CaN)]

**Minor revisions include:**


* Proofreading the manuscript and fixing all identified typos and grammatical errors by reviewers and authors.

* `Pages 32, 35, 51, 56 and 59: ` Add more explanations or expand the derivations in `Appendix G` to enchance clarity. [[NV7d-Clarity](https://openreview.net/forum?id=8cj7ydwaaK&noteId=JAy7KmkRWV)]


* `Pages 3:` Link the notation table in `Appendix A` into `Secion 2` for better reference. [[dffi-W3](https://openreview.net/forum?id=8cj7ydwaaK&noteId=Awog2JkHy3)]

* `Pages 2:` Include continuous function assumption in the introduction. [[NV7d-Q7](https://openreview.net/forum?id=8cj7ydwaaK&noteId=ka86pjAWDg)]

* `Pages 34:` Add asymptotic rate constant calculation in `Remark G.5`.  [[Jq31-W1](https://openreview.net/forum?id=8cj7ydwaaK&noteId=ul4MDJilcj)]

---

We hope these revisions address the reviewers' concerns and improve the overall quality of our paper.

---

### Comment · Area_Chair_FS4X · 2025-11-28

Dear Reviewers,

The discussion phase is now underway, and the authors have finished uploading their responses to reviewers. If you haven't already, please carefully review the authors' responses to understand their perspectives. Engage in thoughtful, constructive discussions with authors, sharing your thoughts and seeking clarifications. Please also update your review or rating if necessary.

It is noted in the guideline that reviewers can leave comments visible to authors **until Dec 2 11:59pm AoE**. Your active participation and contribution to the ongoing discussion are highly encouraged. Thank you very much for your contribution to ICLR.

Best regards,

AC

---

> ### Author Response · Authors · 2025-11-28
>
> Thanks for sending out the reminder! Really appreciated!
>
> Best,
>
> Authors

---

### Note · Authors · 2026-01-27

**Comment:**

### **Record Clarification**

We recognize this review cycle was abnormal, and the AC may have been overloaded.


That said, the meta-review does not align with our stated scope and rebuttal record. We clarify here with utmost respect for the program chairs’ efforts:

- This paper studies **universality of softmax attention itself**. This is stated in our title, abstract, and intro. We aim for new tools to understand softmax attention better. It is therefore puzzling that the evaluations repeatedly requested “practical Transformer” criteria and real-task benchmarks. They are simply beyond that scope.


- Even so, we complied with those requests by adding Transformer-native UAP results in Appendix I. These additions appear unacknowledged in the final assessment. For example, Appendix I is mentioned as “Outstanding concerns” instead of “Addressed concerns”.


- The meta-review also treats worst-case scaling as a defect.  Yet, universality over arbitrary continuous targets typically carries a worst-case exponential price in model size. This is not specific to our construction (see, e.g., Yun et al. 2019; Kajitsuka & Sato 2023; and classical FFN universality results). Improving these worst-case rates without additional regularity assumptions is widely viewed as an open problem, not a paper-specific flaw. Our goal is mechanism-level understanding of softmax attention, as a step toward sharper target-specific theory.




- Given the constructive nature of the theory, real-world benchmarks would be incremental at best and more likely confounded (there’s no new method or model here). Our controlled experiments validate the predicted $O(1/p)$ and $O(1/H)$ trends and the near one-hot selection behavior. We use synthetic settings to isolate the mechanism, not to claim benchmark performance.

All these points were addressed in the rebuttal and incorporated into the revision.

Since post-decision public discussion is now closed, we withdraw with this note to avoid an inaccurate public record. We assume any mismatch was unintentional, but it is not fair to the results presented here.

**Withdrawal Confirmation:**

I have read and agree with the venue's withdrawal policy on behalf of myself and my co-authors.

---

### Meta-Review · Area_Chair_C4Q5 · 2026-01-07

**Summary:**

The paper establishes theoretical universal approximation properties for attention-only Transformer architectures, claiming that softmax attention mechanisms alone can approximate any continuous sequence-to-sequence function without requiring feed-forward networks. The authors present an interpolation-based construction that theoretically demonstrates how attention mechanisms can approximate piecewise-linear functions. While the theoretical framework is mathematically sophisticated and addresses an important gap in understanding attention's expressive power, the paper suffers from significant presentation issues, questionable practical relevance, and limitations in experimental validation. Despite substantial revisions addressing many reviewer concerns, fundamental issues regarding the practical applicability of the theoretical results and the lack of validation on real-world tasks remain unresolved. Based on these reasons, I suggest to reject the paper and encourage the authors to address the reviewers' concerns in the future.

**Reviewer Concerns:**

## Addressed concerns:

* The authors significantly improved the paper's structure and clarity by adding roadmap paragraphs, updating Figure 1 with more intuitive visualizations, and clarifying the relationships between different theorems.
* The authors added explicit discussions about model complexity and parameter scaling (Remark E.1), addressing questions about how the number of attention heads and total parameters scale with approximation precision.
* They clarified the relationship between Theorem 3.4 and Theorem 3.9, explicitly stating that the former serves as an illustrative example while the latter is more compatible with practical transformers.

## Outstanding concerns:

* Despite improvements, the connection between theory and practice remains weak. The constructions require exponentially many anchors and extremely large temperature parameters to achieve small approximation errors, making them impractical for real-world applications.
* The paper still lacks validation on standard benchmarks or real-world tasks. The authors defend their focus on synthetic experiments as necessary for "absolute control," but this fails to demonstrate whether their theoretical insights have practical value for actual Transformer implementations.
* Questions about the applicability to standard Transformer architectures persist. While the authors added an alternative construction in Appendix I, the main text still heavily relies on sequence-wise linear operations that don't exist in practical Transformers.

**Reviewer Scores:**

* **Reviewer Jq31**: Initially rated 8. While impressed by the theoretical rigor, this reviewer had significant concerns about practical relevance. The rebuttal addressed some theoretical questions but didn't fully resolve concerns about parameter scaling and real-world applicability. Likely updated score: 6.
* **Reviewer NV7d**: Initially rated 2. The reviewer had fundamental concerns about clarity, validity, and presentation. The authors made substantial improvements addressing many specific issues raised, particularly regarding paper structure and theorem relationships. However, concerns about practical relevance and the exponential parameter scaling remain. Likely to maintain the score.
* **Reviewer xyoo**: Initially rated 8, but properly recused themselves due to lack of expertise in approximation theory. Their assessment cannot be considered in the final decision.
* **Reviewer dffi**: Initially rated 6. The authors directly addressed their concerns about moving core content to the main text and clarified the scaling properties. However, clarity issues in explaining the core construction remain partially unresolved, and questions about practical applicability persist. Likely to maintain the score.

---

### Decision · Program_Chairs · 2026-01-26

Reject